# InversionView: A General-Purpose Method for Reading Information from Neural Activations

**Xinting Huang**
Saarland University
xhuang@lst.uni-saarland.de

**Madhur Panwar**
EPFL
madhur.panwar@epfl.ch

**Navin Goyal**
Microsoft Research India
navingo@microsoft.com

**Michael Hahn**
Saarland University
mhahn@lst.uni-saarland.de

## Abstract

The inner workings of neural networks can be better understood if we can fully decipher the information encoded in neural activations. In this paper, we argue that this information is embodied by the subset of inputs that give rise to similar activations. We propose InversionView, which allows us to practically inspect this subset by sampling from a trained decoder model conditioned on activations. This helps uncover the information content of activation vectors, and facilitates understanding of the algorithms implemented by transformer models. We present four case studies where we investigate models ranging from small transformers to GPT-2. In these studies, we show that InversionView can reveal clear information contained in activations, including basic information about tokens appearing in the context, as well as more complex information, such as the count of certain tokens, their relative positions, and abstract knowledge about the subject. We also provide causally verified circuits to confirm the decoded information.[1]

## 1 Introduction

Despite their huge success, neural networks are still widely considered black boxes. One of the most important reasons is that the continuous vector representations in these models pose a significant challenge for interpretation. If we could understand what information is encoded in the activations of a neural model, significant progress might be achieved in fully deciphering the inner workings of neural networks, which would make modern AI systems safer and more controllable. Toward this goal, various methods have been proposed for understanding the inner activations of neural language models. They range from supervised probes [2, 5, 4, 55] to projecting to model's vocabulary space [42, 7] to causal intervention [21, 54, 26, 13] on model's inner states. However, to this date, decoding the information present in neural network activations in human-understandable form remains a major challenge. Supervised probing classifiers require the researcher to decide which specific information to probe for, and does not scale when the space of possible outputs is very large. Projecting to the vocabulary space is restricted in scope, as it only produces individual tokens. Causal interventions uncover information flow, but do not provide direct insight into the information present in activations.

Here, we introduce InversionView as a principled general-purpose method for generating hypotheses about the information present in activations in neural models on language and discrete sequences, which in turn helps us identify how the information flows through the model—crucial for obtaining the algorithm implemented by the model. InversionView aims at providing a direct way of reading out

---

[1]Code is available at https://github.com/huangxt39/InversionView

38th Conference on Neural Information Processing Systems (NeurIPS 2024).

the information encoded in an activation. The technique starts from the intuition that the information encoded in an activation can be formalized as its *preimage*, the set of inputs giving rise to this particular activation under the given model. In order to explore this preimage, given an activation, we train a decoder to sample from this preimage. Inspection of the preimage, across different inputs, makes it easy to identify which information is passed along, and which information is forgotten. It accounts for the geometry of the representation, and can identify which information is reinforced or downweighted at different model components. InversionView facilitates the interpretation workflow, and provides output that is in principle amenable to automated interpretation via LLMs (we present a proof of concept in Section 4).

We showcase the usefulness of the method in three case studies: a character counting task, Indirect Object Identification, and 3-digit addition. We also present preliminary results on the factual recall task, demonstrating the applicability of our method to larger models. The character counting task illustrates how the method uncovers how information is processed and forgotten in a small transformer. In Indirect Object Identification in GPT2-Small [54], we use InversionView to easily interpret the information encoded in the components identified by Wang et al. [54], substantially simplifying the interpretability workflow. For 3-digit addition, we use InversionView to provide for the first time a fully verified circuit. Across the case studies, InversionView allows us to rapidly generate hypotheses about the information encoded in each activation site. Coupled with attention patterns or patching methods, we reverse-engineer the flow of information, which we verify using causal interventions.

## 2 Methodology

**Interpretation Framework** What information does an activation in a neural network encode? InversionView answers this in terms of the inputs that give rise to this activation (Figure 1). For instance, if a certain activation encodes solely that "*the subject is John*" and nothing else (Figure 1, right), then it will remain unchanged when other parts in the sentence change while preserving this aspect (e.g., "*John is on leave today.*" ⇒ "*John has a cute dog.*"). From another perspective, if all sentences where the subject is John are represented so similarly that the model cannot distinguish them, given one of these representations, the only information is the commonality "*the subject is John*" (assuming sentences are represented differently when it does not hold). Building on this intuition, given an activation, InversionView aims to find those

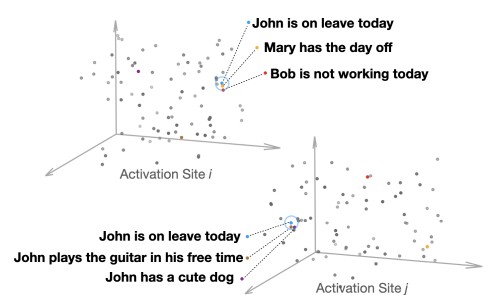

Figure 1: Illustration of the geometry at two different activation sites, encoding different information about the input. Top: the semantics of being on leave are encoded. Bottom: the information that the subject of the input sentence is John is encoded.

inputs that give rise to the same activation, and examine what's common among them to infer what information it encodes. In realistic networks, different inputs will rarely give rise to exactly the same activation. Rather, different changes to an input will change the activation to different degrees. The sensitivity of an activation to different changes reflects the representational geometry: larger changes make it easier for downstream components to read out information than very small changes. This motivates a threshold-based definition of preimages, where we consider information as present in an activation when the activation is sufficiently sensitive to it. Formally speaking, given a space $\mathcal{X}$ of valid inputs, a query input $\mathbf{x}^q \in \mathcal{X}$, a function $f$ that represents the activation of interest as a function of the input, and a query activation $\mathbf{z}^q = f(\mathbf{x}^q)$, define the $\epsilon$-*preimage*:[2]

$$B_{\mathbf{z}^q, f, \epsilon} = \{\mathbf{x} \in \mathcal{X} : D(f(\mathbf{x}), \mathbf{z}^q) \leq \epsilon\}, \tag{1}$$

where $\epsilon > 0$ is a threshold and $D(\cdot, \cdot)$ is a distance metric. Both $\epsilon$ and $D(\cdot, \cdot)$ are chosen by the researcher based on representation geometry; we will define these later in case studies. In practice,

---

[2]Strictly speaking, when $\mathbf{x}^q$ is a sequence, we study the vector $\mathbf{z}^q$ corresponding to a specific position $t$ in this sequence, i.e. $\mathbf{z}^q = f(\mathbf{x}^q)_t$ where $f(\mathbf{x}^q)_t$ represents taking the activation from the site of interest (abstracted by $f$) at position $t$ in input sequence $\mathbf{x}^q$. In this case, the preimage can more rigorously be defined as $B_{\mathbf{z}^q, f, \epsilon} = \{\mathbf{x} : \mathbf{x} \in \mathcal{X}, \exists t \in [1, |\mathbf{x}|] : D(f(\mathbf{x})_t, \mathbf{z}^q) \leq \epsilon\}$.

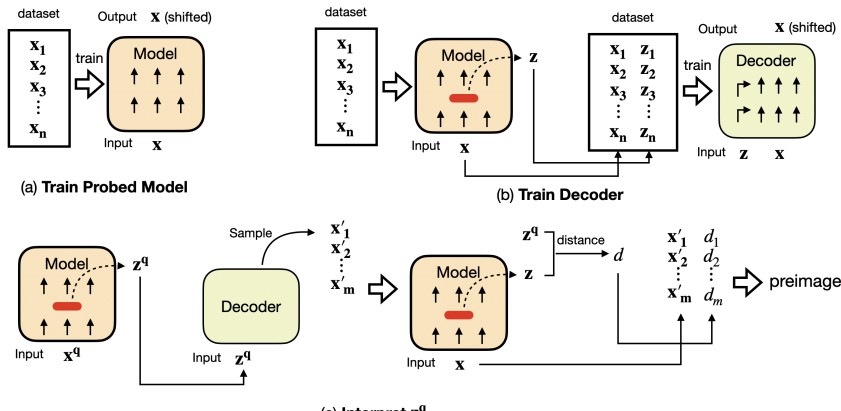

Figure 2: **(a)** The probed model is trained on language modeling objective. **(b)** Given a trained probed model, we first cache the internal activations $\mathbf{z}$ together with their corresponding inputs and activation site indices (omitted in the figure for brevity), then use them to train the decoder. The decoder is trained with language modeling objective, while being able to attend to $\mathbf{z}$. **(c)** When interpreting a specific query activation $\mathbf{z^q}$, we give it to the decoder, which generates possible inputs auto-regressively. We then evaluate the distances on the original probed model.

in all our three case studies, we vary $\epsilon$ and set it so we can read out coherent concepts from the $\epsilon$-preimage (Appendix A.4). With a threshold-based definition, we consider only those pieces of information that have substantial impact on the activation. See more discussion in Appendix A.1.

**Conditional Decoder Model** In this paper, we study the setting where $\mathbf{x}^q$ is a sequence. Directly enumerating $B_{\mathbf{z}^q,f,\epsilon}$ is in general not scalable, as the input space grows exponentially with the sequence length. To efficiently inspect $B_{\mathbf{z}^q,f,\epsilon}$, we train a conditional decoder model that takes as input the activation $\mathbf{z}^q$ and generates inputs giving rise to similar activations in the model under investigation. In the following, we refer to the original model that we are interpreting as the *probed model*, the conditional decoder as the *decoder*, the place in the probed model from which we take the activation as the *activation site* (e.g., the output of $i$th layer), the inputs generated by the decoder as *samples*, and the index of a token in the sequence as *position*.

We implement the decoder as an autoregressive language model conditioned on $\mathbf{z}^q$, decoding input samples $\mathbf{x}$ (see Figure 2, and details in Appendix C). As the decoder's training objective corresponds to recovering $\mathbf{x}$ exactly, sampling at temperature 1 will typically not cover the full $\epsilon$-preimage. Thus, for generating elements of the $\epsilon$-preimage, we increase diversity by drawing samples at higher temperatures and with noise added to $\mathbf{z}^q$ (details in Appendix A.2). We then evaluate $D(f(\mathbf{x}), \mathbf{z}^q)$ at each position in each sample $\mathbf{x}$, select the position minimizing $D$,[3] determine membership in $B_{\mathbf{z}^q,f,\epsilon}$, and subsample in-$\epsilon$-preimage and out-of-$\epsilon$-preimage samples for inspection.

An important question is whether this method, relying on a black-box decoder, produces valid $\epsilon$-preimages. *Correctness* (are all generated samples in the $\epsilon$-preimage?) is ensured by design, as we evaluate $D(f(\mathbf{x}), \mathbf{z}^q)$ for each generated sample. The other angle is *completeness* (are the samples representative of the $\epsilon$-preimage?). If some groups of inputs in $\epsilon$-preimage are systematically missing from the generations, one may overestimate the information contained in activations. But this behavior would be punished by the training objective, since the loss on these examples would be high. We explicitly verify completeness by enumerating inputs in one of our case studies (Appendix B). Another approach is to design counter-examples $\mathbf{x}$ not satisfying a hypothesis about the content of $B_{\mathbf{z}^q,f,\epsilon}$. In our experiments, we found that these examples were always outside of $B_{\mathbf{z}^q,f,\epsilon}$.

## 3 Discovering the Underlying Algorithm by InversionView

**Notation.** In the transformer architecture, outputs from each layer are added to their inputs due to residual connection. The representations of each token are only updated by additive updates,

---

[3]In some cases, including Figure 3b 4b, we fix a particular position for interpretation. See Appendix A.3

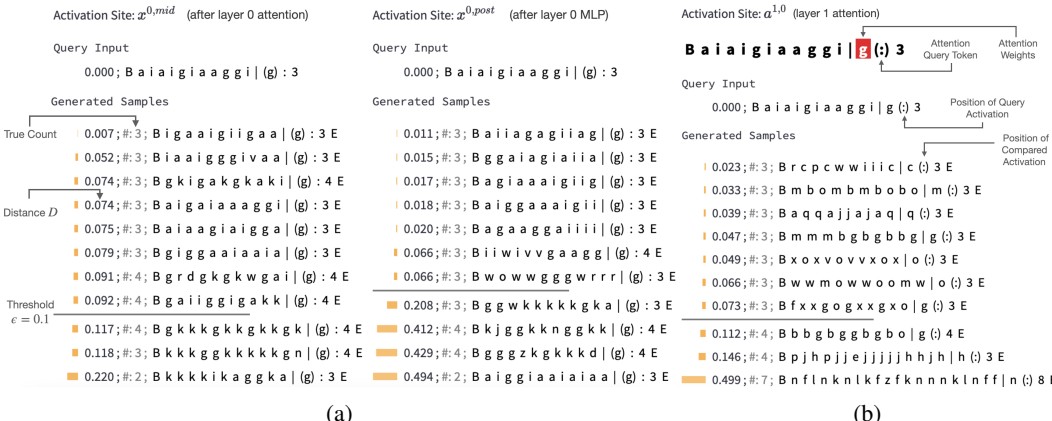

(a)                  (b)

Figure 3: **InversionView on Character Counting Task.** The model counts how often the target character (after '|') occurs in the prefix (before '|'). B and E denote beginning and end of sequence tokens. The query activation conditions the decoder to generate samples capturing its information content. We show non-cherrypicked samples inside and outside the $\epsilon$-preimage ($\epsilon = 0.1$) at three activation sites on the same query input. Distance for each sample is calculated between activations corresponding to the parenthesized characters in the query input and the sample. "True count" indicates the correct count of the target character in the samples (decoder may generate incorrect counts). **(a)** *MLP layer amplifies count information.* Comparing the distances before (left) and after (right) the MLP, we see that samples with diverging counts become much more distant from the query activation. **(b)** In the next layer (":" exclusively attends to target character – copying information from residual stream of target character to the residual stream of ":"), *the count is retained but the identity of the target character is no longer encoded* ("c", "m", etc. instead of "g"), as it is no longer relevant for the predicting the count. Therefore, observing the generations informs us of the activations' content and how it changes across activation sites.

forming a *residual stream* [17]. Using notation based on [17] and [40], we denote the residual stream as $x^{i,\{\text{pre},\text{mid},\text{post}\}} \in \mathbb{R}^{N \times d}$, where $i$ is the layer (an attention (sub)layer + an MLP (sub)layer) index, $N$ is the number of input tokens, $d$ is the model dimension, $\text{pre}$, $\text{mid}$, $\text{post}$ stand for the residual stream before the attention layer, between attention and MLP layer, and after the MLP layer. For example, $x^{0,\text{pre}}$ is the sum of token and position embedding, $x^{0,\text{mid}}$ is the sum of the output of the first attention layer and $x^{0,\text{pre}}$, and $x^{0,\text{post}}$ is the sum of the output of the first MLP layer and $x^{0,\text{mid}}$. Note that $x^{i,\text{post}} = x^{i+1,\text{pre}}$. We use subscript $t$ to refer to the activation at token position $t$, e.g., $x_t^{i,\text{mid}} \in \mathbb{R}^d$. The attention layer output decomposes into outputs of individual heads $h^{i,j}(\cdot)$, i.e., $x^{i,\text{mid}} = x^{i,\text{pre}} + \sum_j h^{i,j}(\text{LN}(x^{i,\text{pre}}))$, where $\text{LN}(\cdot)$ represents layer normalization (GPT style/pre-layer-norm). We denote the attention head's output as $a^{i,j}$, i.e., $a^{i,j} = h^{i,j}(\text{LN}(x^{i,\text{pre}}))$.

**Decoder Architecture.** We train a single two-layer transformer decoder across all activation sites of interest. The query activation $\mathbf{z}^q$ is concatenated with an *activation site embedding* $\mathbf{e}$, a learned embedding layer indicating where the activation comes from, passed through multiple MLP layers with residual connections, and then made available to the attention heads in each layer of the decoder, alongside the already present tokens from the input, so that each attention head can also attend to the post-processed query activation in addition to the context tokens. Each training example is a triple consisting of an activation vector $\mathbf{z}^q \in \mathbb{R}^d$, the activation site index, and the input, on which the decoder is trained with a language modeling objective. Appendix C has technical details.

### 3.1 Character Counting

We train a transformer (2 layers, 1 head) on inputs such as "vvzccvczvvvzvcvc|v:8" to predict the last token "8", the frequency of the target character (here, "v") before the separator "|". For each input, three distinct characters are sampled from the set of lowercase characters, and each character's frequency is sampled uniformly from 1–9. The input length varies between 7 and 31. We created 1.56M instances and applied a 75%-25% train-test split; test set accuracy is 99.53% (Details in

Appendix D). We use $D(\mathbf{z}, \mathbf{z}^q) = \frac{\|\mathbf{z}-\mathbf{z}^q\|_2}{\|\mathbf{z}^q\|_2}$ (i.e., normalized euclidean distance, as the magnitude of activations varies between layers), where $\mathbf{z}$ denotes the aforementioned $f(\mathbf{x})$, and set $\epsilon = 0.1$.

**Interpreting via InversionView and attention.** In layer 0, the target character consistently attends to the same character in the previous context, suggesting that counting happens here. In Figure 3a, we show the $\epsilon$-preimage of $x_{tc}^{0,\mathrm{mid}}$ and $x_{tc}^{0,\mathrm{post}}$, where the subscript $tc$ denotes the target character. We show $\approx 10$ random samples at a single query input, but our hypotheses are based on—and easily confirmed by—rapid visual inspection of dozens of inputs across different query inputs.[4] On the left (before the MLP), the activation encodes the target character, as all samples have "g" as the target character. Count information is not sharply encoded: while the closest activation corresponds to "g" occurring 3 times, two activations corresponding to a count-4 input ("g" occurring 4 times) are also close, even closer than a count-3 input. On the other hand, on the right (after the MLP), only count-3 inputs are inside the $\epsilon$-preimage, and count-4 inputs become much more distant than before. Comparing the $\epsilon$-preimage before and after the MLP in layer 0, we find that the MLP makes the count information more prominent in the representational geometry of the activation. The examples are not cherry-picked; count information is generally reinforced by the MLP across query inputs.

In the next layer, the colon consistently attends to the target character, and InversionView confirms that count information is moved to the colon's residual stream (Figure 3b). More importantly, this illustrates how information is abstracted: We previously found that $x_{tc}^{0,\mathrm{post}}$ encodes identity and frequency of the target character. However, the colon obtains only an abstracted version of the information, in which count information remains while the target character is largely (though not completely) removed. InversionView makes this process visible, by showing that the target character becomes interchangeable with little change to the activation. See more examples in Appendix D.2. Overall, with InversionView, we have found a simple algorithm by which the model makes the right prediction: *In layer 0, the target character attends to all its occurrences and obtains the counts. In layer 1, the colon moves the results from the target character to its residual stream and then produces the correct prediction.* Accounting for other activation sites, we find that the model implements a somewhat more nuanced algorithm, investigated in Appendix D.4. Overall, InversionView shows how certain information is amplified, but also how information is abstracted or forgotten.

**Quantitative verification.** We causally verified our hypothesis using activation patching [53, 21] on (position, head output) pairs. As the attention head in layer 1 attends almost entirely to the target character, only head outputs $a_{tc}^{0,0}$, $a_{:}^{0,0}$, and $a_{:}^{1,0}$ can possibly play a role in routing count information. We patch their outputs with activations from a contrast example flipping a single character before "|". We patch activations cumulatively, starting either at the lowest or highest layer, with some fixed ordering within each layer. For example, we patch $a_{:}^{0,0}$ and observe how final logits change compared to the clean run, then we patch both $a_{:}^{0,0}$ and $a_{tc}^{0,0}$ and do the same, and so forth. By the end of patching, the model prediction will be flipped. When adding an activation to the patched set, we attribute to it the increment in the difference of $LD$ before and after patching, where $LD$ denotes the logit difference between original count and the count in the contrast example. Cumulative patching allows us to observe dependencies: For instance, as we hypothesize that $a_{:}^{1,0}$ is *completely* dependent on $a_{tc}^{0,0}$, we expect that, when $a_{tc}^{0,0}$ is already patched, patching $a_{:}^{1,0}$ will have no further effect, whereas when $a_{tc}^{0,0}$ is not patched, patching $a_{:}^{1,0}$ will have a significant effect. Results (Figure 4a) match our prediction: Patching either of the activation in the hypothesized path ($a_{tc}^{0,0}$ and $a_{:}^{1,0}$) is sufficient to absorb the entire effect on logit differences, confirming the hypothesis. See Appendix D.3 for further details and D.4 for further experiments.

### 3.2 IOI circuit in GPT-2 small

To test the applicability of InversionView to transformers pretrained on real-world data, we apply our method to the activations in the indirect object identification (IOI) circuit in GPT-2 small [48] discovered by Wang et al. [54]. We apply InversionView to the components of the circuit, read out the information, and compare it with the information or function that Wang et al. [54] had ingeniously inferred using a variety of tailored methods, such as patching and investigating effects on logits and

---

[4]The reader can check generations conveniently at `https://inversion-view.streamlit.app`

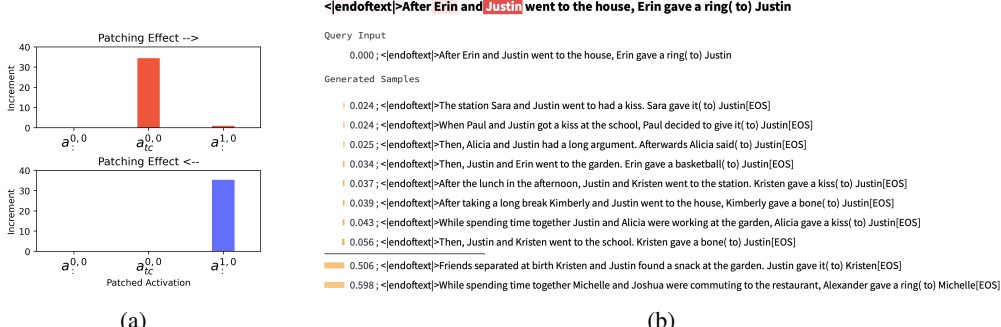

(a)                (b)

Figure 4: **(a) Character Counting.** Activation patching results show that $a_{tc}^{0,0}$ and $a_{:}^{1,0}$ play crucial roles in prediction, as hypothesized based on Figure 3 and Sec. 3.3. In contrast examples, only one character differs. Top: We patch activations cumulatively from left to right. We can see patching $a_{tc}^{0,0}$ accounts for the whole effect, and when $a_{tc}^{0,0}$ is already patched, patching $a_{:}^{1,0}$ has almost no effect. Bottom: On the other hand, if we patch cumulatively from right to left, $a_{:}^{1,0}$ accounts for the whole effect while patching $a_{tc}^{0,0}$ has no effect if $a_{:}^{1,0}$ has been patched. So we verified that $a_{:}^{1,0}$ solely relies on $a_{tc}^{0,0}$ and this path is the one by which the model performs precise counting. The patching effect is averaged across the whole test set. **(b) IOI.** InversionView applied to Name Mover Head 9.9 at "to"; we fix the compared position to "to". Throughout the $\epsilon$-preimage, "Justin" appears as the IO, revealing that the head encodes this name. This interpretation is confirmed across query inputs.

attention. We show that InversionView unveils the information contained in the attention heads' outputs, with results agreeing with those of Wang et al. [54].

The IOI task consists of examples such as "When Mary and John went to the store, John gave a drink to", which should be completed with "Mary". We use S for the subject "John" in the main clause, IO for the indirect object "Mary" introduced in the initial subclause, S1 and S2 for the first and second occurrences of the subject, and END for the "to" after which IO should be predicted. To facilitate comparison, we denote attention heads as in Wang et al. [54] with i.j denoting $h^{i,j}$. Wang et al. [54] discover a circuit of 26 attention heads in GPT-2 small and categorize them by their function. In short, GPT-2 small makes correct predictions by copying the name that occurs only once in the previous context. For InversionView, we train the decoder on the IOI examples (See details in E.1). Despite the size of the probed model, we find the same 2-layer decoder architecture as in Section 3.1 to be sufficient. We use $D(\mathbf{z}, \mathbf{z}^q) = 1 - \frac{\mathbf{z} \cdot \mathbf{z}^q}{||\mathbf{z}|| \cdot ||\mathbf{z}^q||}$ (i.e., cosine distance), and $\epsilon = 0.1$. Euclidean distance leads to similar results, but cosine distance is a better choice for this case (Appendix E.4).

We start with the Name Mover Head 9.9, which Wang et al. [54] found moves the IO name to the residual stream of END. 4b shows the $\epsilon$-preimage at "to". The samples in the $\epsilon$-preimage share the name "Justin" as the IO. The head also shows similar activity at some other positions (Appendix A.3). Results are consistent across query inputs. Therefore, InversionView agrees with the conclusions of Wang et al. [54] on head 9.9. Applying the same analysis to other heads (Table 2), we recovered information in high agreement with the information that Wang et al. [54] had inferred using multiple tailored methods. For example, Wang et al. [54] found S-Inhibition heads were outputting both token signals (value of S) and position signals (position of S1) by patching these heads' outputs from a series of counterfactual datasets. These datasets are designed to disentangle the two effects, in which token and/or position information are ablated or inverted. These two kinds of information can be directly read out by InversionView (Figure 19 shows an example for an S-Inhibition head that contains position information), and there is no need to guess the possible information to design patching experiments. Overall, among the 26 attention heads that Wang et al. [54] identified, InversionView indicates a different interpretation in only 3 cases; these (0.1, 0.10, 5.9) were challenging for the methods used before (Appendix E.3). In summary, InversionView scales to larger models.

### 3.3 3-Digit Addition

We next applied InversionView to the problem of adding 3-digit numbers, between 100 and 999. Input strings have the form "B362+405=767E" or "B824+692=1516E", and are tokenized at the

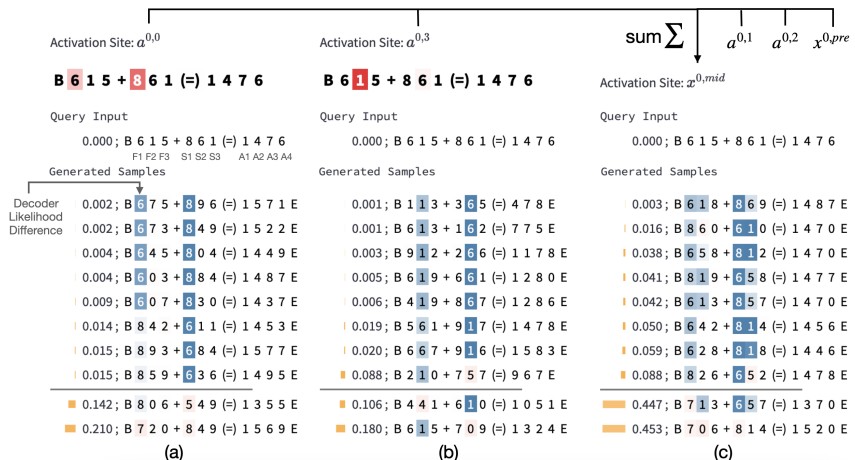

Figure 5: **InversionView applied to 3-digit addition**: Visually inspecting sample inputs inside and outside the $\epsilon$-preimage of the query allows us to understand what information is contained in an activation. The color on each token in generated samples denotes the difference in the token's likelihood between a conditional or unconditional decoder (Appendix G). The shade thus denotes how much the generation of the token is caused by the query activation (darker shade means a stronger dependence). In (a–c), the colored tokens are most relevant to the interpretation. We interpret two attention heads (a,b) and the output of the corresponding residual stream after attention (c). In **(a)**, what's common throughout the $\epsilon$-preimage is that the digits in the hundreds places are 6 and 8. Inputs outside the $\epsilon$-preimage don't have this property. In **(b)**, what's common is that the digits in tens places are 1, 6, or numerically close. Hence, we can infer that *the activation sites $a^{0,0}$ and $a^{0,3}$ encode hundreds and tens place in the input operands respectively;* the latter is needed to provide carry to A1. Also, the samples show that the activations encode commutativity since the digits at hundreds and tens place are swapped between the two operands. In **(c)**, the output of the attention layer after residual connection combining information from the sites in (a) and (b) *encodes "6" and "8" in hundreds place, and the carry from tens place*. Note that $a^{0,1}$ and $a^{0,2}$ contains similar information as $a^{0,0}$. These observations are confirmed across inputs. Taken together, InversionView reveals how information is aggregated and passed on by different model components.

character level. We use F1, F2, F3 to denote the three digits of the first operand and S1, S2, S3 for the digits of the second operand, and A1, A2, A3, A4 (if it exists) for the three or four digits of the answer, and C2, C3 for the carry from tens place and ones place (i.e., C2: whether F2+S2$\geq$10, C3: whether F3+S3$\geq$10). Unlike [47], we do not left-pad answers to have all the same length; hence, positional information is insufficient to determine the place value of each digit.

The probed model is a decoder-only transformer (2 layers, 4 attention heads, dimension 32). We set attention dropout to 0. Other aspects are identical to GPT-2. The model is trained for autoregressive next-token prediction on the full input, in analogy to real-world language models. In testing, the model receives the tokens up to and including "=", and greedily generates up to "E". The prediction counts as correct if all generated tokens match the ground truth. The same train-test ratio as in Section 3.1 is used. The test accuracy is 98.01%. For other training details see Appendix F.1.

**Interpreting via InversionView and attention.** As Section 3.1 we use normalized Euclidean distance for $D(\cdot, \cdot)$ and the threshold $\epsilon = 0.1$. We first trace how the model generates the first answer digit, A1, by understanding the activations at the preceding token, "=". We first examine the attention heads at "=" in the 0-th layer (Figure 5). As for the first head ($a^{0,0}$), only F1 and S1 matter in the samples – indeed, changing other digits, or swapping their order, has a negligible effect on the activation (Figure 5). Across different inputs, each of the three heads $a^{0,0}$, $a^{0,1}$, $a^{0,2}$ encode either one or both of F1 and S1 (Figure 26); taken together, they always encode both. This is in agreement with attention focusing on these tokens. The fourth and remaining head in layer 0 ($a^{0,3}$) encodes F2 and S2, which provide the carry from the tens place to the hundreds place. Combining the information from these four heads, $x^{0,\text{mid}}$ consistently encodes F1 and S1; and approximately represents F2, S2—only the carry to A1 (whether F2+S2$\geq$10) matters here (Figure 5c). Other examples are in

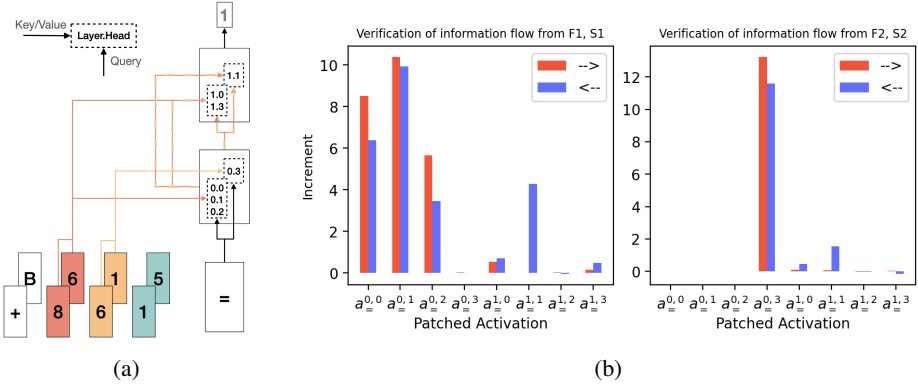

(a)                                                                        (b)

Figure 6: **3-Digit Addition Task**: **(a)** Information flow diagram for predicting A1 inferred via InversionView. The colors denote which places are routed; alternating colors indicate two places are routed. This is a subfigure of Figure 31. **(b)** Validation of (a) via activation patching for the prediction of A1. Like Figure 4a, $\rightarrow$ ($\leftarrow$) means cumulatively patching activation from left to right (right to left) on the horizontal axis. **Left:** Patching with activation containing modified F1 and S1 information. **Right:** Patching with activation containing modified F2 and S2 information. As we can see, components from (a) show a substantial increment if and only if they have a not-yet-patched connection to output (when patching right to left) or input (patching left to right), verifying that (a) causally describes the flow of information. Therefore, InversionView helps us uncover both information flow and content of activations.

Figure 27. We can summarize the function of layer 0 at "=": *Three heads route F1 and S1 to the residual stream of "=" $x_=$. The fourth head routes the carry resulting from F2 and S2.* Layer 1 mainly forwards information already obtained in layer 0, and does not consistently add further information for A1. See more examples in Appendix F.2.

Figure 6a shows the circuit predicting A1. InversionView allows us to diagnose an important deficiency of this circuit: Even though the ones place sometimes receives attention in layer 1, the circuit does not consistently provide the carry from the ones place to the hundreds place, which matters on certain inputs—we find that this deficiency in the circuit accounts for *all* mistakes made by the model (Appendix F.3). Taken together, we have provided a circuit allowing the model to predict A1 while also understanding its occasional failure in doing so correctly. Corresponding findings for A2, A3, and A4 are in Table 3 and Figure 31. From A2 onwards, InversionView allows us to uncover how the model exhibits two different algorithms depending on whether the resulting output will have 3 or 4 digits. In particular, when predicting A3, the layer 0 circuit is the same across both cases, while the layer 1 circuit varies, since this determines whether A3 will be a tens place or ones place. Beyond figures in the Appendix, we encourage readers to verify our claims in our interactive web application.

**Quantitative verification.** We used causal interventions to verify that information about the digits in hundreds and tens place is routed to the prediction of A1 only through the paths determined in Figure 6a, and none else. Like before, we cumulatively patch the head output on "=" preceding the target token A1, with an activation produced at the same activation site by a contrast example changing both digits in a certain place. Results shown in Figure 6b strongly support our previous conclusions. For example, $a^{0,3}$ and $a^{1,2}$ are not relevant to F1 and S1. Important heads detected by activation patching, $a^{0,0}, a^{0,1}, a^{0,2}, a^{1,1}$, all contain F1 and S1 according to Figure 6a. Furthermore, we can also confirm that $a^{1,1}$ relies on the output of layer 0 as depicted in sub-figure (a): When heads in layer 0 are already patched, patching $a^{1,1}$ has no further effect (value corresponding to $\rightarrow$ is zero), but it has an effect when patching in the opposite direction. On the contrary, $a^{1,0}$ shows little dependence on layer 0, consistent with Figure 6a. On the right of Figure 6b, we can confirm that $a^{0,3}$ is important for routing F2 and S2, and the downstream heads in layer 1 rely on it. Findings for other answer digits are similar (See Appendix F.5). Overall, the full algorithm obtained by InversionView is well-supported by causal interventions.

**InversionView reveals granularity of information.** Heads often read from both digits of a place, but only the sum matters for addition. Are the digits represented separately, or only as their sum? Unlike traditional probing, InversionView answers this question without designing tailored probing tasks. In Figure 7 (left), $a^{0,2}$ exactly represents F2 and S2 (here, 2 and 5). Other inputs where F1+S1=5+2 have high $D$. In contrast, on the right, F2 and S2 are represented only by their sum: throughout the $\epsilon$-preimage, F2+S2=9. In fact, we find such sum-only encoding only when F2+S2=9—a special case where the ones place of operands affects the hundreds place of the answer via cascading carry. We hypothesize that the model encodes them similarly because these inputs require special treatment. Therefore, even though encoding number pairs by their sum is a good strategy for the addition task from a human perspective, the model only does it as needed. We also observe intermediate cases (Figure 29).

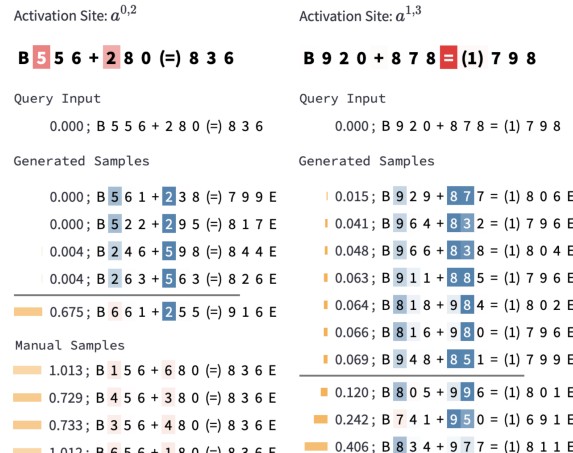

Figure 7: **3-Digit Addition Task**: InversionView uncovers different ways in which digit representation is encoded in activations. **Left:** The digits in the hundreds place are encoded separately and hence generations denote them as separate entities. **Right:** The digits in the tens place are encoded as a sum (9 in this case) and the generations represent different 2-partitions (7+2, 6+3, 1+8, 5+4, etc.) of that sum.

### 3.4 Factual Recall

To test whether InversionView can be applied to larger language models, we explore how GPT-2 XL (1.5B parameters) performs the task of recalling factual associations. In this case study, our intention is not to provide a full interpretation of the computations performed to solve this task, which we deem out of scope for this paper. Instead, we show that InversionView produces interpretable results on larger models by focusing on a relatively small set of important attention heads in upper layers. The decoder model in this case study is based on GPT-2 Medium, because we expect a more complex inverse mapping from activation to inputs to be learned. We observe the resulting $\epsilon$-preimage can express high-level knowledge (Figure 39-44), and sometimes can predict the failure of the model (Appendix H.6). Using InversionView, we again shed light on the underlying mechanism of the model. We present detailed findings in Appendix H.

## 4  Discussion and Related Work

**Comparison with other Interpretation Methods**    Supervised **probing classifiers**, assessing how much information about a variable of interest is encoded in an activation site, are arguably the most common method for uncovering information from activations [e.g. 2, 6, 5, 4, 55, 52, 33, 34]. It requires a hypothesis in advance and is thus inherently limited to hypotheses conceived a priori by the researcher. InversionView, on the other hand, helps researchers form hypotheses without any need for prior guesses, and allows fine-grained per-activation interpretation. Inspecting attention patterns [e.g. 12] is a traditional approach to inferring **information flow**, and we have drawn on it in our analyses. More recently, path patching [54, 26, 13, 28, 35] causally identifies *paths* along which information flows. While the information flow provides an *upper bound* on the information passed along by tracing back to the input token, it is insufficient for determining how information is processed and abstracted. For instance, in Section 3.1, occurrences of the target character are causally connected to $a_{tc}^{0,0}$, which then connects to $a_{:}^{1,0}$ (direct or mediated by MLP layer 0). Without looking at encoded information, we only know that the information in these paths is related to the occurrences of the target character, but not whether it is their identity, positions, count, etc. More generally, when a component reads a component that itself has read from multiple components, connectivity does not tell us which pieces of information are passed on. Similar considerations apply to other intervention methods.

Geva et al. [24] intervene on attention weights to study information flow. Activation patching, a causal intervention method, can be used to study the causal effect of an activation on the output, and can help localize where information is stored [37, 49], or find alignment between a high-level causal model and inner states of a neural model [21, 22, 57]. Many recent works obtain insights about information content by projecting representations or parameters into the **vocabulary space** [42, 7, 44, 31, 54, 23, 24, 16]. This technique is sometimes referred to as Direct Logit Attribution (DLA). We argue that DLA is only suitable for studying model components that *directly* affect model's final output. For those components whose effect is mediated by other components, their output information is meant to be read by a downstream component, thus not necessarily visible when projecting to the vocabulary space. We provide further discussion in Appendix I. Generalizing this approach, Ghandeharioun et al. [25] patch activations into an LLM. Another line of recent research [10, 51, 14] decomposes activations into interpretable features using **sparse autoencoders**.

Some other interpretation methods also generate in input space, but differ from InversionView in goals and methods. This includes feature visualization [43, 41], adversarial or counterfactual example generation [27, 59, 46, 45], and GAN inversion methods [58]. We discuss the similarities and differences of these works compared to InversionView in Appendix I.4.

InversionView offers distinctive advantages and makes analyses feasible that are otherwise very hard to do with other methods. It can also improve the interpretability workflow in coordination with other methods. For example, one may first use methods such as path patching or attribution [50, 18] to localize activity to specific components, and then understand the function of these components using InversionView. In sum, InversionView is worth adding to the toolbox of interpretability research.

**Transformer Circuits for Arithmetic**    Related to Section 3.3, [47] interpret the algorithm implemented by a 1-layer 3-head transformer for $n$-digit addition ($n \in \{5, 10, 15\}$), finding that the model implements the usual addition algorithm with restrictions on carry propagation. In their one-layer setup, attention patterns are sufficient for generating hypotheses. Lengths of operands and results are fixed by prepending 0. Our results, in contrast, elucidate a more complex algorithm computed by a *two*-layer transformer on a more realistic version without padding, which requires the model to determine which place it is predicting. We also contribute by providing a detailed interpretation, including how digits are represented in activations.

**Automated Interpretation for InversionView**    Recent work has started using LLMs to generate interpretations [8, 10]. The samples produced by InversionView can be easily fed into LLMs for automated interpretation. We show a proof of concept by using Claude 3 to interpret the model trained for 3-digit addition. See results in Table 5. The LLM-provided interpretation reflects the main information in almost all cases of the addition task. Despite some flaws, the outcome is informative in general, suggesting this as a promising direction for further speeding up hypothesis generation.

**Limitations**    InversionView relies on a black-box decoder, which needs to be trained using relevant inputs and whose completeness needs to be validated by counter-examples. Also, InversionView, while easing the human's task, is still not automated, and interpretation can be laborious when there are many activation sites. We focus on models up to 1.5B parameters; scaling the technique to large models is an interesting problem for future work, which will likely require advances in localizing behavior to a tractable number of components of interest. Fourth, interpretation uses a metric $D(\cdot, \cdot)$. The geometry, however, in general could be nonisotropic and treating each dimension equally could be sub-optimal. We leave the exploration of this to future work.

## 5    Conclusion

We present InversionView, an effective method for decoding information from neural activations. In four case studies—character counting, IOI, 3-digit addition, and factual recall—we showcase how it can reveal various types of information, thus facilitating reverse-engineering of algorithm implemented by neural networks. Moreover, we compare it with other interpretability methods and show its unique advantages. We also show that the results given by InversionView can in principle be interpreted automatically by LLMs, which opens up possibilities for a more automated workflow. This paper only explores a fraction of the opportunities this method offers. Future work could apply it to subspaces of residual stream, to larger models, or to different modalities such as vision.

## Acknowledgements

Funded by the Deutsche Forschungsgemeinschaft (DFG, German Research Foundation) – Project-ID 232722074 – SFB 1102. We thank anonymous reviewers for their encouraging and constructive feedback.

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

# Contents

## A  Practical Guidelines

### A.1  Observing Larger Neighborhoods is Important

Here, we illustrate the importance of inspecting $\epsilon$-preimages up to the threshold $\epsilon$, rather than just top-$k$ nearest neighbors of the query activation. In Figure 8, an initial glance at the samples on the left may suggest that the residual stream of "+" encodes F1 and F2. However, observing a broader neighborhood (as depicted on the right) reveals that this conclusion is not even robust to tiny perturbations of the activation. Indeed, after a more comprehensive calculation over all possible $x_+^{0,\text{post}}$, we find that the maximum possible metric value between any pair of $x_+^{0,\text{post}}$ is 0.0184. So for any $\epsilon \geq 0.0184$ the $\epsilon$-preimage covers the entire input space. Hence, the activation is unlikely to contain usable information.

We further prove this by causal intervention. We found that $x_+^{0,\text{post}}$ has no effect on the model's output. Concretely we patch $x_+^{0,\text{post}}$ with its mean on the test set (mean ablation [54]) and for each prediction target (A1, A2 etc.), we compare 1) the KL divergence between the distribution before and after patching. 2) logit decrement rate, which is the difference between the maximum logit value before patching and the logit value of the same target token after patching, divided by the former. E.g., 1.0 means the logit is reduced to zero (assuming it is originally positive). The results are shown in Table 1. We can see the effect of $x_+^{0,\text{post}}$ is negligible.

|                      | A1                    | A2                   | A3                   | A4/E                 |
|----------------------|-----------------------|----------------------|----------------------|----------------------|
| KL divergence        | $8.1 \times 10^{-8}$  | $5.4 \times 10^{-7}$ | $2.5 \times 10^{-8}$ | $3.6 \times 10^{-9}$ |
| Logit decrement rate | $-2.3 \times 10^{-5}$ | $8.7 \times 10^{-7}$ | $1.4 \times 10^{-5}$ | $2.8 \times 10^{-7}$ |

Table 1: Activation patching results for $x_+^{0,\text{post}}$.

### A.2  Sampling with Decoder Model

In Section 2, we mentioned that the distribution $p(\mathbf{x}|\mathbf{z}^q)$ is modeled by the decoder. Strictly speaking, $p(\mathbf{x}|\mathbf{z}^q)$ represents the data distribution in the $\epsilon$-preimage defined by $\epsilon = 0$. For example, when the

Activation Site: $x^{0,post}$

Query Input

0.000; B 9 8 2 (+) 3 4 7 = 1 3 2 9

Generated Samples

| | |
|---|---|
| 0.000; B 9 8 6 (+)⋯ | 0.005; B 0 1 3 (+)⋯ |
| 0.000; B 9 8 6 (+)⋯ | 0.005; B 1 8 0 (+)⋯ |
| 0.000; B 9 8 5 (+)⋯ | 0.005; B 8 8 8 (+)⋯ |
| 0.000; B 9 8 5 (+)⋯ | 0.006; B 1 0 9 (+)⋯ |
| 0.000; B 9 8 9 (+)⋯ | 0.006; B 2 0 2 (+)⋯ |
| 0.000; B 9 8 6 (+)⋯ | 0.006; B 4 8 8 (+)⋯ |
| 0.000; B 9 8 7 (+)⋯ | 0.007; B 4 0 2 (+)⋯ |
| 0.000; B 9 8 2 (+)⋯ | 0.007; B 1 4 0 (+)⋯ |
| 0.003; B 8 9 0 (+)⋯ | 0.007; B 1 2 5 (+)⋯ |
| 0.003; B 8 9 1 (+)⋯ | 0.008; B 7 9 1 (+)⋯ |

Figure 8: Addition Task: Inspecting $\epsilon$-preimage avoids pitfall of inspecting simple top-k similar activations. Generation based on query activation $x_+^{0,\text{post}}$ of a random example. Contents after "+" is omitted since they do not affect the activation due to causal masking.

probed model is using causal masking, and a certain activation is relevant to all previous context (by non-zero attention weights), then $p(\mathbf{x}|\mathbf{z}^q)$ is the distribution over those inputs that share the same previous context (i.e., they have same prefix). This requires that the decoder can distinguish any tiny difference in activation and decode the full information (imagine a token attended with 0.0001 attention weight). Such a decoder must be very powerful and perhaps trained without any regularization. But in practice, the decoder is a continuous function of activation and tiny changes in activation are not perceivable by the decoder. We observe that the decoder rarely generates the sample that lies at the same point (producing the same activation) as the query input in vector space, instead it usually generates samples that are in the neighborhood of the query input. Because we need to observe the whole neighborhood of the query input and prevent samples from being too concentrated, we adjust the sampling temperature to control how concentrated they are. Importantly, even if the decoder is too powerful and can always recover the same activation, we can still obtain the neighborhood by adding random noise to the query activation before giving it to the decoder. This motivates decoding with temperature and noise, as described in the next paragraph.

**Increasing Coverage by Temperature and Noise.** In our experiments, we use both ways to control the generation, i.e., by adjusting the temperature and adding random noise to the query activation. We denote temperature as $\tau$ and noise coefficient as $\eta$. The noise vector consists of independent random variables sampled from the standard normal distribution and then multiplied by $\text{std}(\mathbf{z}^q) \cdot \eta$ where $\text{std}(\cdot)$ stands for standard deviation. In our web application, we provide multiple sampling configurations: four configurations in which $\tau = \{0.5, 1.0, 2.0, 4.0\}$ and $\eta = 0.0$ (only for addition task); one figuration named "Auto" which is sampled by following procedure: we iterate over a few predefined $\tau$ (ranging from 0.5 to 2.0) and $\eta$ (0.0 or 0.1) and sample a certain amount of inputs (e.g., 250) for each parameter combination. We then calculate the metric value for all inputs collected from different sampling configurations. We then randomly choose a small part of them (100) with different probability for in-$\epsilon$-preimage inputs and out-of-$\epsilon$-preimage inputs. We dynamically adjust the probability such that the in-$\epsilon$-preimage inputs account for 60%-80% of the chosen set of inputs (when this is possible). Note that we use different noise in factual recall task, which will be described later in Appendix H.4.

When inspecting the samples, we choose a configuration for which the distances $D(\cdot, \cdot)$ to the query activation best cover the interval $[0, \epsilon]$. The choice is usually specific to the activation site that we are inspecting and can be performed manually in the web application.

**<|endoftext|>After Erin and Justin went to the house, Erin gave a ring( to) Justin**

```
Query Input
```
0.000 ; <|endoftext|>After Erin and Justin went to the house, Erin gave a ring( to) Justin

```
Generated Samples
```
0.024 ; <|endoftext|>The station Sara and Justin went to had a kiss. Sara gave it( to) Justin[EOS]

0.024 ; <|endoftext|>When Paul and Justin got a kiss at the school, Paul decided to give it( to) Justin[EOS]

0.025 ; <|endoftext|>Then, Alicia and Justin had a long argument. Afterwards Alicia said( to) Justin[EOS]

0.030 ; <|endoftext|>Then, Justin and Erin went to the garden. Erin( gave) a basketball to Justin[EOS]

0.037 ; <|endoftext|>After the lunch in the afternoon, Justin and Kristen went to the station. Kristen gave a kiss( to) Justin[EOS]

0.039 ; <|endoftext|>After taking a long break Kimberly and Justin went to the house, Kimberly gave a bone( to) Justin[EOS]

0.042 ; <|endoftext|>While spending time together Justin and Alicia were working at the garden, Alicia( gave) a kiss to Justin[EOS]

0.048 ; <|endoftext|>Then, Justin and Kristen went to the school. Kristen( gave) a bone to Justin[EOS]

0.198 ; <|endoftext|>Friends separated at birth Kristen and Justin found a snack at the garden(.) Justin gave it to Kristen[EOS]

0.579 ; <|endoftext|>While spending time together Michelle and Joshua were commuting to the restaurant(,) Alexander gave a ring to Michelle[EOS]

Figure 9: IOI: InversionView applied to Name Mover Head 9.9 at "to"; Unlike Figure 4b, here the position minimizing $D(\cdot, \cdot)$ is in parentheses. The head also copies the name "Justin" in other circumstances, e.g., at "gave". The name "Justin" is always contained

## A.3 Selecting Position in Samples

As the decoder outputs an input but not the position of the activation, we then assign the position minimizing $D(\cdot, \cdot)$ to the query activation. Usually, there is only one position with a small $D(\cdot, \cdot)$, matching the structural position (not necessarily the absolute position) of the position the query activation was taken from (e.g., the target character in Figure 3a). In certain cases, we visualize $D(\cdot, \cdot)$ for an activation from a position not minimizing $D(\cdot, \cdot)$ for expository purposes. For example, in Figure 3b, because the target character exclusively attends to itself in layer 1, resulting $a_{tc}^{1,0} \approx a_{:}^{1,0}$, so sometimes the metric value of $a_{tc}^{1,0}$ is smaller than $a_{:}^{1,0}$. Throughout the appendix and our web application, we use italic font and rounded bars to visualize $D(\cdot, \cdot)$ in such cases.

We also find that selecting the position minimizing $D(\cdot, \cdot)$ can reveal that components are active in similar ways at other positions than the one originally investigated. For example, in Figure 9, we can see sometimes "gave" is selected. This is reasonable, because the IO is also likely to appear right after "gave" and the head needs to move the IO name for this prediction. We can see that activation at the period "." can also be somewhat similar to the query activation, this is not surprising. Because the model needs to predict the subject for the next sentence and copying a name from the previous context is helpful. In summary, the copying mechanism can be triggered in circumstances different from IOI, selecting position minimizing $D(\cdot, \cdot)$ reveals more information about this.

## A.4 Threshold-Dependence of Claims about Activations

One question people may have is whether our conclusion about the information in activation depends significantly on the threshold we choose. To address this potential concern, we show more details about the geometry of the vector space in Figure 10. On the one hand, we can see that with different thresholds $\epsilon$ we can make different conclusions about the query activation. On the other hand, the conclusions made with different thresholds are "in alignment". In other words, the conclusions do not differ fundamentally, instead, the difference between them is about granularity or the amount of details being ignored.

Specifically, in Figure 10a, $\epsilon_1$ results in the conclusion that the count is 5, the target character is either 't' or 'm', and also approximate sequence length is retained. $\epsilon_2$ results in a conclusion only about the count and the sequence length. In Figure 10b, if we set the threshold to $\epsilon_1$ (i.e., a value between 0.000 and 0.009), the obtained information will be F1=5, S1=7. If we set the threshold to $\epsilon_2$, the information will be 5 and 7 are in the hundreds place. If we set the threshold to $\epsilon_3$ the information will be "5 is in the hundreds place". In Figure 10c, $\epsilon_1$ results in conclusion that 9, 8 are in hundreds place and 2, 7 are in tens place; $\epsilon_2$ results in conclusion that 9, 8 are in hundreds place and F2+S2=9; $\epsilon_3$ results in conclusion that F1+S1≈17 and F2+S2≈9. Therefore, changing the threshold value will not lead us in a different direction, because the $\epsilon$-preimage is based on the same underlying geometry.

**(a)**

B m u a u m m m m | [m] (:) 5

Query Input
0.000; B m u a u m m m m | m (:) 5

Generated Samples

*0.012*;#:5; B t m t q m m m m | m (:) 5 E
*0.016*;#:5; B o o p m m m m m | m (:) 5 E
*0.015*;#:5; B p t t p t p t p p p s p | t (:) 5 E
*0.016*;#:5; B p t p t p p a p t p t p | t (:) 5 E
*0.017*;#:5; B m a m a m m o m | m (:) 5 E
*0.020*;#:5; B q t t q j q t q t t q q q | t (:) 5 E
*0.021*;#:5; B h c o t t t t o o o o t o o | t (:) 5 E
*0.021*;#:5; B m m m m m o a a | m (:) 5 E
*0.023*;#:5; B m t t q m m q t m t q t q | t (:) 5 E
*0.024*;#:5; B m m m m m a h h | m (:) 5 E
*0.024*;#:5; B y t v y t y t t t y y y y g | t (:) 5 E
*0.025*;#:5; B t t t t h z h h h t h z z z h | t (:) 5 E
*0.024*;#:5; B o h t h t h o t o o o o t t | t (:) 5 E
*0.027*;#:5; B d t q q t q t t q q q t q | t (:) 5 E
*0.033*;#:5; B t t t o t t p o o o o o o | t (:) 5 E
*0.035*;#:5; B m o y o m o m o m o m | m (:) 5 E
ε₁
*0.037*;#:5; B h t t z h z h h z h z z z | h (:) 5 E
*0.037*;#:5; B m m m m m o o o o | m (:) 5 E
*0.039*;#:5; B m d x m m h d d m d m | m (:) 5 E
*0.042*;#:5; B t f t o t f f t t f f f o o | t (:) 5 E
*0.041*;#:5; B z g z g z g g z z g z | g (:) 5 E
*0.041*;#:5; B u m m m m p m | m (:) 5 E
*0.043*;#:5; B h h n m h h d h | h (:) 5 E
*0.044*;#:5; B m m m m m o t | m (:) 5 E
*0.045*;#:5; B m m c m c c m c c m c | m (:) 5 E
*0.049*;#:5; B o h h o m o o o h h h | h (:) 5 E
*0.052*;#:5; B c c a d c d c c d d d | c (:) 5 E
*0.062*;#:5; B r z z z z z c | z (:) 5 E
*0.068*;#:5; B m m a m m m a a a m a | a (:) 5 E
*0.075*;#:5; B m m m m m m m a a a a a | a (:) 5 E
ε₂
*0.124*;#:6; B a g o g o g g g g | g (:) 6 E
*0.129*;#:6; B o m o m m m o m o m | m (:) 5 E
*0.136*;#:6; B m m m m m m t h | m (:) 5 E
*0.142*;#:6; B t t t t t t q q q q q q g | t (:) 5 E
*0.158*;#:6; B t t r t t q t r r q t q q q q | t (:) 6 E
*0.148*;#:6; B m m g m m m g m g | m (:) 5 E
*0.180*;#:3; B i i g e i b b b e g g g g | i (:) 4 E
*0.307*;#:3; B a r g a a a g g a r r a | r (:) 5 E
*0.316*;#:3; B a k m m u m a | m (:) 5 E
*0.618*;#:0; B m z n n n t t t n z n b n z n z t t t n z | p (:) 4 E
*0.341*;#:3; B s v g n v s s n n g g s n | g (:) 4 E
*0.431*;#:1; B p g p g p p p b i p q | i (:) 1 E
*0.375*;#:2; B t y t y | t (:) 4 E
*0.441*;#:0; B x b b p x h b b h h b h h x h h b | w (:) 2 E

(a)

**(b)**

B [5] 5 0 + [7] 3 7 (=) 1 2 8 7

Query Input
0.000; B 5 5 0 + 7 3 7 (=) 1 2 8 7

Generated Samples

0.000; B 5 3 9 + 7 0 9 (=) 1 2 4 8 E
0.000; B 5 6 0 + 7 9 4 (=) 1 3 5 4 E
0.000; B 5 4 8 + 7 8 8 (=) 1 3 3 6 E
0.000; B 5 2 1 + 7 0 2 (=) 1 2 2 3 E
0.000; B 5 1 9 + 7 8 4 (=) 1 3 0 3 E
0.000; B 5 5 6 + 7 2 9 (=) 1 2 8 5 E
0.000; B 5 4 1 + 7 1 9 (=) 1 2 6 0 E
0.000; B 5 4 6 + 7 8 0 (=) 1 3 2 6 E
0.000; B 5 1 7 + 7 9 2 (=) 1 3 0 9 E
0.000; B 5 2 3 + 7 7 2 (=) 1 2 9 5 E
0.000; B 5 5 0 + 7 4 1 (=) 1 2 9 1 E
0.000; B 5 3 0 + 7 8 6 (=) 1 3 1 6 E
0.000; B 5 6 5 + 7 9 9 (=) 1 3 6 4 E
0.000; B 5 5 4 + 7 9 8 (=) 1 3 5 2 E
ε₁
0.000; B 5 0 7 + 7 9 2 (=) 1 2 9 9 E
0.009; B 7 1 0 + 5 7 1 (=) 1 2 8 1 E
0.009; B 7 7 4 + 5 0 7 (=) 1 2 8 1 E
0.009; B 7 5 7 + 5 1 0 (=) 1 2 6 7 E
0.009; B 7 2 9 + 5 5 2 (=) 1 2 8 1 E
0.009; B 7 0 3 + 5 7 7 (=) 1 2 8 0 E
0.009; B 7 9 4 + 5 9 8 (=) 1 3 9 2 E
0.009; B 7 5 0 + 5 0 1 (=) 1 2 5 1 E
0.009; B 7 1 2 + 5 2 6 (=) 1 2 3 8 E
0.009; B 7 1 0 + 5 2 0 (=) 1 2 3 0 E
0.009; B 7 5 4 + 5 3 0 (=) 1 2 8 4 E
0.009; B 7 6 0 + 5 2 2 (=) 1 2 8 2 E
ε₂
0.073; B 5 7 0 + 6 9 1 (=) 1 2 6 1 E
0.073; B 6 6 0 + 5 1 2 (=) 1 1 7 2 E
0.073; B 6 0 3 + 5 2 9 (=) 1 1 3 2 E
0.073; B 5 5 8 + 6 9 9 (=) 1 2 5 7 E
0.102; B 8 0 5 + 5 0 7 (=) 1 3 1 2 E
0.102; B 8 6 0 + 5 4 2 (=) 1 4 0 2 E
0.106; B 5 1 6 + 8 6 5 (=) 1 3 8 1 E
0.106; B 5 0 7 + 8 8 0 (=) 1 3 8 7 E
0.106; B 5 0 9 + 8 8 7 (=) 1 3 9 6 E
0.106; B 5 6 2 + 8 9 8 (=) 1 4 6 0 E
0.106; B 5 1 3 + 8 9 7 (=) 1 4 1 0 E
0.126; B 1 7 9 + 5 0 1 (=) 6 8 0 E
0.129; B 9 5 7 + 5 1 1 (=) 1 4 6 8 E
0.130; B 5 5 1 + 1 9 9 (=) 7 5 0 E
ε₃
0.146; B 5 8 6 + 5 8 4 (=) 1 1 7 0 E
0.383; B 6 (5) 5 + 8 1 7 (=) 1 4 7 2 E
0.436; B 6 1 9 + 4 5 7 (=) 1 0 7 6 E
0.477; B 6 9 1 + 4 0 8 (=) 1 0 9 9 E
0.509; B 6 7 6 + 1 8 5 (=) 8 6 1 E

(b)

**(c)**

B 9 2 0 + 8 7 8 [=] (1) 7 9 8

Query Input
0.000; B 9 2 0 + 8 7 8 = (1) 7 9 8

Generated Samples

0.003; B 9 2 8 + 8 7 0 = (1) 7 9 8 E
0.004; B 9 2 0 + 8 7 3 = (1) 7 9 3 E
0.005; B 8 2 0 + 9 7 4 = (1) 7 9 4 E
0.012; B 9 2 1 + 8 7 4 = (1) 7 9 5 E
0.015; B 8 7 4 + 9 2 7 = (1) 8 0 1 E
0.015; B 8 7 4 + 9 2 7 = (1) 8 0 1 E
0.015; B 9 2 3 + 8 7 9 = (1) 8 0 2 E
0.016; B 9 2 4 + 8 7 6 = (1) 8 0 0 E
ε₁
0.030; B 9 6 6 + 8 3 1 = (1) 7 9 7 E
0.040; B 9 6 2 + 8 3 7 = (1) 7 9 9 E
0.041; B 9 6 4 + 8 3 2 = (1) 7 9 6 E
0.047; B 9 6 4 + 8 3 7 = (1) 8 0 1 E
0.051; B 9 3 5 + 8 6 4 = (1) 7 9 9 E
0.058; B 9 4 2 + 8 5 1 = (1) 7 9 3 E
0.059; B 8 3 9 + 9 6 4 = (1) 8 0 3 E
0.060; B 9 8 2 + 8 1 8 = (1) 8 0 0 E
0.062; B 9 1 1 + 8 8 4 = (1) 7 9 5 E
0.063; B 9 1 1 + 8 8 5 = (1) 7 9 6 E
0.063; B 8 8 7 + 9 1 8 = (1) 8 0 5 E
0.064; B 9 5 6 + 8 4 1 = (1) 7 9 7 E
0.065; B 9 4 6 + 8 5 1 = (1) 7 9 7 E
0.066; B 8 1 6 + 9 8 0 = (1) 7 9 6 E
0.067; B 9 1 4 + 8 8 3 = (1) 7 9 7 E
0.067; B 9 1 5 + 8 8 7 = (1) 8 0 2 E
0.069; B 9 1 4 + 8 8 4 = (1) 7 9 8 E
0.082; B 9 5 7 + 8 4 2 = (1) 7 9 9 E
0.099; B 8 4 4 + 9 5 8 = (1) 8 0 2 E
0.120; B 8 0 5 + 9 9 6 = (1) 8 0 1 E
0.124; B 9 9 5 + 8 0 9 = (1) 8 0 4 E
0.124; B 9 9 5 + 8 0 9 = (1) 8 0 4 E
ε₂
0.211; B 9 0 1 + 7 9 4 = (1) 6 9 5 E
0.226; B 8 5 8 + 8 4 9 = (1) 7 0 7 E
0.242; B 7 4 1 + 9 5 0 = (1) 6 9 1 E
0.249; B 9 2 7 + 7 7 7 = (1) 7 0 4 E
0.251; B 8 5 4 + 8 5 7 = (1) 7 1 1 E
0.255; B 8 6 6 + 8 4 2 = (1) 7 0 8 E
0.260; B 8 1 8 + 8 8 4 = (1) 7 0 2 E
0.287; B 8 5 8 + 8 4 2 = (1) 7 0 0 E
0.289; B 8 6 0 + 8 3 4 = (1) 6 9 4 E
0.289; B 8 6 0 + 8 3 7 = (1) 6 9 7 E
0.294; B 8 4 7 + 8 5 0 = (1) 6 9 7 E
0.296; B 8 4 4 + 8 5 1 = (1) 6 9 5 E
0.328; B 8 5 2 + 7 5 4 = (1) 6 0 6 E
0.406; B 8 3 4 + 9 7 7 = (1) 8 1 1 E
0.417; B 9 2 6 + 8 8 9 = (1) 8 1 5 E
ε₃

(c)

Figure 10: **(a)** Activation site $a^{1,0}$. **(b)** Activation site $a^{0,2}$. **(c)** Activation site $a^{1,3}$. In all three cases, we use normalized Euclidean distance as the distance metric. We use $\epsilon_1, \epsilon_2, \cdots$ to mark varying threshold values by which different interpretations will be made.

In practice, rather than selecting a threshold first and treating inputs in a black-and-white manner, we first observe the geometry of the vector space and obtain a broad understanding of the encoded information, then choose a reasonable threshold that best *summarizes* our findings. In other words, the threshold value is used to simplify our findings so that we can focus more on the big picture of the model's overall algorithm, and it should also be set according to the difference that is likely to be readable for the model. As the interpretation progresses, one can see if the chosen threshold leads to a plausible algorithm and can adjust it if necessary. Finally, verification experiments are conducted to verify the hypothesis.

## B    Experimental Verification of Completeness

In Section 2, we described that an ideal strategy for obtaining samples in the $\epsilon$-preimage satisfies two desiderata: it only provides samples that are indeed within the $\epsilon$-preimage (Correctness), and it provides all such samples at reasonable probability (Completeness). As further described there, we can directly ensure Correctness by evaluating $D(\cdot)$ for every sample. Ensuring completeness is more challenging, due to the exponential size of the input space; the most general approach is to design counterexamples not satisfying a hypothesis about the content of the $\epsilon$-preimage, and verifying that $D(\cdot)$ is indeed large.

Here, we provide a direct test of completeness in one domain (3-digit addition). The primary concern with completeness is that, if some groups of inputs in $B_{\mathbf{z}^q, f, \epsilon}$ are systematically missing from the generated samples, one may overestimate the information contained in activations. To see if may happen in reality, we plotted log-probability against distance, each of which includes *all* inputs in 3 digit addition task, as shown in Figure 11. We next evaluated the sampling probability for different sampling configurations, as described in Appendix A.2. When adding noise, we calculate probability of an input using a Monte Carlo estimate: Concretely, because the probability of inputs is conditioned on the noise vector added to the query activation, we randomly sample 500 noise vectors from the normal distribution (with the standard deviation described in Appendix A.2) and calculate input probability given these noise vectors, then average to obtain the estimated probability, and then compute the logarithm.

Across setups, we can see that there is a triangular blank area in the bottom left corner, i.e., the bottom left frontier stretches from the upper left towards the lower right. In all sub-figures, not a single input close to the query input is assigned disproportionately low probability. All inputs in the $\epsilon$-preimage (the dots on the left of the red vertical line) are reasonably likely to be sampled from the decoder, with probability decreasing as the input becomes more distant, alleviating concerns about completeness for these query activations. On the other hand, some inputs distant from the query input are also likely to appear in samples, but this is not a problem for our approach, as we can easily tell that they are not in the $\epsilon$-preimage by calculating the distance (correctness is ensured).

We can see that sometimes the distribution of the decoder itself (at temperature 1 and no noise) is quite sharp, and in-$\epsilon$-preimage inputs can have low probability as they are near the boundary of preimage. By comparing the sub-figures, we can see both increasing the temperature and adding noise substantially smooth the distribution within the $\epsilon$-preimage, lowering the difference of the probability of inputs that are at similar distance.

## C    Decoder Model

The overall training and sampling pipelines are shown in Figure 2. In this section, we describe the architecture of the decoder model in detail.

The decoder model is basically a decoder-only transformer combined with some additional MLP layers. In order to condition the decoder on the query activation, the query activation is first passed through a stack of MLP layers to decode information depending on the activation sites and then made available to each attention layer of the transformer part of the decoder, as depicted in Figure 12.

**Processing Query Activation.**    The query activation $\mathbf{z}^q \in \mathbb{R}^d$ is first concatenated with a trainable activation site embedding $\mathbf{e}_{act} \in \mathbb{R}^{d_{site}}$, producing the intermediate representation $\mathbf{z}^{(0)} = [\mathbf{z}^q; \mathbf{e}_{act}]$. We chose $d_{site}$ to be the number of possible activation sites in the training set. The result $\mathbf{z}^{(0)}$ is

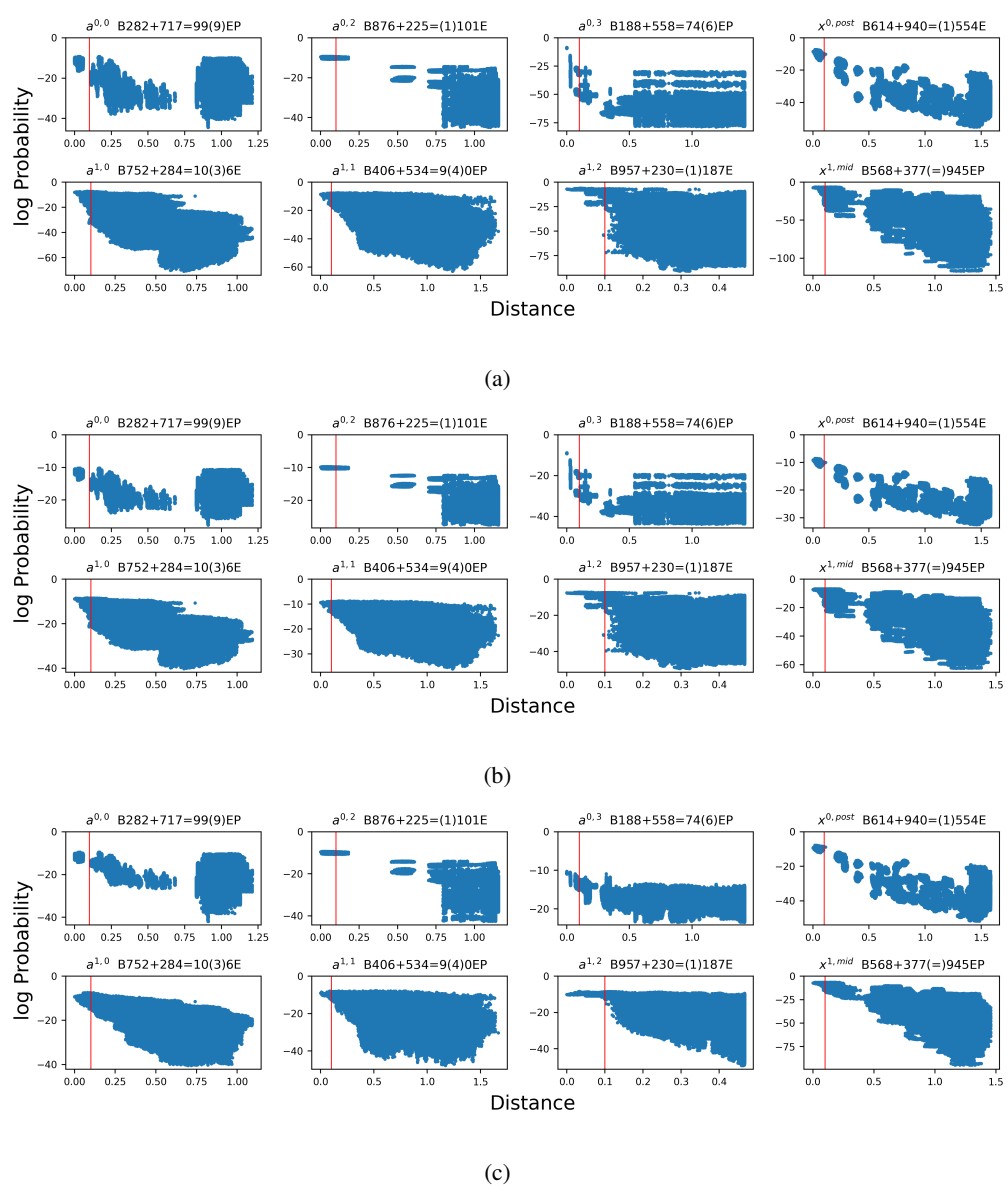

Figure 11: Addition Task: Exhaustive verification of the decoder's completeness for 8 random query activation. Failure of completeness would mean that some inputs result in an activation very close to the query activation but nonetheless are assigned very small probability. Here, we show that this does not happen, by verifying that *all* inputs within the $\epsilon$-preimage are assigned higher probability by the decoder than most other inputs. We also show that by increasing the temperature and adding random noise, we can increase the probability of inputs near the boundary of $\epsilon$-preimage. Each sub-figure – (a), (b), (c) – contains 8 scatter plots, each of which contains 810000 dots representing all input sequences in the 3-digit addition task. The y-axis of scatter plots is the log-probability of the input sequence given by the decoder (which reads the query activation), the x-axis is the distance between the query input and the input sequence. As before, distance is measured by the normalized Euclidean distance between the query activation (the activation site, query input, and selected position are shown in the scatter plot title) and the most similar activation along the sequence axis. In addition, the red vertical line represents the threshold $\epsilon$, which is 0.1 in the case study. **(a)** Temperature $\tau = 1.0$, no noise is added. **(b)** Temperature $\tau = 2.0$, no noise is added. **(c)** Temperature $\tau = 1.0$, noise coefficient $\eta = 0.1$ (See Appendix A.2 for explanation of $\eta$).

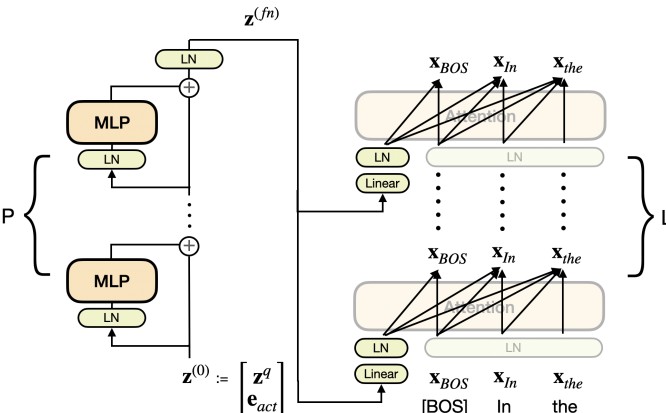

Figure 12: The decoder model architecture used in this paper. The query activation is processed by a stack of MLP layers before being available as part of the context in attention layers. We use transparent blocks to represent model components inherited from original decoder-only transformer model.

then fed through multiple MLP layers (each layer indexed by $p \in \{0, 1, \cdots, P - 1\}$) with residual connections:

$$\mathbf{z}^{(p+1)} = \text{MLP}(\text{LN}(\mathbf{z}^{(p)})) + \mathbf{z}^{(p)} \tag{2}$$

where LN represents layer normalization. For each MLP layer, the input, hidden and output dimensions are $d + d_{site}$, $d$, and $d + d_{site}$, respectively. The activation function is ReLU. There is also a final layer normalization, $\mathbf{z}^{(fn)} = \text{LN}(\mathbf{z}^{(P)})$.

**Integrating Query Activation.** As we want to make the query activation available to each attention layer of the decoder, we separately customize it to the needs of each layer using a linear layer. That is, for each layer of the transformer part of the decoder (indexed by $\ell$, so $\ell \in \{0, 1, \cdots, L - 1\}$), we define a linear layer $\text{Linear}^{(\ell)} : \mathbb{R}^{d+d_{site}} \rightarrow \mathbb{R}^{d_{decoder}}$ and a layer normalization $\text{LN}^{(\ell)}$, where $d_{decoder}$ is the model dimension of the decoder model:

$$\hat{\mathbf{z}}^{(\ell)} = \text{LN}^{(\ell)}(\text{Linear}^{(\ell)}(\mathbf{z}^{(fn)})) \tag{3}$$

We add superscript $(\ell)$ to the model components to emphasize they are layer-specific. In the $\ell$-th layer of the transformer part, $\hat{\mathbf{z}}^{(\ell)}$ is concatenated with the input of the attention layer along the length axis before computing keys and values, so that each attention head can also attend to $\hat{\mathbf{z}}^{(\ell)}$ in addition to the context tokens. This means that each head in the $\ell$-th layer, instead of attending to $x_1^{(\ell)}, x_2^{(\ell)}, ...,$ now computes its attention weights over $\hat{\mathbf{z}}^{(l)}, x_1^{(\ell)}, x_2^{(\ell)}, ....$ Here $x_t^{(\ell)}$ is the residual stream corresponding to the $t$-th token input into the $\ell$-th layer.

**Motivation for Architecture Design.** At the beginning, we train separate decoders for each activation site, but this is not very scalable when there are many activation sites. In the architecture above mentioned, we use activation site embedding $\mathbf{e}_{act}$ as a signal to trigger different processing, and functions or model components that are needed for all activation sites are shared. Similar reason applies to the linear layer $\text{Linear}^{(\ell)}$, we expect the $\mathbf{z}^q$ should be transformed differently for each transformer layer, but having separate MLP stacks to process $\mathbf{z}^q$ for each layer would largely increase the number of parameters. In preliminary experiments, we also try "encoder-decoder" attention layer. That is, instead of providing query activation in self attention layer, we add new attention layers that analogous to the encoder-decoder attention layer in original transformer architecture, where each token can attend to the processed query activation as well as a blank representation (similar to the function of "BOS", so that "no-op" is possible). However, we do not find significant difference between this design and the aforementioned one. Therefore, other than adding components for processing $\mathbf{z}^q$, we do not modify the decoder-only transformer architecture, so that we can also choose to use pretrained models. We note that there are other possible choices for conditioning the generation on the activation, and we didn't optimize this choice thoroughly.

**Decoder Hyperparameters.** Regarding processing query activation, the decoder has 6 MLP layers, i.e., $P = 6$. The decoder model has 2 transformer layers ($L = 2$), 4 heads per layer, and a model dimension of 256. The attention dropout rate is 0. Other settings are the same as the GPT-2. We use the this architecture for the 3 tasks—character counting, IOI, and 3-digit addition— in the paper. Regarding the factual recall task, we use the same architecture for processing query activation, i.e. $P = 6$, and use pretrained GPT-2 Medium (24 layers) as the transformer part of the decoder.

**Training Details.** We construct the training dataset by feeding in-domain inputs to the probed model, and collect activations from random activation sites and random position as query activations (the choice of activation sites and position is specific to each task and is described later), we also record the activation site they come from. For each input, we could obtain many possible training examples because of many choices of activation sites and position. So we do not iterate over all possible training examples. We sample certain amount of examples to train the decoder for 1 epoch, using constant learning rate of 0.0001 and AdamW optimizer with weight decay of 0.01. Other details (e.g., amount of examples, training steps) are task-specific, and can be found later in their own section.

During training, we regularly calculate the in-preimage rate, which serves as a proxy for generation quality. Concretely, for a fixed set of query activations used for testing, the decoder generates samples with temperature=1, we then compute the fraction of samples inside of $\epsilon$-preimage (with $\epsilon = 0.1$, $D$ as normalized Euclidean distance). The rate is calculated for each activation site. We usually observe difference between the ratios for each activation site, indicating some inverse mappings are easier to learn (we also observe these activation sites tend to have clearer information). Overall, we usually see a continuous improvement on the average rate during training.

# D  Character Counting: More Details and Examples

## D.1  Implementation Details

To construct the dataset, we enumerate all the 3-combinations from the set of lowercase characters defined in ASCII. For each combination, we generate 600 distinct data points by varying the occurrence of each character and the order of the string. The occurrences are sampled uniformly from 1-9 (both inclusive). So the length of the part before the pipe symbol ("|") lies in [3, 27] (not considering "B"). Like 3-digit addition, we split the dataset into train and test sets, which account for 75% and 25% of all data respectively. The input is tokenized on the character level.

The model is a two-layer transformer with one head in each layer, the model dimension is 64. All dropout rates are set to 0. The model is trained with cross-entropy loss on the last token, the answer of the counting task. The model is trained with a batch size of 128 for 100 epochs, using a constant learning rate of 0.0005, weight decay of 0.01, and AdamW [36] optimizer. The training loss is shown in Figure 13, we can see the stair-like pattern. An interesting future direction is to investigate what happens when the loss rapidly decreases using InversionView.

With regard to the decoder model, the architecture is described in C. We select $x^{0,\mathrm{pre}}, x^{i,\mathrm{mid}}, x^{i,\mathrm{post}}, a^{i,j}, m^i$ as the set of activation sites we are interested in, where $i \in \{0, 1\}, j \in \{0\}$, and $m$ denotes MLP layer output. The query activation is sampled from those activations corresponding to only the target character and colon. We sample 100 million training examples (all activation sites are included) and train the decoder with batch size of 512, resulting in roughly 200K steps. During training, as we mentioned before in C, we test the generation quality by measuring in-preimage rate. For those activation sites for which the decoder has a low generation quality, we increase their probability of being sampled in the training data. The final in-preimage rate averaged across activation sites is 67.7%.

## D.2  More Examples of InversionView

See Figures 14 and 15. Here, we show results similar to Figure 3 for other query inputs.

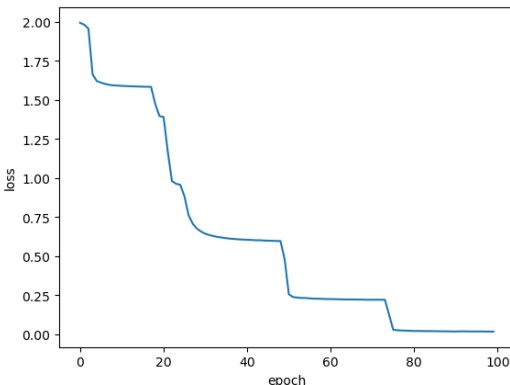

Figure 13: Training loss of the Character Counting task. Each data point is the averaged loss over an epoch.

Activation Site: $x^{0,mid}$

Query Input

   0.000; B q e q q q a q a e q a a a q e q q a a | (a) : 7

Generated Samples

  0.009; #: 7; B a a q q q q q a q a q q a a q e e a e | (a) : 8 E
  0.013; #: 7; B a a q q a q q q a q a q q a q e e a e | (a) : 8 E
  0.049; #: 7; B q e a q a e a q a m a e q q a q e q e a | (a) : 8 E
  0.054; #: 7; B q q a q q q a a q a q a q q a r a r q | (a) : 8 E
  0.065; #: 8; B a q q a q f a q q q q f a q a q a a a | (a) : 9 E
  0.065; #: 7; B a a q q q a q a a g g q q q q q a q a q | (a) : 8 E
  0.066; #: 8; B a a q a q a q q q a q q q q q a q g a a | (a) : 9 E

  0.110; #: 6; B a o o a e e e a a a q q o q e e e q e a | (a) : 7 E
  0.117; #: 7; B a a h c c q c q e a c a h a a h c q q q a | (a) : 8 E
  0.122; #: 6; B g a g a a q g q q a q a h q q q q q a | (a) : 7 E

Activation Site: $x^{0,post}$

Query Input

   0.000; B q e q q q a q a e q a a a q e q q a a | (a) : 7

Generated Samples

  0.006; #: 7; B q q q q q a q q q a a a a a e e q e a | (a) : 7 E
  0.010; #: 7; B a q a a q q q a q a e q e a q q e a q | (a) : 7 E
  0.011; #: 7; B q a q q q q q a q a q a q a a q a e e | (a) : 7 E
  0.016; #: 7; B q q q q a q q q q a q a a a e e a e a | (a) : 7 E
  0.016; #: 7; B a q a a a e a q q e e e q q q q q a a | (a) : 7 E

  0.158; #: 8; B q q n a a q a a q q a g a a a q q q q | (a) : 7 E
  0.180; #: 8; B q a a q q q q q a i q a a a q q q a a | (a) : 7 E
  0.193; #: 6; B e e p q e a a q e p e e a a q p a a p q | (a) : 7 E
  0.207; #: 6; B q q a e e e a q e a e g a d a q q q a | (a) : 7 E
  0.215; #: 6; B q t q q q q a s s a t a a q q q a t a | (a) : 7 E

Figure 14: $\epsilon$-preimage showing function of MLP layer 0

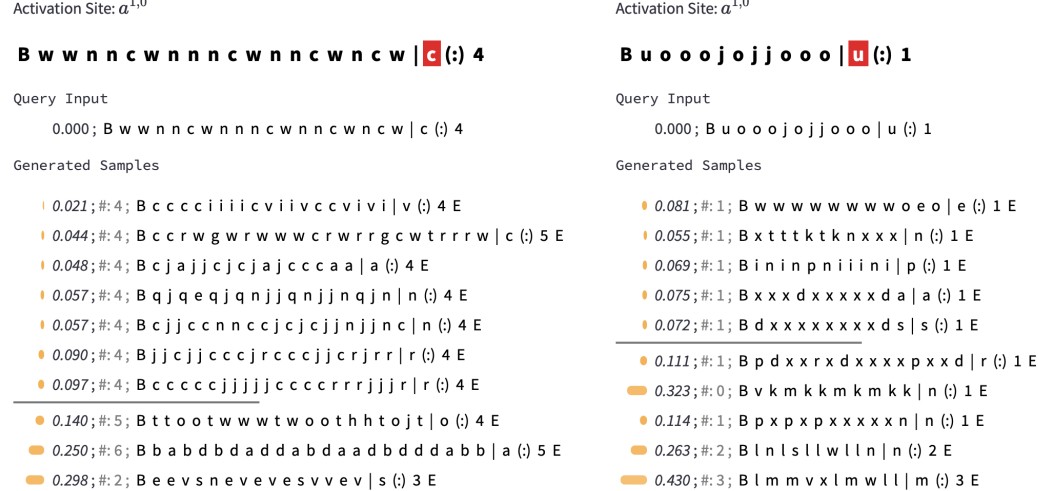

Figure 15: $\epsilon$-preimage of $a_:^{1,0}$. As we mentioned, we hypothesize that the attention head is reading the subspace where the count information is stored. One can presumably find this "count subspace" by optimizing a projection matrix such that after projecting the activation there is only pure count information in the $\epsilon$-preimage, and compare it with the subspace read by the value matrix of the attention head. Therefore, InversionView can be potentially useful for subspace study.

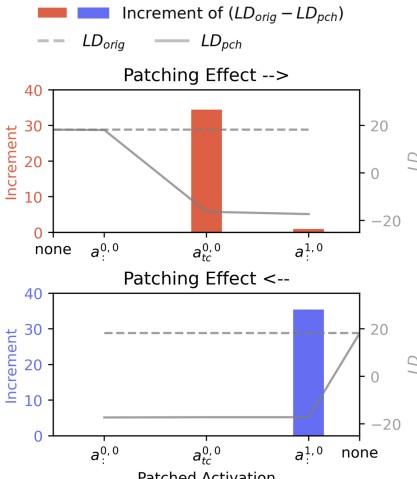

Figure 16: Results of activation patching for model trained on character counting task. Same figure as 4a with intermediate steps of calculation shown using line plot. Note that the gray lines correspond to the y-axis on the right. In contrast examples, only one character differs. $LD$ stands for logit difference between the original count and the count in the contrast example. $LD_{pch}$ and $LD_{orig}$ correspond to the $LD$ with and without patching, respectively. Top: We patch activations cumulatively from left to right, flipping the sign of $LD$. The "none" on the left end of x-axis denotes the starting point, i.e., nothing is patched. Bottom: We patch from right to left. Similarly, "none" on the right end of x-axis denotes the starting point.

### D.3  Causal Intervention Details

For an input example $\mathsf{x}_{orig}$ with $t_{orig}$ as the count (final token), we construct a contrast example $\mathsf{x}_{con}$ with a different count $t_{con}$ by changing a random character before "|". The contrast example is a valid input (the count token is the count of the target character). We also ensure that the contrast example is within the dataset distribution (the count is in the range [1-9] and there are 3 distinct characters in the input).

We run three forward passes. 1) The model takes as input $\mathsf{x}_{orig}$ and produces logit values for count prediction, we record the logit difference $LD_{orig}$ between $t_{orig}$ and $t_{con}$ (former minus latter). 2) We feed the model with $\mathsf{x}_{con}$ and store all activations. 3) We run a forward pass using $\mathsf{x}_{orig}$, replacing the interested activations (e.g, $\{a_{:}^{0,0}, a_{tc}^{0,0}\}$) with the stored activation in the same position and activation sites, and record the new logit difference $LD_{pch}$. Because the model can make the right prediction in most cases, we can see that average $LD_{pch}$ changes from positive to negative values as we patch more and more activations. We do the same for all inputs in the test set and report the average results.

Figure 16 shows the $LD_{orig}$ and $LD_{pch}$. We cumulatively patch the activations we study. For example, on the top of the figure, we patch $\{a_{:}^{0,0}\}$, $\{a_{:}^{0,0}, a_{tc}^{0,0}\}$, $\{a_{:}^{0,0}, a_{tc}^{0,0}, a_{:}^{1,0}\}$ respectively. Patching more activation results in increases of $LD_{orig} - LD_{pch}$, we attribute the increment to the newly patched activation. Hence, the causal effect of each activation is measured conditioned on some activations already being patched.

We sort the activations according to their layer indices and show the results of patching from bottom to top ($\rightarrow$) and from top to bottom ($\leftarrow$). In this way, we can verify the dependence between activations in the top and bottom layer. For example, at the top of Figure 16, when $a_{tc}^{0,0}$ is already patched, patching $a_{:}^{1,0}$ has almost no effect. On the bottom, we also see patching $a_{tc}^{0,0}$ has no effect if $a_{:}^{1,0}$ has been patched. So we verified that $a_{tc}^{0,0}$ is the only upstream activation that $a_{:}^{1,0}$ relies on, and $a_{:}^{1,0}$ is the only downstream activation that reads $a_{tc}^{0,0}$.

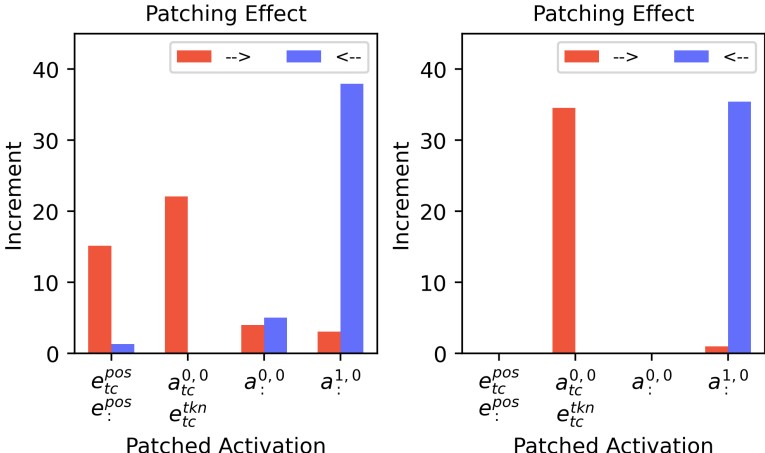

Figure 17: Results of activation patching for model trained on character counting task. $\rightarrow$ and $\leftarrow$ means the same as previously. **Left**: Patching with activation from examples with different counts. **Right**: Patching with activation from examples in which only one character differs.

## D.4 Extended Algorithm with Positional Cues

In Section 3.1, we verified the information flow by an activation patching experiment in which the contrast example only differs by one character. These experiments verified that the algorithm we described is complete in distinguishing between such minimally different contrast examples. We now show that the model implements a somewhat more complex algorithm that combines this algorithm with position-based cues, which become visible once we consider contrast example that differ in more than one character, in particular, those that differ in length.

To show this, we conduct another activation patching experiment in which the contrast example is a random example in the dataset with a different count. In other words, everything can be different in contrast examples, including the sequence length and the target character. Thus, we cumulatively patch four places:

1. $e_{tc}^{pos}$ and $e_{:}^{pos}$, where $e^{pos}$ stands for position embedding, because the final count correlates with positional signal, so the model may utilize it. They are patched together because the attention pattern of the colon in layer 1 relies on their adjacency.

2. $a_{tc}^{0,0}$ (as before) and $e_{tc}^{tkn}$, where $e^{tkn}$ stands for token embedding. They are patched together because patching only one of them would result in a conflict between character information in the patched and the un-patched activation;

3. 3) $a_{:}^{0,0}$; (as before)

4. $a_{:}^{1,0}$ (as before).

The result is shown on the left of Figure 17. We can see that, besides the components we had detected previously based on minimal contrast examples ($a_{tc}^{0,0}$, $a_{:}^{1,0}$), some other signal also contributes notably to the final logits. We compare with patching experiments for the same set of activations on contrast examples that differ in one character, shown on the right of Figure 17.

Overall, besides the algorithm identified in Section 3.1, we find other 3 sources of information influencing the model's output. 1) The position embedding, $e_{tc}^{pos}$ and $e_{:}^{pos}$. This is observable on the left of Figure 17, from which we can also know $a_{:}^{1,0}$ contains the position information (because the bars of $e_{tc}^{pos}$ and $e_{:}^{pos}$ are not symmetric). This is confirmed by InversionView. As shown in Figure 18, we see the inputs in $\epsilon$-preimage roughly follow the query input length, being independent of the count. Therefore, the model is also utilizing the correlation between the count and the sequence length. 2) Attention output of colon, $a_{:}^{0,0}$, which attends to all previous token equally. From InversionView, we observe it contains fuzzy information about the length (same as position signal), and the characters that occur in the context, as well as their approximate count. Our causal experiment also shows that

Activation Site: $a^{1,0}$

**B u u u u b b m u u b m u u b | m (:) 2**

Query Input

0.000 ; B u u u u b b m u u b m u u b | m (:) 2

Generated Samples

| 0.038 ; #: 2 ; B k e k k k e k e e e e e z z k k k | z (:) 2 E
| 0.058 ; #: 2 ; B a b a a b h b a b a a a b a h b b | h (:) 2 E
| 0.059 ; #: 2 ; B u u j u u u a a u u j j u j j j u | a (:) 2 E
| 0.069 ; #: 2 ; B a j h a j j j h a a a j a j j a a j | h (:) 2 E
| 0.071 ; #: 2 ; B v n l v v n l v a v b v v v | n (:) 3 E
| 0.086 ; #: 2 ; B b b b a n a a a a n b a a b a a b b | n (:) 2 E
| 0.092 ; #: 2 ; B m j m a m j m m m j j m m j a m | a (:) 2 E
| 0.126 ; #: 2 ; B e b a e e e b e a e e e b e | a (:) 2 E
| 0.230 ; #: 3 ; B s s p o k o s s s k z k s k s k s k o | o (:) 2 E
| 0.617 ; #: 7 ; B i t v v t t v t v v v i t i t j v v t v | t (:) 3 E

Activation Site: $a^{1,0}$

**B t t t t t n n n t t n t x t | t (:) 9**

Query Input

0.000 ; B t t t t t n n n t t n t x t | t (:) 9

Generated Samples

| 0.022 ; #: 9 ; B t t t t t t t t t s z z z z | t (:) 9 E
| 0.025 ; #: 9 ; B t t m t t l t t m m t m t t | t (:) 9 E
| 0.028 ; #: 9 ; B t t t t t f t f t t f f f t | t (:) 9 E
| 0.034 ; #: 9 ; B t t t t t t t t h r r r r t | t (:) 9 E
| 0.063 ; #: 9 ; B v i c c v j j c c c c v c v c c | c (:) 9 E
| 0.100 ; #: 8 ; B t o s s t o c t c t t l t t t | t (:) 9 E
| 0.111 ; #: 8 ; B t t t t t t p t p t p p p e | t (:) 9 E
| 0.159 ; #: 8 ; B r m c m c r c r c r r r r r c | r (:) 9 E
| 0.170 ; #: 8 ; B p f a q q q p f q f q q d q q | q (:) 9 E
| 0.458 ; #: 0 ; B m n v n b v v b B b b v b b v m m b v v | q (:) 7 E

Figure 18: $\epsilon$-preimage of $a^{1,0}_{:}$ to show the position information is also encoded and is independent of count information.

it does not contain a precise count. Therefore, it contributes to the model's prediction in manner similar to the position signal. 3) Attention output of the pipe sign, $a^{0,0}_{|}$. From the attention pattern we observe sometimes in layer 0, pipe sign attends selectively to one type of character, e.g. "x", "k", or "j". InversionView shows that it indeed contains the approximate count in that case (though the decoder has not been trained on activation corresponding to pipe sign). In next layer, the colon also attends to the pipe sign if target character is the same as the character attended by pipe sign in layer 0. This explains why we can observe nonzero effect of patching $a^{1,0}_{:}$ when other activation is already patched (the red bar corresponding to $a^{1,0}_{:}$ on both sub-figures of Figure 17).

Whereas patching with minimally different contrast examples allowed us to extract an algorithm *sufficient* for solving the task in Section 3.1, patching with arbitrarily different contrast examples allowed us to uncover that the model combines this algorithm with position-based cues. The model performs precise counting using the algorithm we found earlier in Section 3.1, while it also makes use of simple mechanisms such as correlation to obtain a coarse-grained distribution over counts. Overall, we have found the full algorithm by alternating between different methods – InversionView, traditional inspection of attention patterns, and causal interventions, and confirming results from one with others.

# E    IOI Task: Details and Qualitative Results

## E.1    Implementation Details

In order to train a decoder model, we construct a dataset that consists of IOI examples. We used the templates of IOI examples from ACDC [13] implementation. For example, "Then, [B] and [A] went to the [PLACE]. [B] gave a [OBJECT] to [A]", in which "[B]" and "[A]" will be replaced by two random names (one token name), "[PLACE]" and "[OBJECT]" will also be replaced by random item from the predefined set. Besides "BABA" template (i.e., S is before IO) we also use "ABBA" templates (S is after IO) by swapping the first "[B]" and "[A]". We generate 250k data points.

The architecture of the decoder model is the same as before, as described in Appendix C. The set of activation sites the decoder is trained on consists of the output of all attention heads and MLP layers (no residual stream). Note that when producing query activation using GPT-2, we always add the "< |endoftext| >" token as the BOS token. We do so as during the training of GPT-2 this or multiple such tokens that usually appear in the previous context can be used as a BOS token, which is possibly important to the model's functioning. We use a new token "[EOS]" as the EOS token when training the decoder. The query activation is sampled uniformly from all positions excluding EOS and padding tokens, and uniformly from all activation sites the decoder is trained for. We sample 20 million training examples (all activation sites are included) and train the decoder with batch size of

**<|endoftext|>Then in the morning, Stephanie and Nicole were thinking about going to the hospital. Stephanie wanted to `give` a ring( `to`) Nicole**

```
Query Input
```

0.000 ; <|endoftext|>Then in the morning, Stephanie and Nicole were thinking about going to the hospital. Stephanie wanted to give a ring( to) Nicole

```
Generated Samples
```

0.020 ; <|endoftext|>Then in the morning, Erin and Sarah were thinking about going to the house. Erin wanted to give a ring( to) Sarah[EOS]

0.027 ; <|endoftext|>Then in the morning, Sarah and Sarah were thinking about going to the station. Sarah wanted to give a ring( to) Sarah[EOS]

0.039 ; <|endoftext|>Then in the morning, Lindsey and Emily were thinking about going to the station. Lindsey wanted to give a ring( to) Emily[EOS]

0.055 ; <|endoftext|>Then, Lindsey and Anthony were thinking about going to the house. Lindsey wanted to give a necklace( to) Anthony[EOS]

0.062 ; <|endoftext|>Then in the morning, Kelly and Richard were thinking about going to the office. Kelly wanted to give a ring( to) Richard[EOS]

0.075 ; <|endoftext|>Then, Kelly and Patrick were thinking about going to the office. Kelly wanted to give a ring( to) Patrick[EOS]

0.086 ; <|endoftext|>Then, Brian and Katie were thinking about going to the school. Brian wanted to give a ring( to) Katie[EOS]

0.170 ; <|endoftext|>Then, Michelle and David were thinking about going to the restaurant. David wanted to give a ring( to) Michelle[EOS]

0.179 ; <|endoftext|>Then in the morning, John and Amber were thinking about going to the office. Amber wanted to give a necklace( to) John[EOS]

0.317 ; <|endoftext|>Then in the morning, Elizabeth and Tyler were thinking about going to the station. Tyler wanted to give a necklace( to) Elizabeth[EOS]

Figure 19: $\epsilon$-preimage of S-Inhibition Head 7.3. The relative position of S1–but not its identity–is contained in the head output (together with some template information). That means, in the samples within the $\epsilon$-preimage, S1 always appears before the IO. While the relative position is encoded, the absolute position can vary, as can the identities of the names.

**<|endoftext|>Friends separated at birth Bradley and `Cody` found a computer at the store.`( Cody)` gave it to Bradley**

```
Query Input
```

0.000 ; <|endoftext|>Friends separated at birth Bradley and Cody found a computer at the store.( Cody) gave it to Bradley

```
Generated Samples
```

0.000 ; <|endoftext|>Then in the morning afterwards, Cody and Jacob had a long argument. Afterwards Jacob said to( Cody)[EOS]

0.000 ; <|endoftext|>Then in the morning, Cody and Scott had a lot of fun at the school.( Cody) gave a bone to Scott[EOS]

0.000 ; <|endoftext|>Then in the morning afterwards, Cody and Ryan had a long argument. Afterwards( Cody) said to Ryan[EOS]

0.000 ; <|endoftext|>After taking a long break Cody and Lauren went to the station,( Cody) gave a computer to Lauren[EOS]

0.001 ; <|endoftext|>Afterwards, Cody and Danielle went to the restaurant.( Cody) gave a basketball to Danielle[EOS]

0.001 ; <|endoftext|>Then, Cody and Jonathan had a lot of fun at the house. Jonathan gave a bone to( Cody)[EOS]

0.001 ; <|endoftext|>Then, Paul and Cody had a long argument. Afterwards( Cody) said to Paul[EOS]

0.392 ; <|endoftext|>Then in the morning afterwards, Courtney and Dustin had a long argument. Afterwards( Dustin) said to Courtney[EOS]

0.392 ; <|endoftext|>Friends separated at birth Alicia and Dustin found a snack at the school.( Dustin) gave it to Alicia[EOS]

0.395 ; <|endoftext|>When soon afterwards Jason and Nicholas got a snack at the hospital, Nicholas decided to give the snack to( Jason)[EOS]

Figure 20: $\epsilon$-preimage of Duplicate Token Head 0.1. S name is contained in head output.

256, resulting in roughly 80K steps. The final average in-preimage rate is 58.0%, despite that the decoder is trained for 157 activation sites.

## E.2 More Examples of InversionView

See Figures 19, 20, 21.

## E.3 Qualitative Examination Results

The qualitative examination results is shown in Table 2. We summarize the description in [54] of each head category to facilitate comparison. Figure 22 shows the IOI circuit in GPT-2 small, which is taken from their paper. For more details, please refer to [54].

There are some heads for which InversionView indicates a different interpretation. First, Head 0.1 and 0.10: [54] only shows that they usually attend to the previous occurrence of a duplicate token and validates the attention pattern on different datasets. However, there is no evidence for the information moved by these heads. Thus they only hypothesize that the position of previous occurrence is copied. Second, Head 5.9: The path patching experiments in [54] show that head 5.9 influences the final logits notably via S-Inhibition Heads' keys. But there are no further experiments to explain the concrete function of this head. While the authors refer to it as a Fuzzy Induction Head, the induction score (measured by the attention weight from a token $T$ to the token after $T$'s last occurrence) of this head shows a very weak induction pattern. Even if such pattern occurs, it cannot tell us what

**<|endoftext|>After taking a long break Bradley** `and` **Rebecca went to the store,( Bradley) gave a basketball to Rebecca**

```
Query Input
    0.000 ; <|endoftext|>After taking a long break Bradley and Rebecca went to the store,( Bradley) gave a basketball to Rebecca

Generated Samples
    0.002 ; <|endoftext|>After taking a long break Bradley and Gregory went to the station,( Bradley) gave a computer to Gregory[EOS]
    0.014 ; <|endoftext|>After taking a long break Kelly and Aaron went to the restaurant,( Kelly) gave a bone to Aaron[EOS]
    0.015 ; <|endoftext|>After taking a long break Kelly and Aaron went to the hospital,( Kelly) gave a necklace to Aaron[EOS]
    0.019 ; <|endoftext|>After taking a long break Kyle and Paul went to the hospital,( Kyle) gave a snack to Paul[EOS]
    0.020 ; <|endoftext|>After taking a long break Travis and Andrea went to the restaurant, Andrea gave a kiss to( Travis)[EOS]
    0.022 ; <|endoftext|>After taking a long break Megan and Benjamin went to the restaurant, Benjamin gave a ring to( Megan)[EOS]
    0.024 ; <|endoftext|>After taking a long break Lauren and Joshua went to the garden, Joshua gave a necklace to( Lauren)[EOS]
    0.028 ; <|endoftext|>After taking a long break Michelle and Amanda went to the restaurant,( Michelle) gave a kiss to Amanda[EOS]
    0.032 ; <|endoftext|>While spending time together Dustin and Sarah were working at the station, Sarah gave a snack to( Dustin)[EOS]

    0.370 ; <|endoftext|>After taking a long break Bradley and Andrea went to the restaurant, Patrick gave a computer to( Andrea)[EOS]
```

Figure 21: $\epsilon$-preimage of Induction Head 5.5. Position – but not identity – of the current token (token in parenthesis)'s last occurrence is contained in head output

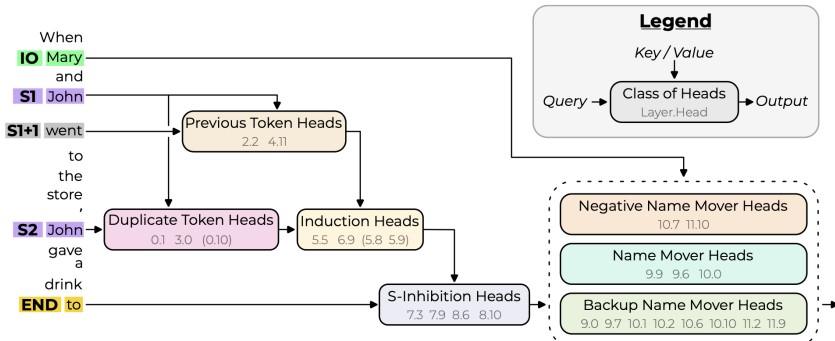

Figure 22: IOI circuit in GPT-2 small. Figure 2 from [54]

information is captured by this head. Interpretation with InversionView suggests that the head barely contains any information, within the input space of IOI-like patterns. One possibility is that head 5.9 recognizes the IOI pattern (i.e., there are two names and one is duplicated in the previous context), so that if an IOI-like pattern exists, S2 should be attended to by S-Inhibition heads. As the decoder model is trained on IOI examples and generates mostly IOI examples – that is, the input space $\mathcal{X}$ in (1) consists of IOI-like inputs, this information is by definition not visible. Expanding the input space to arbitrary language modeling would allow capturing such information; we leave this to future work.

### E.4 Choice of Distance Metric in IOI

As described in Section 3.2, we used cosine distance for the IOI task, while for the other two tasks, we use normalized Euclidean distance. In this section, we show that both distance metrics produce similar interpretations while cosine distance makes the meaningful patterns easier to identify.

In Figure 23 and 24 we show two examples of using normalized Euclidean distance as the distance metric. Readers can see more examples on our web application. We see that the samples generated by the decoder tend to have larger distances under normalized Euclidean distance. Nonetheless, we can still see the top-ranked samples show meaningful commonalities – indeed if we set $\epsilon = 0.4$, we obtain the same interpretation as the one we obtain based on Figure 4b and 19.

While normalized Euclidean distance can produce a similar interpretation, we have to set a much larger $\epsilon$, which we find less intuitive. We believe this to be because of the differences in the activation dimensionality, which is 768 in the IOI task, much larger than in character counting (64) and addition (32) tasks: Under Euclidean distance, the ratio of all close samples to all possible samples becomes lower and lower when the dimension becomes higher. In other words, the volume of the $\epsilon$-preimage accounts for a very tiny proportion of the whole space in the high-dimensional case. By using cosine

| Head Category | Function According to [54] | Position | Observation from InversionView | Whether Consistent |
|---|---|---|---|---|
| Name Mover Heads | Copy whatever it attends to (increase its logit). It will always attend to IO because of previous components. | END | 9.9 9.6 10.0: IO name | Yes |
| Negative Name Mover Heads | Copy whatever it attends to (decrease its logit), it attends to IO | END | 10.7 11.10: IO name | Yes |
| S-Inhibition Heads | Move info about S and cause Name Mover Heads to attend less to S. They attend to S2. They adds two signals to residual stream, one is token value of S, the other is position of S1, and position signal has a greater effect. | END | 7.3 8.6: Position of S1 (relative position to IO, same for the following part when we say position) 
 7.9: S name; Position of S1 
 8.10: S name; Position of S1 (most of the time) | Yes |
| Duplicate Token Heads | Attend from S2 to S1, more generally, attend to previous duplicate token, and copy the position of this previous occurrence. | S2 | 0.1 0.10: **S name (instead of position)** 
 3.0: Position of the duplicated name | Partly |
| Previous Token Heads | Attend the previous token, move the token (name) to S1+1, then this information is used as key of S1+1 in Induction Heads | S1+1 | 2.2: S name (most of the time, sometimes attend to "and" when S1 is after IO and make S name less important) 
 4.11: S name; Position of the previous token (relative to the other name) | Yes |
| Induction Heads | Attend to S1+1 and move position signal (similar to the function of Duplicate Token Heads, while different from normal induction heads) | S2 | 5.5 6.9: Position of the current token's last occurrence 
 5.8: Position of the current token's last occurrence (most of the time) 
 5.9: **Almost no info**, possibly some info about the template. | Partly |
| Backup Name Mover Heads | Do not move IO normally, but act as Name Mover Heads when they are knocked out. 
 There are 4 categories: 
 9.0, 10.1, 10.10, 10.6: similar to Name Mover 
 10.2, 11.9: attend to both S1 and IO, and move both 
 11.2: attends to S1 and move S 
 9.7: attends to S2 and writes negatively | END | 9.0 10.1 10.10: IO name 
 10.6 : S and/or IO name (Most of the time: IO, sometimes: IO and S, occasionally: S) 
 10.2: S and IO name (most of the time), S or IO name (sometimes) 
 11.9: S and/or IO name (no obvious difference in frequency) 
 11.2: S and IO name (most of the time), S name (sometimes) 
 9.7: S name | Yes |

Table 2: Column "Position" means the query activation is taken from that position. "S1+1" means the token right after S1. Rows are ordered according to the narration in the original paper. When we say "S name", it means the the name of S in the query input, but the name is not necessarily S in the samples. This also applies to "IO name". The information learned by InversionView which is different from the information suggested by Wang et al. [54] is in **bold**.

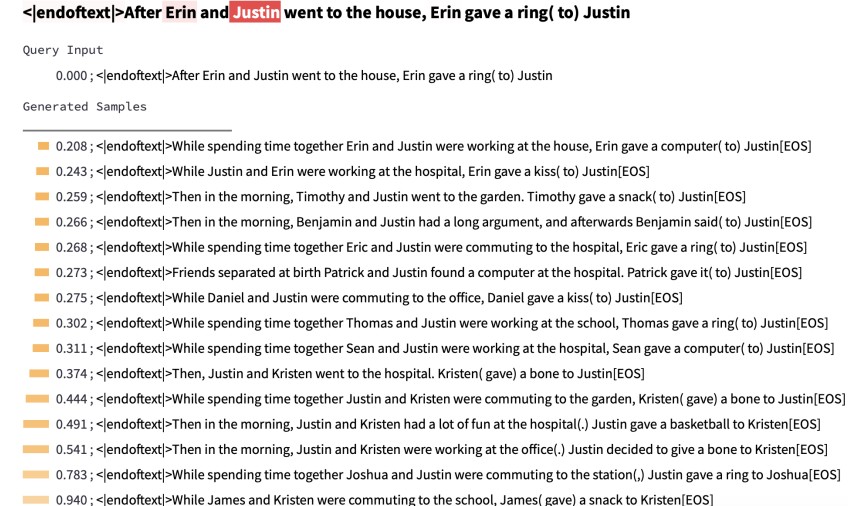

**<|endoftext|>After Erin and `Justin` went to the house, Erin gave a ring( to) Justin**

`Query Input`

0.000 ; <|endoftext|>After Erin and Justin went to the house, Erin gave a ring( to) Justin

`Generated Samples`

0.208 ; <|endoftext|>While spending time together Erin and Justin were working at the house, Erin gave a computer( to) Justin[EOS]
0.243 ; <|endoftext|>While Justin and Erin were working at the hospital, Erin gave a kiss( to) Justin[EOS]
0.259 ; <|endoftext|>Then in the morning, Timothy and Justin went to the garden. Timothy gave a snack( to) Justin[EOS]
0.266 ; <|endoftext|>Then in the morning, Benjamin and Justin had a long argument, and afterwards Benjamin said( to) Justin[EOS]
0.268 ; <|endoftext|>While spending time together Eric and Justin were commuting to the hospital, Eric gave a ring( to) Justin[EOS]
0.273 ; <|endoftext|>Friends separated at birth Patrick and Justin found a computer at the hospital. Patrick gave it( to) Justin[EOS]
0.275 ; <|endoftext|>While Daniel and Justin were commuting to the office, Daniel gave a kiss( to) Justin[EOS]
0.302 ; <|endoftext|>While spending time together Thomas and Justin were working at the school, Thomas gave a ring( to) Justin[EOS]
0.311 ; <|endoftext|>While spending time together Sean and Justin were working at the hospital, Sean gave a computer( to) Justin[EOS]
0.374 ; <|endoftext|>Then, Justin and Kristen went to the hospital. Kristen( gave) a bone to Justin[EOS]
0.444 ; <|endoftext|>While spending time together Justin and Kristen were commuting to the garden, Kristen( gave) a bone to Justin[EOS]
0.491 ; <|endoftext|>Then in the morning, Justin and Kristen had a lot of fun at the hospital(.) Justin gave a basketball to Kristen[EOS]
0.541 ; <|endoftext|>Then in the morning, Justin and Kristen were working at the office(.) Justin decided to give a bone to Kristen[EOS]
0.783 ; <|endoftext|>While spending time together Joshua and Justin were commuting to the station(,) Justin gave a ring to Joshua[EOS]
0.940 ; <|endoftext|>While James and Kristen were commuting to the school, James( gave) a snack to Kristen[EOS]

Figure 23: $\epsilon$-preimage of the same activation as Figure 4b using normalized Euclidean distance instead of cosine similarity (Appendix E.4). The line shown in the figure still represents threshold of 0.1. While at $\epsilon = 0.1$, some query inputs do result in a substantial number of in-$\epsilon$-preimage samples, many cases result in very small or even empty (as here) sample sets, suggesting that the $\epsilon$-preimage at 0.1 – at least as accessible to the decoder – is extremely small in this model, which we speculate is related to the models higher dimensionality compared to the other tasks (768 vs 32/64), which tends to make Euclidean distances large except for extremely similar vectors (see Appendix E.4 for more discussion). We find it more convenient and natural to use similar thresholds ($\epsilon = 0.1$) across tasks and account for the different geometries using different distance metrics. What is key, however, is that for an appropriately higher $\epsilon$ (e.g., $\epsilon = 0.4$) we again always obtain a substantial number of samples, and – most importantly – these samples lead to the same interpretation as we obtained in our main experiments with the cosine similarity. Here, for example, we can obtain the same interpretation by setting $\epsilon = 0.4$, i.e., the IO name is encoded in the activation.

---

**<|endoftext|>Then in the morning, Stephanie and Nicole were thinking about going to the hospital. Stephanie wanted to `give` a ring( to) Nicole**

`Query Input`

0.000 ; <|endoftext|>Then in the morning, Stephanie and Nicole were thinking about going to the hospital. Stephanie wanted to give a ring( to) Nicole

`Generated Samples`

0.186 ; <|endoftext|>Then, Michelle and Patrick were thinking about going to the hospital. Michelle wanted to give a ring( to) Patrick[EOS]
0.195 ; <|endoftext|>Then in the morning, Sarah and Jason were thinking about going to the station. Sarah wanted to give a ring( to) Jason[EOS]
0.239 ; <|endoftext|>Then in the morning, Lindsey and Joseph were thinking about going to the garden. Lindsey wanted to give a ring( to) Joseph[EOS]
0.277 ; <|endoftext|>Then, Shannon and Erin were thinking about going to the office. Shannon wanted to give a ring( to) Erin[EOS]
0.277 ; <|endoftext|>Then in the morning, Katherine and William were thinking about going to the store. Katherine wanted to give a ring( to) William[EOS]
0.285 ; <|endoftext|>Then, Kelly and Rachel were thinking about going to the house. Kelly wanted to give a necklace( to) Rachel[EOS]
0.302 ; <|endoftext|>Then in the morning, Kelly and Courtney were thinking about going to the house. Kelly wanted to give a ring( to) Courtney[EOS]
0.317 ; <|endoftext|>Then in the morning, Jamie and Robert were thinking about going to the garden. Jamie wanted to give a necklace( to) Robert[EOS]
0.323 ; <|endoftext|>Then in the morning, Lindsey and Bryan were thinking about going to the station. Lindsey wanted to give a necklace( to) Bryan[EOS]
0.331 ; <|endoftext|>Then, Lindsey and Jessica were thinking about going to the office. Lindsey wanted to give a necklace( to) Jessica[EOS]
0.331 ; <|endoftext|>Then in the morning, Allison and William were thinking about going to the house. Allison wanted to give a basketball( to) William[EOS]
0.476 ; <|endoftext|>Then in the morning, John and Amy were thinking about going to the garden. Amy wanted to give a ring( to) John[EOS]
0.484 ; <|endoftext|>Then in the morning, Richard and Scott were thinking about going to the garden. Scott wanted to give a ring( to) Richard[EOS]
0.512 ; <|endoftext|>Then, Jamie and Melissa were thinking about going to the store. Melissa wanted to give a ring( to) Jamie[EOS]
0.539 ; <|endoftext|>Then in the morning, Lindsey and Christina were thinking about going to the garden. Christina wanted to give a bone( to) Lindsey[EOS]

Figure 24: $\epsilon$-preimage of the same activation as Figure 19 using normalized Euclidean distance. The line shown in the figure still represents the threshold of 0.1. As explained in Figure 23, due to the representation geometry, normalized Euclidean distance tends to require a much higher threshold to obtain a sufficient sample size for interpretation. Importantly, as also explained there, we still obtain the same interpretation as in our main experiments if we use normalized Euclidean distance but take a higher threshold (e..g, $\epsilon = 0.4$): here, the relative position of S1 is encoded in the activation.

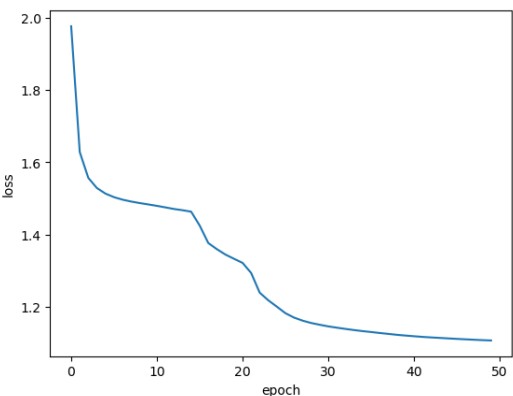

Figure 25: Training loss of the 3-digit addition task. Each data point is the averaged loss over an epoch. The final loss is still big since the two operands of the addition is unpredictable.

similarity, we allow more input samples to lie in a close distance with the query input, as cosine similarity ignores the magnitude difference. This suggests cosine similarity may overall be more suitable when applying InversionView in high-dimensional activations.

# F    3-Digit Addition: More Details and Examples

## F.1    Implementation Details

We constructed 810,000 instances and applied a random 75%-25% train-test split. The probed model is trained with a constant learning rate of 0.0001, a batch size of 512, for 50 epochs (59350 steps). We save the last checkpoint as the trained model, and test it on the test set. We use the AdamW [36] optimizer with weight decay of 0.01. The training loss is shown in Figure 25.

With regard to the decoder model, we select $x^{0,\mathrm{pre}}, x^{i,\mathrm{mid}}, x^{i,\mathrm{post}} a^{i,j}$ as the set of activation sites we are interested, where $i \in \{0, 1\}, j \in \{0, 1, 2, 3\}$. When sampling training data, we select an activation site and a token position uniformly at random. We sample 100 million training examples (all activation sites are included) and train the decoder with batch size of 512, resulting in roughly 200K steps. The final average in-preimage rate is 88.6%.

## F.2    More Examples of InversionView

See Figures 26 to 29.[5]

## F.3    Model Deficiency

In Section 3.3, we mention that there is no firm and clear path of obtaining the carry from ones to tens, so the model may make wrong prediction. We examine those instances for which the model makes wrong prediction and find they all satisfy one condition: F2+S2=9. In other words, it fails to make the right prediction because the ones place matters. This is consistent with our interpretation of the model. Furthermore, we check the model's accuracy on this special subset where F2+S2=9, and find that it is significantly higher than chance level. The accuracy on training subset (training data where F2+S2=9) is 80.45%, and on test subset is 80.06%, while chance level is 50%. So, we can infer that the probed model obtains some information about the ones place by means other than memorization. Indeed, we observe fuzzy information about ones place in $a^{1,0}$ and $a^{1,1}$ occasionally (See Figure 30).

---

[5]We have also tried value-weighted attention pattern [32]; it makes a negligible difference, so we always show the original attention pattern in our paper.

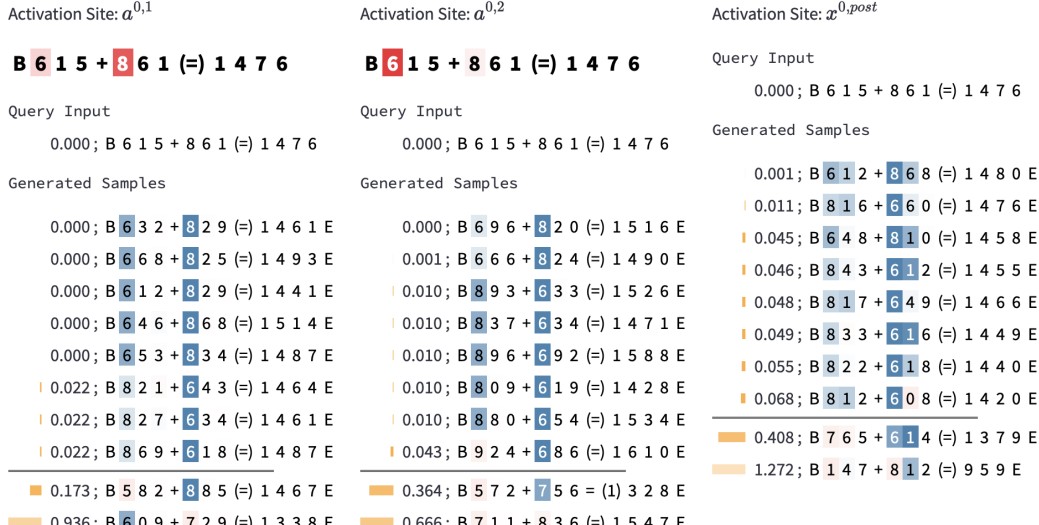

Figure 26: The $\epsilon$-preimage of $a_{\underline{\underline{=}}}^{0,1}$, $a_{\underline{\underline{=}}}^{0,2}$ and $x_{\underline{\underline{=}}}^{0,\mathrm{post}}$ for the same query input as Figure 5.

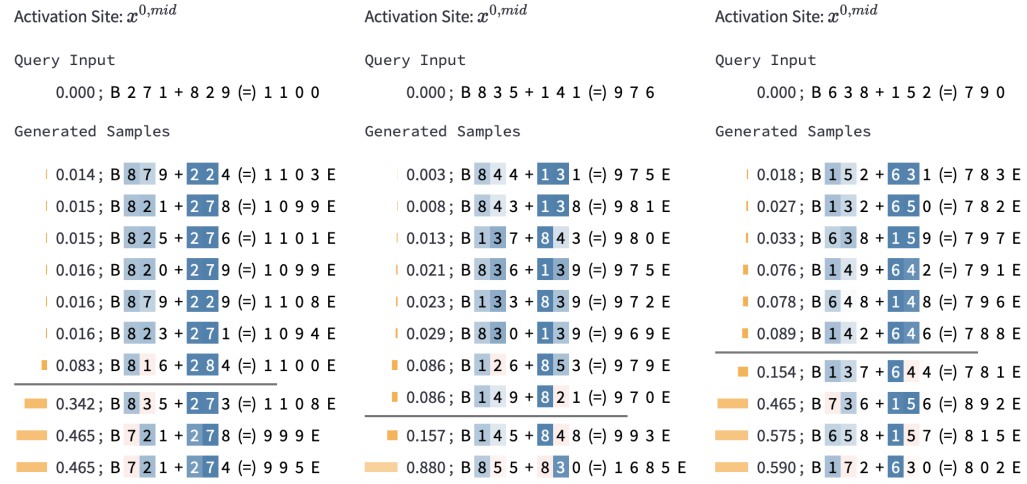

Figure 27: The $\epsilon$-preimage of $x_{\underline{\underline{=}}}^{0,\mathrm{mid}}$ of different examples.

## F.4 Qualitative Examination Results

We present our overall qualitative results in Figure 31 and Table 3. We have found that the model obtains required digits by attention, and primary digits are assigned more heads than secondary digits, e.g., $a^{0,0}$, $a^{0,1}$, $a^{0,2}$ for hundreds and $a^{0,3}$ for tens when predicting A1. More importantly, the primary digits are encoded precisely while the secondary digits are encoded approximately in the residual stream. In addition, the model routes the information differently based on whether A1=1, i.e., the length of the answer is 3 or 4. When predicting A2, this information is known before layer 0, thus paths differ from the start. On the contrary, when predicting A3, the information is obtained in layer 0, thus paths differ only in layer 1. Furthermore, in Figure 31, sub-figure (c) and (d) are very similar, indicating model uses almost the same algorithm to predict digit in tens place. While sub-figure (e) shares the layer 0 with (d), its layer 1 is similar to (f).

## F.5 Causal Intervention: Details and Full Results

We use similar activation patching method as character counting task, described in D.3. In Figure 31, we split the overall algorithm into individual ones for digits in the answer, under different

| Pred Target | A1 | A2 | A3 | A4 / E |
|---|---|---|---|---|
| $a^{0,0}$ | 1-2 digits from F1 and S1 | whether A1=1 | whether A1=1, so the model knows A3 is in tens or ones place | whether A1=1, so the model knows the next token is A4 or E |
| $a^{0,1}$ | | **If** A1=1: Almost no info **Else**: 1-2 digits from F2 and S2 | 1-2 digits from F2 and S2 | 1-2 digits from F3 and S3 |
| $a^{0,2}$ | | **If** A=1: both F1 and S1 **Else**: 1-2 digits from F2 and S2 | | |
| $a^{0,3}$ | 1-2 digits from F2 and S2; C2 | **If** A=1: 1-2 digits from F1 and S1 **Else**: 1-2 digits from F3 and S3; C3 | 1 digit from F3 and S3 (2 when F3=S3); C3 | |
| $x^{0,\mathrm{mid}}$ | F1 and S1; C2 | A1 **If** A1=1: F1 and S1 **Else**: F2 and S2; C3 | A2; F2 and S2; C3; whether A1=1 | A3; F3 and S3; whether A1=1 |
| $x^{0,\mathrm{post}}$ | same as $x^{0,\mathrm{mid}}$ | same as $x^{0,\mathrm{mid}}$ | same as $x^{0,\mathrm{mid}}$ | same as $x^{0,\mathrm{mid}}$ |
| $a^{1,0}$ | Fuzzy info about F1, S1 and C2; Fuzzy info about F3 and S3 (sometimes) | **If** A1=1: F1 and S1 (sometimes fuzzy); C2 (sometimes) **Else**: 1-2 digits from F2 and S2; | **If** A1=1: 1-2 digits from F2 and S2; C3 (sometimes); For $a^{1,2}$, also 1-2 digits from F1 and S1 (fuzzy); **Else**: 1-2 digits from F3 and S3 | 1-2 digits from F3 and S3; For $a^{1,0}$ and $a^{1,1}$, they also contain info about whether next token should be E |
| $a^{1,1}$ | **If** A1=1 (likely to be): 1-2 digit from F1 and S1 (sometimes their sum); C2; **Else**: Fuzzy info, including some info about F3 and S3 (sometimes) | **If** A1=1: Uncertain. F1 and S1 (sometimes); 1 digit from F3 and S3 (sometimes) **Else**: F2 and S2 | | |
| $a^{1,2}$ | Fuzzy info about F2 and S2 (sometimes) | 1-2 digits from F1 and S1 (sometimes fuzzy); 1-2 digits from F2 and S2 (sometimes fuzzy); | | |
| $a^{1,3}$ | 1-2 digits from F1 and S1 (sometimes fuzzy) | **If** A1=1: F1 and S1 (sometimes fuzzy); C2 **Else**: F2 and S2 | | |
| $x^{1,\mathrm{mid}}$ | same as $x^{0,\mathrm{mid}}$ | **If** A1=1: info from $x^{0,\mathrm{mid}}$ + C2 **Else**: same as $x^{0,\mathrm{mid}}$ | **If** A1=1: same as $x^{0,\mathrm{mid}}$ **Else**: info from $x^{0,\mathrm{mid}}$ + F3 and S3 | same as $x^{0,\mathrm{mid}}$ |
| $x^{1,\mathrm{post}}$ | same as $x^{0,\mathrm{mid}}$ | same as $x^{1,\mathrm{mid}}$ except that current input token A1 is less important | same as $x^{1,\mathrm{mid}}$ except that current input token A2 is less important | same as $x^{0,\mathrm{mid}}$ except that current input token A3 is blurred |

Table 3: Summary of our observations for each activation site and position. "same as" denotes that there is no obvious difference between the two sites for indicated position.

Activation Site: $a^{1,1}$

**B 5 6 2 + 1 1 9 = 6 8 (1)**

Query Input

 0.000; B 5 6 2 + 1 1 9 = 6 8 (1)

Generated Samples

 0.029; B 4 8 2 + 2 9 9 = 7 8 (1) E
 0.033; B 3 7 2 + 5 0 9 = 8 8 (1) E
 0.033; B 3 7 2 + 5 0 9 = 8 8 (1) E
 0.046; B 1 4 2 + 7 5 9 = 9 0 (1) E
 0.047; B 2 9 2 + 4 1 9 = 7 1 (1) E
 0.048; B 4 1 2 + 4 4 9 = 8 6 (1) E
 0.053; B 1 6 2 + 7 6 9 = 9 3 (1) E
 0.114; B 4 3 1 + 2 3 9 = 6 7 (0) E
 0.261; B 7 9 1 + 6 6 8 = 1 4 5 (9) E
 0.271; B 7 7 2 + 4 1 9 = 1 1 9 (1) E

Activation Site: $a^{1,3}$

**B 2 6 8 + 8 9 9 = 1 (1) 6 7**

Query Input

 0.000; B 2 6 8 + 8 9 9 = 1 (1) 6 7

Generated Samples

 0.010; B 2 6 9 + 8 9 8 = 1 (1) 6 7 E
 0.011; B 8 6 9 + 2 9 8 = 1 (1) 6 7 E
 0.020; B 8 6 7 + 2 9 9 = 1 (1) 6 6 E
 0.067; B 8 6 8 + 2 9 8 = 1 (1) 6 6 E
 0.071; B 1 6 2 + 9 9 9 = 1 (1) 6 1 E
 0.073; B 8 6 5 + 2 9 9 = 1 (1) 6 4 E
 0.079; B 6 6 6 + 4 9 9 = 1 (1) 6 5 E
 0.141; B 9 6 9 + 1 9 6 = 1 (1) 6 5 E
 0.260; B 7 7 9 + 9 9 6 = 1 (7) 7 5 E
 0.291; B 3 6 4 + 1 9 7 = (5) 6 1 E

Activation Site: $a^{1,3}$

**B 1 0 7 + 5 5 9 = 6 (6) 6**

Query Input

 0.000; B 1 0 7 + 5 5 9 = 6 (6) 6

Generated Samples

 0.000; B 3 0 7 + 6 8 9 = 9 (9) 6 E
 0.000; B 2 0 7 + 8 2 9 = 1 0 (3) 6 E
 0.001; B 3 7 7 + 9 5 9 = 1 3 (3) 6 E
 0.004; B 3 9 7 + 8 0 9 = 1 2 (0) 6 E
 0.004; B 2 4 7 + 7 9 9 = 1 0 (4) 6 E
 0.005; B 7 6 7 + 6 4 9 = 1 4 (1) 6 E
 0.005; B 3 4 7 + 3 4 9 = 6 (9) 6 E
 0.045; B 1 6 7 + 7 3 8 = 9 0 (5) E
 0.628; B 7 3 6 + 7 2 8 = 1 4 6 (4) E
 0.706; B 8 3 6 + 8 6 9 = 1 7 0 (5) E

Figure 28: $\epsilon$-preimage of more examples

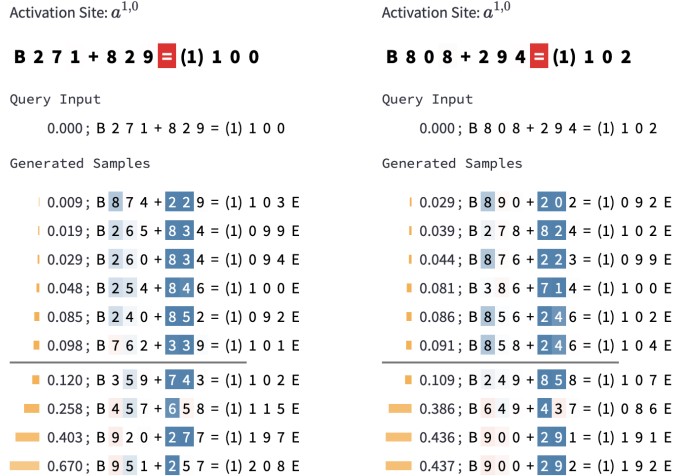

Activation Site: $a^{1,0}$

**B 2 7 1 + 8 2 9 = (1) 1 0 0**

Query Input

 0.000; B 2 7 1 + 8 2 9 = (1) 1 0 0

Generated Samples

 0.009; B 8 7 4 + 2 2 9 = (1) 1 0 3 E
 0.019; B 2 6 5 + 8 3 4 = (1) 0 9 9 E
 0.029; B 2 6 0 + 8 3 4 = (1) 0 9 4 E
 0.048; B 2 5 4 + 8 4 6 = (1) 1 0 0 E
 0.085; B 2 4 0 + 8 5 2 = (1) 0 9 2 E
 0.098; B 7 6 2 + 3 3 9 = (1) 1 0 1 E
 0.120; B 3 5 9 + 7 4 3 = (1) 1 0 2 E
 0.258; B 4 5 7 + 6 5 8 = (1) 1 1 5 E
 0.403; B 9 2 0 + 2 7 7 = (1) 1 9 7 E
 0.670; B 9 5 1 + 2 5 7 = (1) 2 0 8 E

Activation Site: $a^{1,0}$

**B 8 0 8 + 2 9 4 = (1) 1 0 2**

Query Input

 0.000; B 8 0 8 + 2 9 4 = (1) 1 0 2

Generated Samples

 0.029; B 8 9 0 + 2 0 2 = (1) 0 9 2 E
 0.039; B 2 7 8 + 8 2 4 = (1) 1 0 2 E
 0.044; B 8 7 6 + 2 2 3 = (1) 0 9 9 E
 0.081; B 3 8 6 + 7 1 4 = (1) 1 0 0 E
 0.086; B 8 5 6 + 2 4 6 = (1) 1 0 2 E
 0.091; B 8 5 8 + 2 4 6 = (1) 1 0 4 E
 0.109; B 2 4 9 + 8 5 8 = (1) 1 0 7 E
 0.386; B 6 4 9 + 4 3 7 = (1) 0 8 6 E
 0.436; B 9 0 0 + 2 9 1 = (1) 1 9 1 E
 0.437; B 9 0 0 + 2 9 2 = (1) 1 9 2 E

Figure 29: Some examples where we can see intermediate states between representing digits separately and representing digits as their sum. In these examples, we see in the $\epsilon$-preimage the digits in hundreds place are either (2,8) or (3,7), while the digits in tens place are mostly encoded as their sum.

condition. Each algorithm predict target token based on multiple types of information (digits in hundreds/tens/ones place).

In order to verify the paths responsible for routing each type of information, we construct contrast examples as follows: Given a prediction target (e.g., A1), a set of tokens that can be changed $T_{chg}$ (corresponds to a certain type information, e.g., F1 and S1), we construct a contrast example $\mathsf{x}_{con}$ that contains a different token $t_{con}$ as prediction target by changing tokens in $T_{chg}$. Note that the contrast example still follows the rule that the answer is the sum of the two operands.

We now give a detailed explanation for Figure 6b shown in the main paper, in which prediction target is A1 and we patch head output corresponding to the preceding token "=". On the left of Figure 6b, $T_{chg} = \{F1, S1\}$. So in the contrast examples the F1 and S1 are changed and other digits in operands remains the same. In the third run where we calculate $LD_{pch}$, activations are replaced by new activations from contrast example, so the new activations contain modified F1 and S1 information. Therefore, for activations that contributes to routing F1 and S1 (e.g., $a^{0,0}_{=}$), patching them with new activations can effect model's prediction. On the contrary, patching $a^{0,3}_{=}$ has no effect

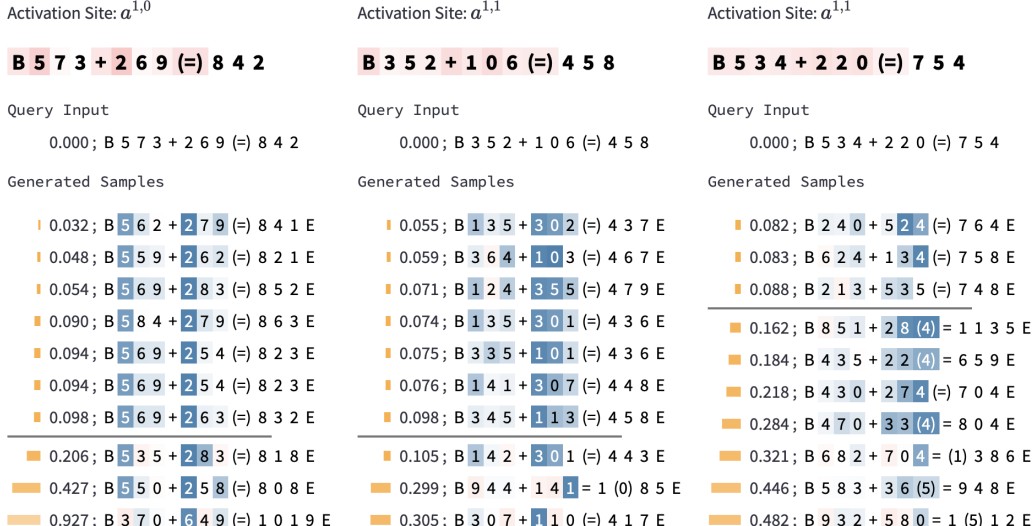

**Activation Site: $a^{1,0}$**

**B 5 7 3 + 2 6 9 (=) 8 4 2**

Query Input

   0.000; B 5 7 3 + 2 6 9 (=) 8 4 2

Generated Samples

  0.032; B 5 6 2 + 2 7 9 (=) 8 4 1 E
  0.048; B 5 5 9 + 2 6 2 (=) 8 2 1 E
  0.054; B 5 6 9 + 2 8 3 (=) 8 5 2 E
  0.090; B 5 8 4 + 2 7 9 (=) 8 6 3 E
  0.094; B 5 6 9 + 2 5 4 (=) 8 2 3 E
  0.094; B 5 6 9 + 2 5 4 (=) 8 2 3 E
  0.098; B 5 6 9 + 2 6 3 (=) 8 3 2 E

  0.206; B 5 3 5 + 2 8 3 (=) 8 1 8 E
  0.427; B 5 5 0 + 2 5 8 (=) 8 0 8 E
  0.927; B 3 7 0 + 6 4 9 (=) 1 0 1 9 E

**Activation Site: $a^{1,1}$**

**B 3 5 2 + 1 0 6 (=) 4 5 8**

Query Input

   0.000; B 3 5 2 + 1 0 6 (=) 4 5 8

Generated Samples

  0.055; B 1 3 5 + 3 0 2 (=) 4 3 7 E
  0.059; B 3 6 4 + 1 0 3 (=) 4 6 7 E
  0.071; B 1 2 4 + 3 5 5 (=) 4 7 9 E
  0.074; B 1 3 5 + 3 0 1 (=) 4 3 6 E
  0.075; B 3 3 5 + 1 0 1 (=) 4 3 6 E
  0.076; B 1 4 1 + 3 0 7 (=) 4 4 8 E
  0.098; B 3 4 5 + 1 1 3 (=) 4 5 8 E

  0.105; B 1 4 2 + 3 0 1 (=) 4 4 3 E
  0.299; B 9 4 4 + 1 4 1 = 1 (0) 8 5 E
  0.305; B 3 0 7 + 1 1 0 (=) 4 1 7 E

**Activation Site: $a^{1,1}$**

**B 5 3 4 + 2 2 0 (=) 7 5 4**

Query Input

   0.000; B 5 3 4 + 2 2 0 (=) 7 5 4

Generated Samples

  0.082; B 2 4 0 + 5 2 4 (=) 7 6 4 E
  0.083; B 6 2 4 + 1 3 4 (=) 7 5 8 E
  0.088; B 2 1 3 + 5 3 5 (=) 7 4 8 E

  0.162; B 8 5 1 + 2 8 (4) = 1 1 3 5 E
  0.184; B 4 3 5 + 2 2 (4) = 6 5 9 E
  0.218; B 4 3 0 + 2 7 4 (=) 7 0 4 E
  0.284; B 4 7 0 + 3 3 (4) = 8 0 4 E
  0.321; B 6 8 2 + 7 0 4 = (1) 3 8 6 E
  0.446; B 5 8 3 + 3 6 (5) = 9 4 8 E
  0.482; B 9 3 2 + 5 8 0 = 1 (5) 1 2 E

Figure 30: Some examples where information about ones place is also encoded.

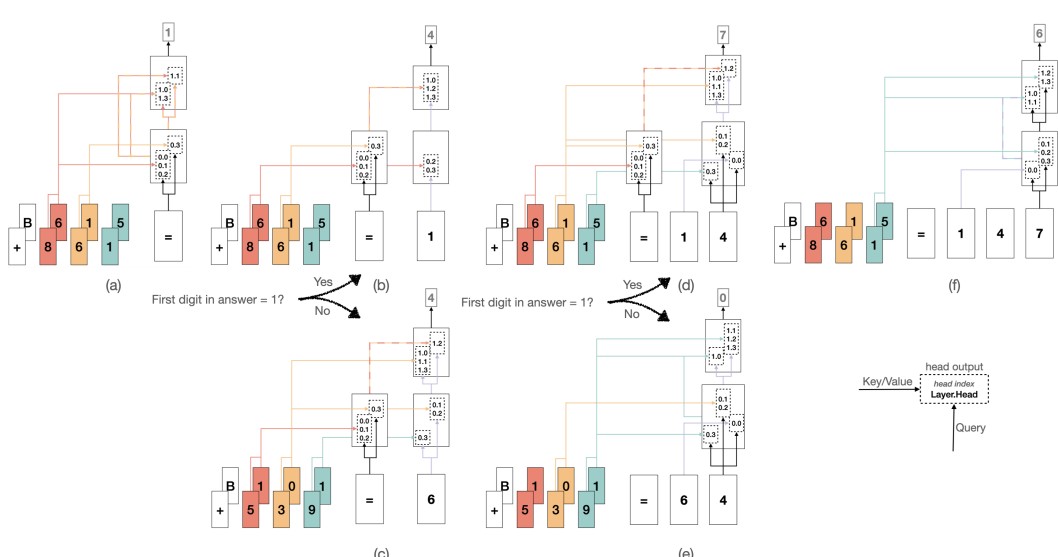

Figure 31: The information flow diagrams for predicting the digits in answer. F1 and S1 are aligned, F2 and S2 are aligned, and so forth. Color of the lines represents the information being routed, and alternating color represents a mixture of information. The computation is done from left to right (or simultaneously during training), and from bottom to top in each sub-figure. Note that the figure represents what information we find in activation, rather than the information being used by the model. Also note that the graphs are based on our qualitative examination using InversionView and attention pattern, and are an approximate representation of reality. We keep those stable paths that almost always occur. Inconsistently present paths such as routing the ones place when predicting A1 are not shown.

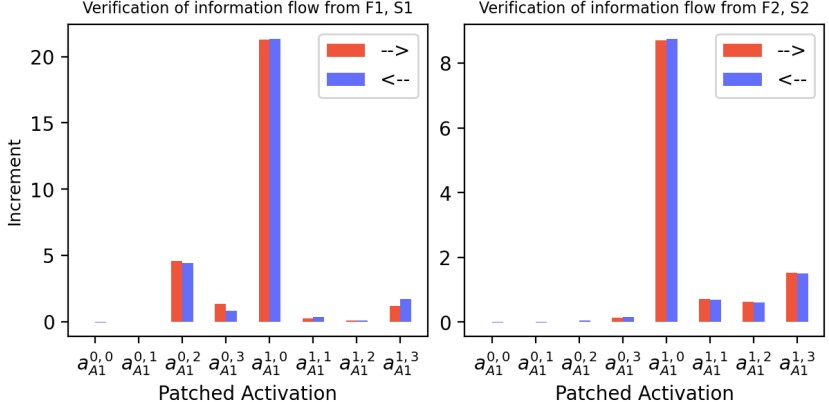

Figure 32: Causal verification results for the information flow in sub-figure (b) in Figure 31: predicting A2 when A1=1. We only consider data in $x_{orig}$ where A1=1. The constructed contrast data $x_{con}$ also satisfies this constraint. **Left**: $T_{chg} = \{F1, S1\}$. **Right**: $T_{chg} = \{F2, S2\}$. Note that the included data from $x_{orig}$ all satisfy F1+S1≥10, because, if F1+S1=9 and A1=1, no contrast example obtained by changing F2 and S2 would satisfy the constraint. The results confirm that information about the digits in hundreds and tens places is routed through the paths that we hypothesized based on InversionView in Figure 31b.

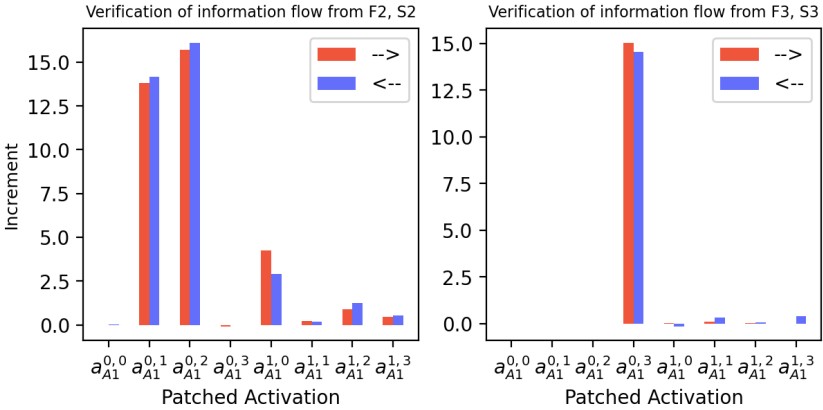

Figure 33: Causal verification results for the information flow in sub-figure (c) in Figure 31: predicting A2 when $A1 \neq 1$. We exclude those data in $x_{orig}$ where A1=1. The constructed contrast data $x_{con}$ also satisfies this constraint. **Left**: $T_{chg} = \{F2, S2\}$. **Right**: $T_{chg} = \{F3, S3\}$. We further exclude those data in $x_{orig}$ where F1+S1=9 and F2+S2=9 because we cannot find a contrast example in those cases. The results confirm that information about the digits in hundreds and tens places is routed through the paths that we hypothesized based on InversionView in Figure 31c.

because it contains information about F2 and S2, which are the same in contrast examples. On the right of Figure 6b, $T_{chg} = \{F2, S2\}$, so we are verifying the activations that plays a role in routing F2 and S2. Note that we exclude those data in $x_{orig}$ where F1+S1≥10, because changing F2 and S2 cannot change A1 in those cases.

Activation patching results for other cases are shown in Figures 32 to 36.

Overall, among all the intervention experiments and their corresponding information flow diagrams in Figure 31, the activation with the highest increment not included in the information flow diagram is $a_{A1}^{1,1}$ in Figure 32 (right), accounting for only 5.92% of the cumulative increment. In this sense, the information flow diagrams coupled with interpretations present an almost exhaustive characterization of the algorithm used by the model to predict the answer digits.

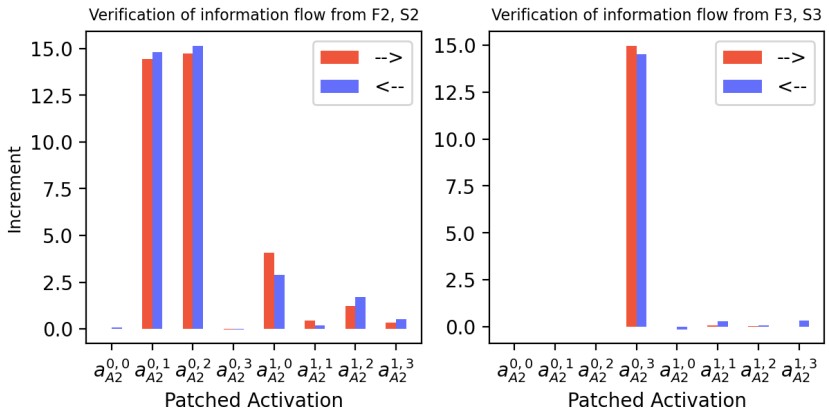

Figure 34: Causal verification results for the information flow in sub-figure (d) in Figure 31: predicting A3 when $A1 = 1$. We exclude those data in $\mathsf{x}_{orig}$ where A1$\neq$1. The constructed contrast data $\mathsf{x}_{con}$ also satisfies this constraint. **Left**: $T_{chg} = \{F2, S2\}$. **Right**: $T_{chg} = \{F3, S3\}$. We further exclude those data in $\mathsf{x}_{orig}$ where F1+S1=9 and F2+S2=9 because we cannot find a contrast example in those cases.

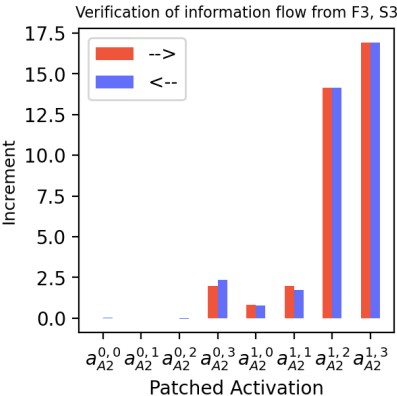

Figure 35: Causal verification results for the information flow in sub-figure (e) in Figure 31: predicting A3 when $A1 \neq 1$. $T_{chg} = \{F3, S3\}$. We exclude those data in $\mathsf{x}_{orig}$ where A1=1. The constructed contrast data $\mathsf{x}_{con}$ also satisfies this constraint.

# G    Decoder Likelihood Difference

In figures of addition task, we also show the decoder likelihood difference for each token in generated samples. It indicates what *might* be relevant to the activation. It is calculated as follows: During generation, we sample the next token from the distribution produced by the decoder model, and we record the probability of the sampled token in that distribution. We denote it as $p_{act}$. Then, we run the decoder again with the same input tokens, but this time it is fed with a "blank" activation (the activation corresponding to BOS token from the same activation site: because of the causal masking of the probed model, this activation does not contain any information). Therefore, it produces a different distribution. The probability of the same token (the token already sampled in the normal run) in the new distribution is $p_{blank}$. The difference $p_{act} - p_{blank}$ indicates if the decoder model can be more confident about a token when it receives the information from the activation. If $p_{act} - p_{blank} > 0$, the color is blue, and if $p_{act} - p_{blank} < 0$ it is red. The depth of color is proportional to the magnitude of the value. Therefore, it highlights what can be learned from the query activation, in addition to what is in the context.

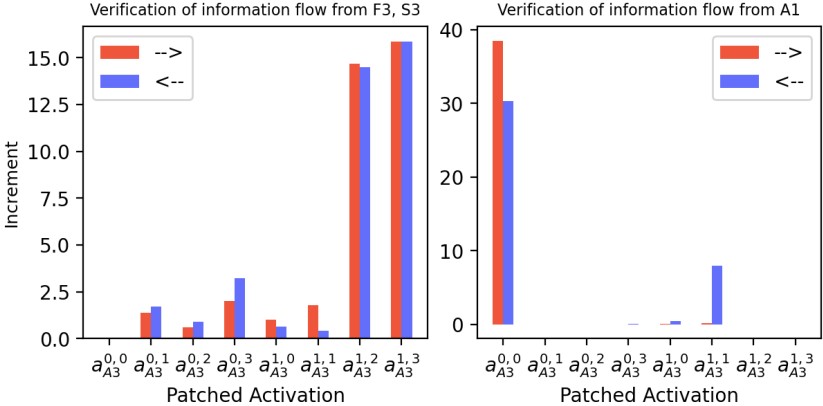

Figure 36: Causal verification results for the information flow in sub-figure (f) in Figure 31: predicting A4/E. **Left**: $T_{chg} = \{F3, S3\}$. We exclude those data in $\mathsf{x}_{orig}$ where A1$\neq$1, since in that case the prediction the target position is almost always E (end of the text). Changing F3 and S3 will not change E. Even when it does, i.e., when F1+S1=9 and F2+S2=9 and F3+S3<9, changing F3 and S3 will cause A1 to change. But we need to keep other variables the same. Based on the same reason, the contrast examples should also satisfy the constraint A1=1. **Right**: $T_{chg} = \{F1, S1\}$. We change F1 and S1 in order to change A1, thus changing A4 to E or vise versa. There is no constraint in this case, since we can always find contrast examples.

Note that the decoder has learned how to handle the activation of BOS, since it also appears in the training set because we sample uniformly at random among all tokens in the input (as in IOI and addition task).

Importantly, unlike the distance metric value, the decoder likelihood difference depends on the capability of the decoder model, and we should bear in mind that it might be inaccurate. We caution that this difference does not necessarily highlight the directly relevant part, complicating its interpretation. As shown in Figure 37, the color highlights digits in the second operand, which does not reflect the actual flow of information. In these examples the query activation corresponds to digits in answer. For example, in the left most sub-figure, $x^{0,\mathrm{pre}}$ contains the information "6 is at the position of A1", but the color does not highlight A1. This is because, on one hand, decoder predicts S1 conditioned on F1, so knowing their sum will significantly increase the confidence of S1, and S1 is highlighted. On the other hand, when predicting A1, the previous digits can already determine the answer. There is a high confidence even without knowing A1, thus it is not highlighted. In essence, the information contained in a query activation may manifest itself early, and the decoder likelihood difference does not necessarily align with the part of the input from which the information has actually been obtained.

An interesting direction for future work could be developing decoders that generate samples in a permuted order, and generate most confident tokens first, possibly based on architectures like XLNet [60].

## H    Factual Recall: Detailed Findings

### H.1    Background

We use InversionView to investigate how GPT-2 XL (1.5B parameters) performs the task of recalling factual associations. In recent research [24, 11, 20], the task has been formalized in terms of triples $(s, r, a)$, where $s, r, a$ are subject, relation, attribute respectively. The model should predict $a$ based on input containing $s$ and $r$ in natural language. E.g., given the prompt: *"LGA 775 is created by"*, the model is expected to predict *"Intel"*. In this section of the paper, we refer the last token as END, and the last subject token as SUBJ. In the above example, END is *"by"* and SUBJ is *"75"*. Geva et al. [24] found a high-level pattern that GPT-2 XL uses in solving this task: Early-middle MLP layers at SUBJ integrate information about the subject into its residual stream. Meanwhile, the relation

Activation Site: $x^{0,pre}$

Query Input

    0.000; B 3 0 4 + 3 0 9 = (6) 1 3

Generated Samples

    0.000; B 1 9 6 + 4 3 4 = (6) 3 0 E
    0.000; B 3 0 8 + 3 0 9 = (6) 1 7 E
    0.000; B 3 0 8 + 3 0 4 = (6) 1 2 E
    0.000; B 5 3 5 + 1 5 2 = (6) 8 7 E
    0.000; B 2 9 7 + 3 7 5 = (6) 7 2 E
    0.000; B 3 2 5 + 3 5 0 = (6) 7 5 E
    0.000; B 2 5 1 + 4 1 6 = (6) 6 7 E
    0.000; B 1 5 7 + 5 3 1 = (6) 8 8 E
    0.000; B 3 6 2 + 2 6 3 = (6) 2 5 E
    0.603; B 4 4 1 + 1 0 5 = (5) 4 6 E

Activation Site: $x^{0,pre}$

Query Input

    0.000; B 9 3 1 + 5 9 1 = 1 (5) 2 2

Generated Samples

    0.000; B 2 3 1 + 2 2 2 = 4 (5) 3 E
    0.000; B 7 7 1 + 7 6 2 = 1 (5) 3 3 E
    0.000; B 8 4 3 + 7 1 3 = 1 (5) 5 6 E
    0.000; B 8 9 9 + 6 9 6 = 1 (5) 9 5 E
    0.000; B 8 9 9 + 6 7 2 = 1 (5) 7 1 E
    0.000; B 2 9 7 + 2 5 5 = 5 (5) 2 E
    0.000; B 7 6 7 + 8 2 6 = 1 (5) 9 3 E
    0.000; B 8 3 3 + 7 6 3 = 1 (5) 9 6 E
    0.568; B 2 7 8 + 6 7 1 = 9 (4) 9 E
    0.852; B 3 (5) 4 + 8 2 1 = 1 1 7 5 E

Activation Site: $x^{0,pre}$

Query Input

    0.000; B 9 8 2 + 3 4 7 = 1 3 (2) 9

Generated Samples

    0.000; B 1 6 8 + 3 7 4 = 5 4 (2) E
    0.000; B 5 7 5 + 3 0 7 = 8 8 (2) E
    0.000; B 2 9 8 + 3 4 4 = 6 4 (2) E
    0.000; B 3 4 7 + 8 7 9 = 1 2 (2) 6 E
    0.000; B 7 1 2 + 1 8 0 = 8 9 (2) E
    0.000; B 3 8 9 + 5 2 3 = 9 1 (2) E
    0.000; B 3 9 7 + 4 4 5 = 8 4 (2) E
    0.000; B 7 8 2 + 8 4 1 = 1 6 (2) 3 E
    0.405; B 1 4 8 + 9 8 2 = 1 1 (3) 0 E
    0.689; B 5 4 9 + 5 2 7 = 1 0 (7) 6 E

Figure 37: The $\epsilon$-preimage of $x^{0,\mathrm{pre}}$. Here the information contained is A1, A2, A3 respectively, while the decoder likelihood difference highlights digits in operands. Because given the first operand and the final sum, the digits in second operand can be inferred.

information is incorporated into END's residual stream through early attention layers. In upper layers, the correct attribute is moved from SUBJ to END's residual stream by attention layers.

## H.2 Selecting Activation Sites

As we mentioned earlier, our goal here is not to provide a full interpretation of the computations performed to solve this task; rather, it is to test whether InversionView can successfully produce interpretable results on larger models. As the factual recall task is complex and involves many model components [61, 11], we decide to focus on 25 attention heads in the upper part of the model (layer 24-47) that contribute most frequently to the final prediction. We select these activation sites because the attribute retrieval tends to happen there [24] and we expect more abstract information in $\epsilon$-preimage. Here, we note that preliminary experiments revealed that it is necessary to restrict the number of activation sites that the decoder is trained for, given a limited compute budget, as more activation sites require the decoder to learn more a complex overall inverse mapping from activations to inputs. Scaling the approach to apply to many activation sites simultaneously is left to future work.

Concretely, we use the attribution method introduced in [18] to estimate the head importance at END position. Formally, given $y = z_1 + \cdots z_m$, the importance of each term $z_j$ to the sum $y$ is estimated as follows:

$$\text{importance}(z_j, y) = \frac{\text{proximity}(z_j, y)}{\sum_k \text{proximity}(z_k, y)}, \tag{4}$$

$$\text{proximity}(z_j, y) = \max(-\|z_j - y\|_1 + \|y\|_1, 0). \tag{5}$$

Regarding the intuition and estimation quality of this method, please refer to [18]. In our case, $x_{\text{END}}^{i,mid} = x_{\text{END}}^{i,pre} + \sum_j a_{\text{END}}^{i,j}$, we calculate importance$(a_{\text{END}}^{i,j}, x_{\text{END}}^{i,mid})$ on each example on a subset of COUNTERFACT [37]. The subset contains around 1,000 factual prompts known by GPT-2 XL (we used the same subset as described in [37] Appendix B.1). We consider an attention head activated on a certain input prompt if its importance is higher than the threshold of 0.02, and calculate the frequency of being activated over the subset. Finally we select top 25 most frequent heads in model's upper layers.[6]

There are multiple methods that can be used to find important attention heads [38, 39, 50, 29], because we do not have strict requirements for finding the most important heads, we choose this one because of its simplicity.

---

[6] $h^{31,0}$, $h^{33,0}$, $h^{38,22}$, $h^{29,9}$, $h^{37,7}$, $h^{32,12}$, $h^{31,8}$, $h^{24,24}$, $h^{28,3}$, $h^{25,7}$, $h^{28,21}$, $h^{27,16}$, $h^{42,24}$, $h^{31,4}$, $h^{34,20}$, $h^{30,23}$, $h^{24,8}$, $h^{30,8}$, $h^{25,10}$, $h^{33,9}$, $h^{32,15}$, $h^{30,1}$, $h^{29,20}$, $h^{36,17}$, $h^{35,19}$, ordered by frequency, last one is the most frequent

## H.3  Decoder Training

To train the decoder model, we collect text from 3 datasets, including the factual statements from COUNTERFACT [37][7] and BEAR [56] [8], as well as general text from *MiniPile* [30]. The factual statements we used are complete sentences containing the final attribute. Note that in COUNTERFACT, for each attribute, there are always multiple different subjects associated with it. So if the information contained is solely the attribute, a well-trained decoder should generate different subjects. Regarding *MiniPile*, we randomly select 10% of it, and split the text into sentences [9] and remove sentences longer than 100 characters.

Importantly, for each head we selected, we extract a separate subset of sentences from the collected data on which the head is "activated". By being "activated" we mean the attention weight on BOS token is less than $0.6$[9]. We find this is important based on preliminary experiments, because in many cases attention heads in higher layers execute "no-op" by exclusively attending to BOS, resulting in attention output containing no information. Training decoder on these activations discourages it to learn to read information from the query activation. So each head correspond to a subset of text, on which the head output will be captured to create training data. The number of sentences in subsets ranges from 0.6 to 2.6 million.

The training data for decoder consists of two parts, each of which accounts for 50%. In one part, the activations are taken from GPT-2 XL when processing factual statements from COUNTERFACT and BEAR, and activations correspond to the END position (the sentence structure is provided in these datasets). In the other part, the activations correspond to text from all 3 datasets (thus mainly composed of *MiniPile* text), and correspond to the position with least attention weight on BOS token. By doing so, we emphasize the importance of factual statements domain while covering a large variety of text.

We use GPT-2 Medium as the backbone model for decoder. Concretely, the newly added components in the decoder architecture (e.g. those parts responsible for processing query activation as shown in Figure 12) are trained from scratch, but the transformer layers in the decoder are equipped with pretrained weights. In this way, we also make use of the existing capacity of language models. Note that in other case studies, we use a small 2 layer transformer as decoder. The reason why we use a much larger decoder for this task is we expect a much more complex inverse mapping to be learned. Specifically, we expect $\epsilon$-preimage for some activation sites to contain multiple different subjects sharing a certain attribute, the decoder needs to generate these subjects given the attribute. So it needs enough capacity to memorize the knowledge.

Similar as before, we always add the "$< |\text{endoftext}| >$" token as the BOS token when capturing query activation from GPT-2 XL. We again use a new token "[EOS]" as the EOS token when training the decoder. We sample query activation with equal probability of activation sites. We sample 32 million training examples (all activation sites are included) and train the decoder with batch size of 512, resulting in roughly 60K steps. The final average in-preimage rate is 36.0%.

## H.4  Sampling from Decoder

Similar as before, when using decoder to generate samples, we perturb the query activation before feeding it to the decoder in order to cover the neighbourhood around the query. In this larger model with high-dimensional activations, we found it useful to craft this process more carefully and use different disturbance. In preliminary experiments, we found that cosine similarity usually produces more interpretable $\epsilon$-preimages, therefore, we randomly sample activations that have a certain angle $\theta$ to the query activations, and then randomly scale it so that its magnitude ranges from $e^{-1} \cdot \|\mathbf{z}^q\|_2$ to $e \cdot \|\mathbf{z}^q\|_2$. We repeat this process for different $\theta$ values with in the range $\cos\theta \in [0.75, 0.9]$. To further encourage diversity, we lower the probability of tokens that have already appeared too many times in the generated samples.[10]

---

[7]We make use of all prompts of each data point. Besides the *"prompt"*, we also include *"paraphrase_prompts"*, *"neighborhood_prompts", and "attribute_prompts"* concatenated with their corresponding answer.

[8]We use BEAR$_{\text{big}}$

[9]we do not experiment with other values

[10]We chose a simple and heuristic approach, without any attempt at optimizing it: Any token that has appeared $> 5$ times in the same position, while appearing less than 50K times in the miniPile training set, has its probability set to zero at this position.

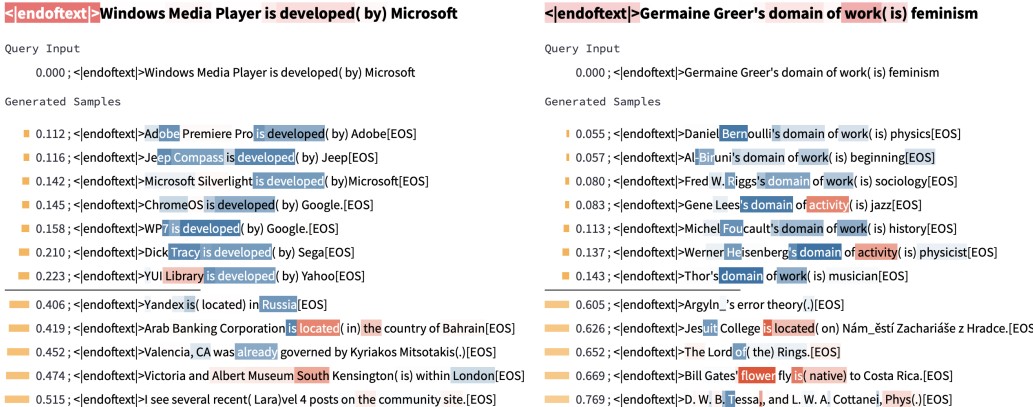

Figure 38: $\epsilon$-preimage for head output $a^{24,24}$, showing encoded information is the relation. On the left, the relation "developed by" appears throughout the $\epsilon$-preimage. On the right, the relation "domain of work is" is consistent $\epsilon$-preimage.

We use the first 200 examples of the aforementioned subset of COUNTERFACT (the subset containing examples known by GPT-2 XL) as query inputs and generate samples. The results are also available in our web application.

## H.5 Observation

**Choice of Threshold.** We use the same distance metric as in IOI task, and we set the threshold $\epsilon = 0.25$. As we mentioned earlier, larger threshold produces more coarse-grained information. Because we observe that the trained decoder in many cases does not generate enough and diverse samples within a close distance (e.g., 0.1), we increase the threshold in order to draw more reliable conclusions.

A priori, the sparsity of sampled $\epsilon$-preimages at smaller thresholds may reflect multiple possibilities. One is that there indeed are no close neighbors for the query input. For example, if the subject's text is copied, then only samples containing the same subject or the same token will lie in $\epsilon$-preimage. Such a phenomenon may generally reflect the high-dimensional geometry of larger models. Another one is that the decoder is not complete; and that it would either require more training or more capacity, or more training data.[11] Training larger decoders on more data would mitigate this problem, which can be done in future work. Nevertheless, in some cases, we do observe a substantial number of diverse samples even within $\epsilon = 0.1$, allowing us to infer information at a higher level of granularity in these cases.

**Some heads have fixed behavior** We observe that some heads' outputs almost always contain only relation information, if they contain any information at all (being "activated").[12] In Figure 38, we show two examples for one of these heads. We can see that samples in $\epsilon$-preimage share the same relation. The results are consistent across query inputs. We can infer that the these heads move the relation information to END's residual stream.

On the other hand, some other heads almost always move information about the subject, when they are "activated".[13]. Figure 39 shows one of these heads. We can see that the information is only about the subject – so the function of these heads can be summarized as moving information about subject. Interestingly, while a head can move certain attribute about the subject (e.g., nationality, or profession, etc), the attributes it moves for different subjects are diverse. For instance, while a head might, on a certain input, move information that the subject plays certain kind of sports, it may, on another input, move information that the subject is an electronic product. One head, 31.0 tends to usually show

---

[11]For example, if the query activation contains information "related to Amazon and related to audio", there are only a few satisfactory subjects in the dataset ("Audible.com", "Amazon Music").

[12]$h^{24,24}$, $h^{25,7}$, $h^{27,16}$, $h^{28,21}$, $h^{29,20}$, $h^{33,9}$

[13]$h^{29,9}$, $h^{30.8}$, $h^{32,12}$, $h^{33,0}$, $h^{37.7}$

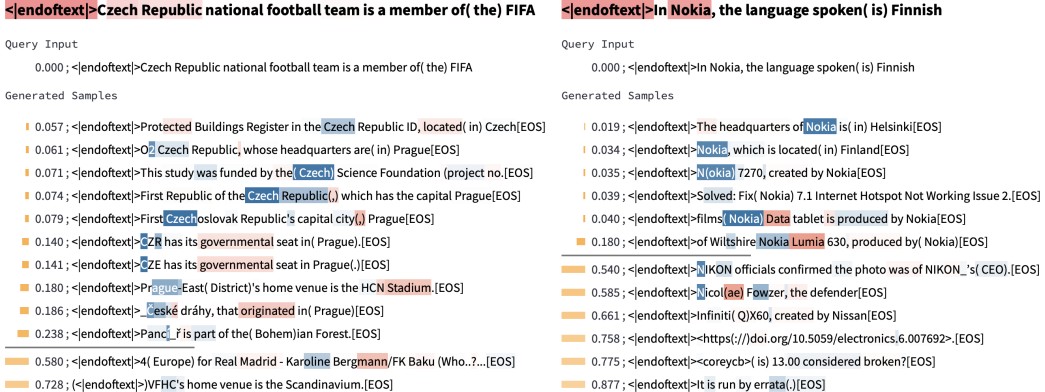

Figure 39: $\epsilon$-preimage for head output $a^{29,9}$, which contains information about the subject. On the left, Czech-related words are common throughout the $\epsilon$-preimage, e.g., "Czech", "Prague", "Bohem...", so the information contained is the subject is "Czech-related". On the right, "Nokia" is shared in $\epsilon$-preimage, so the information is simply "Nokia".

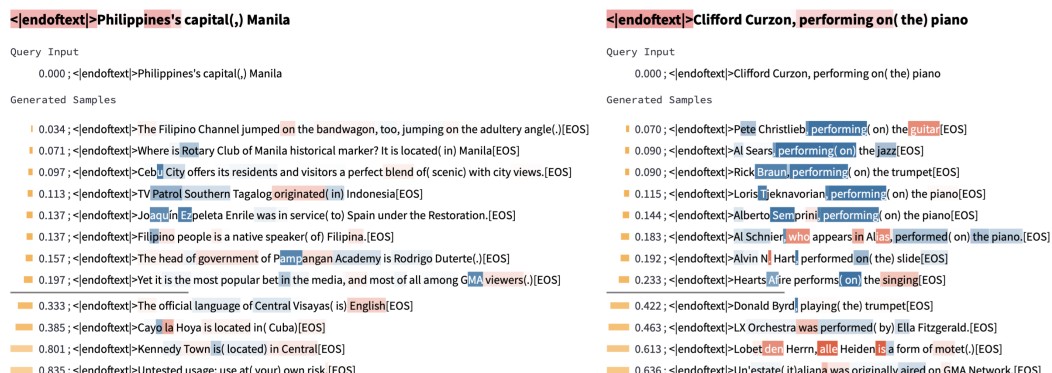

Figure 40: $\epsilon$-preimage for head output $a^{35,19}$, which exhibit different information on different inputs. On the left, Philippines-related words are common throughout the $\epsilon$-preimage. For example, "Cebu": a city in Philippines, "TV Patrol Southern Tagalog": a TV program in Philippines, "Enrile": A municipality of Philippines and a surname of many famous Filipinos, "GMA": a Philippine television channel / broadcasting company. So the information contained is the subject is "Philippines-related". Note that because the samples are generated, *the fact stated in sample is not true in many cases*. On the right, the relation "performs on" is encoded.

nationality or language information about the subject, but usually the heads we studied show no clear preference for a type of attribute. In addition, these is no obvious correlation between the subject attributes encoded by such heads and the attribute queried by the relation.

**Other heads exhibit a mix of behaviors**  The other heads among the 25 heads we inspect, when "activated", move information about subject or relation (in some cases, both). Which behavior is exhibited varies between inputs. Figure 40 shows one of these heads. On the left of the figure, the head moves information about subject, while on the right it moves information about the relation.

**Relation-agnostic retrieval**  In the factual recall task, only one specific attribute is sought. Geva et al. [24] found evidence that the model "queries" the residual stream at SUBJ for the specific attribute asked by the relation part of the prompt, and the representation at END can be viewed as such a relation query. In other words, the attribute extracted from the subject representation depends on the relation. However, we do not observe reliable evidence for this phenomenon among the 25 heads we inspect. In the figures we have shown so far, we can see the commonality shared between samples in $\epsilon$-preimage is not the attribute requested in the prompt. This still holds if one reduces the

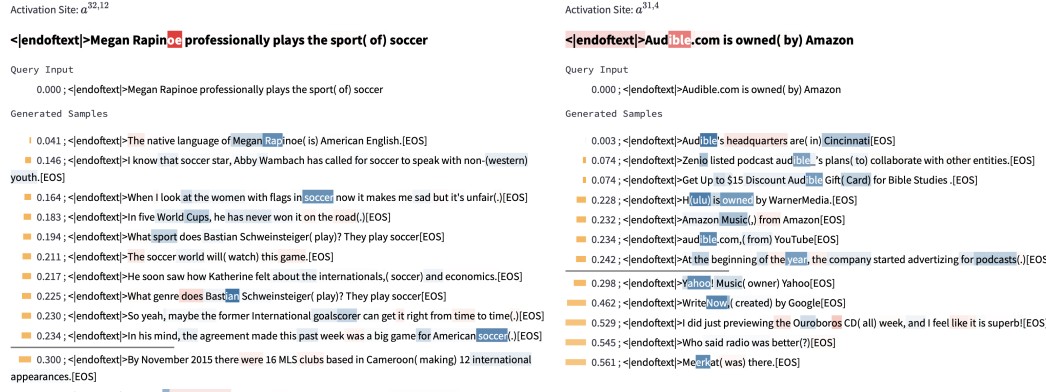

Figure 41: Examples showing relation-agnostic retrieval. On the left, the information encoded is "soccer", which is indeed the requested attribute. However, the first sample shows this is not dependent on the relation, since the "soccer" is still retrieved when relation is "speaks language". On the right, the information "audio-related" is encoded, while the relation in the query input is "owned by".

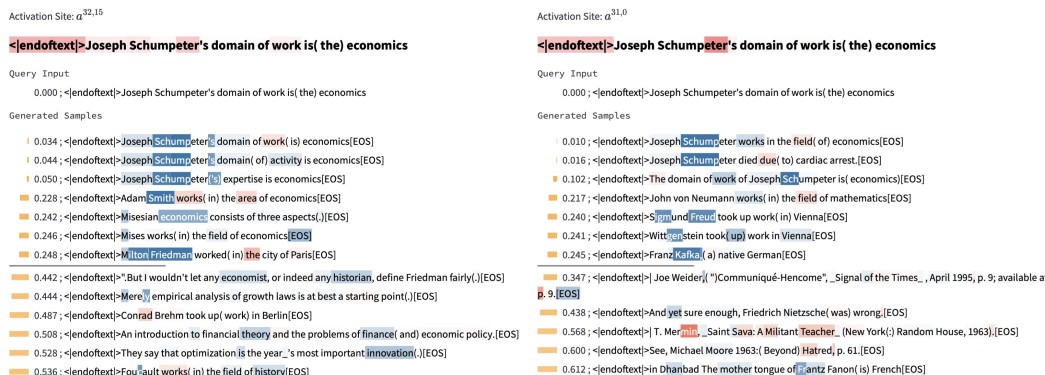

Figure 42: Examples showing different attributes of the same subject are extracted by different heads. In the query input, "Joseph Schumpeter" is an Austrian political economist. On the left, the information encoded is "economist". On the right, the information is about language/nationality (areas around Austria). Again, we emphasize that the facts stated in the sample are not necessarily true.

threshold. Figure 41 shows two more examples showing this characteristic. In general, we find that the information moved by the 25 attention heads tends to be the most important attribute or the "main" attribute about the subject. On the left of Figure 41, the most important attribute coincides with the requested attribute. We know this is a coincidence because the closest sample in $\epsilon$-preimage has a different relation. Note that other attributes about the subject could be moved by heads that we have not studied.

In addition, while heads that attend to subject usually move the most important attribute, we do observe sometimes different attributes are moved by different heads. In Figure 42, we show that information about profession and information about language/nationality of the subject are extracted by two heads. In fact, we observe that head $h^{31,0}$ tends to extract language/nationality information in general. But we do not find other obvious pattern of attribute category extracted by other heads.

Our findings echo those from [11], who argue that the primary mechanism for the factual recall task is additive. With our running example *"LGA 775 is created by"*, simply speaking, some heads promote attributes associated with the subject (*chip*, *hardware*, *Intel*, etc.), some heads promote attributes associated with the relation (*Apple*, *Nintendo*, *Intel*, etc). When the results from these independent mechanisms are added together, the intersection (*Intel*) will stand out. Therefore, the model can solve the task by adding two simple circuits, while humans find Q-composition [17] (i.e., relation information is used as queries in attention heads) more intuitive. From another perspective, this

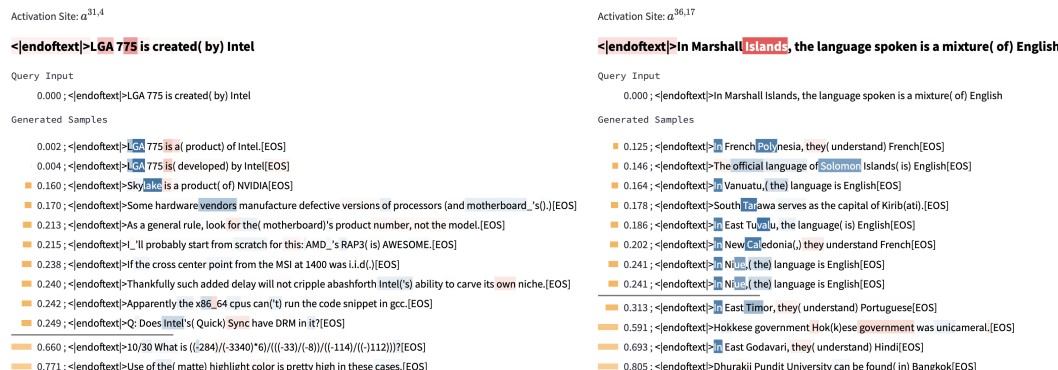

Figure 43: $\epsilon$-preimage showing information about the subject moved by the attention head. On the left, the information is "cpu/computer-hardware-related". On the right, the information is "island country". Note that some statements are not correct.

mechanism implies vector arithmetic. Instead of vector addition in vocabulary space, we can think of it as first summing two vectors (e.g., the output of subject heads and the output of relation heads) and then projecting them to vocabulary space.

The evidence found by Geva et al. [24] can also be explained by this mechanism. They use DLA to inspect attention layers' updates to END's residual stream, and find that the token most strongly promoted by each update usually matches the attribute predicted at the final layer. In other words, after projecting attention layer's output to vocabulary space, the top-1 token is usually the exact requested attribute. Because their experiments study attention layer's output as a whole, instead of individual heads, an alternative explanation is the additive mechanism mentioned above. Importantly, because we only inspect 25 heads, other mechanisms including Q-composition are also possible. Moreover, different mechanism can exist in other models, readers should not draw strong conclusions.

### H.6  More Examples of InversionView

See Figure 43, Figure 44. Interestingly, when inspecting the $\epsilon$-preimage shown on the right of Figure 44, we also find an example showing InversionView can detect a flaw of the model. Our interpretation for this activation is "Canada-related", and we can see "York University" (which is in Canada) inside the preimage, while "University of York" (a different university, located in England) is outside of the preimage. However, we find that the "U of York" is *inside* the preimage.

Checking the model's prediction about "U of York", we find that *the model indeed believes "U of York" is in Canada*. More specifically, given the prompt "U of York is located in", the top-12 predictions for the next token are: the, Toronto, York, downtown, a, one, New, central, Canada, North, Scarborough, London. On the contrary, with the prompt "University of York is located in" the model's top-12 predictions are: the, York, North, East, central, Yorkshire, north, a, northeast, London, England, northwest.

## I  Notes on Attention, Path Patching, DLA and others

### I.1  InversionView Reveals Information from Un-attended Tokens

In Section 4, we mention that attention pattern is not sufficient to form hypothesis when the model has more than one layer. Because unlike in layer 0 each residual stream contains information only about the corresponding token, in higher layers each residual stream contains a mixture of tokens from the context, making it difficult to determine what information is routed by attention. Besides this point, we also find that sometimes attention pattern can be misleading even in layer 0.

InversionView reveals how components can know more than what they attend to. At the top of Figure 45, we show the attention weights of head $h^{0,3}$. Here, "=" attends almost solely to S2, so the head output $a^{0,3}$ should only contain information that there is an "8" in tens place. The generated $\epsilon$-preimage, however, shows that it contains information about F2: The number in tens place other

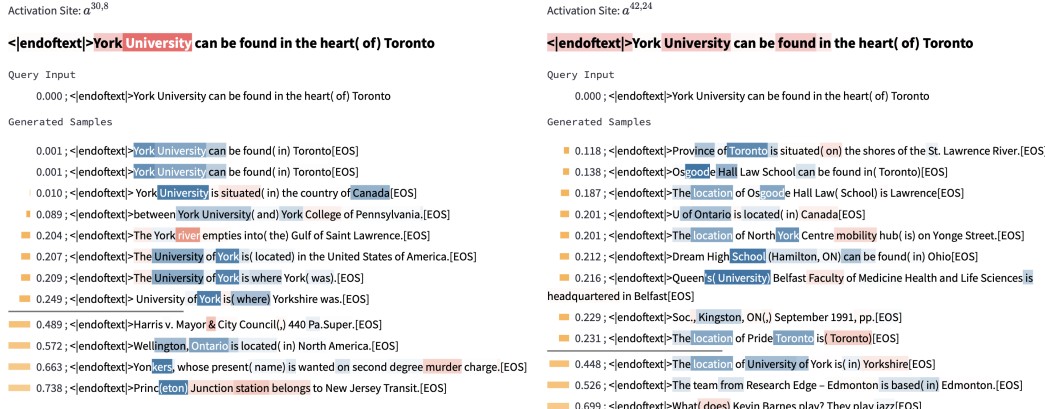

Figure 44: $\epsilon$-preimage showing different information of subject is moved by different head. On the left, the information is superficial text content "York" and/or "University". Samples containing these words are in $\epsilon$-preimage, such as "University of York", even though it is a different university located in England. This can be confirmed by the fact that "Wellington, Ontario" is far away. On the right, the information is "Canada-related", which is more high-level. We can also see "University of York" is outside of the preimage.

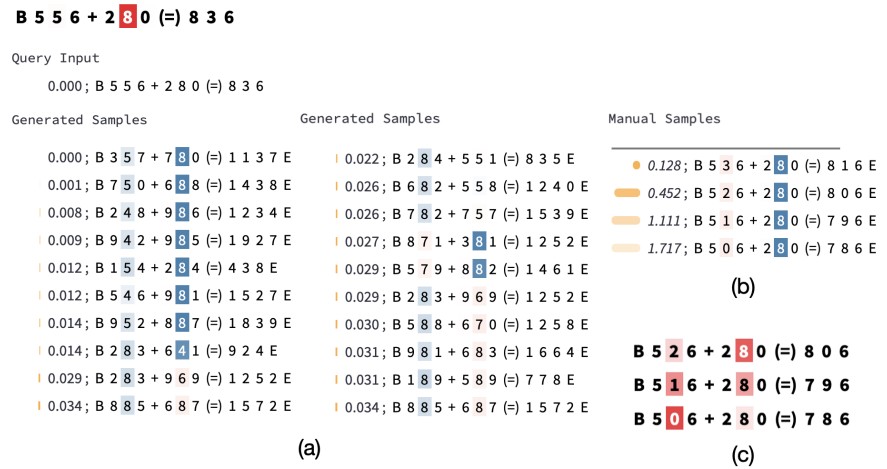

Figure 45: For activation site $a^{0,3}$, InversionView reveals how activations can encode information without an attention edge: (a) Even though, on this input, $h^{0,3}$ attends only to one tens place digit, it also encodes the approximate identity (range 4–8) of another tens place digit. It encodes that the sum of tens places is greater than ten. (b) We verify our hypothesis by manually create some samples and calculate $D$. (c) Attention patterns for manually created inputs outside of the $\epsilon$-preimage. The attention pattern differs from that of the query input. In the query input, the attention head infers information about the second tens place digit from the *absence* of an attention edge.

than "8" is always in a certain range ($\geq 4$), resulting in a carry to the hundreds place. To verify this, we manually constructed examples (rightmost column in Figure 45) where the other number is outside of the range, and found that, for these, the activation distance is indeed very large, confirming the information suggested by the decoder. In layer 0, how does the model obtain information about a token without attending it? At the bottom of Figure 45, we show the attention weights of those manually inserted examples. So the answer is: a different attention pattern would arise if F2 is not in that range. Information can be passed not only by values, but also by queries and keys. InversionView successfully shows this hidden information, even without comparing across different examples.

## I.2 Additional Discussion about Path Patching

Besides our argument in Section 4, another important aspect of circuit discovery methods is that, in many tasks (including our character counting task), the computational nodes do not correspond to fixed positions, and directly applying path patching is problematic. It's not really clear how to apply path patching when varying input positions matter, as the literature on circuit discovery defines circuits in terms of components, not in terms of input positions. In the case of Character Counting Task, such an interpretation would just define a circuit linking the embeddings, attention heads, and MLPs, without capturing the role of different positions, and the fact that characters from varying positions feed into the computation. Such a view would not provide any nontrivial information about the mechanics of the implemented algorithm. This reflects a more general conceptual challenge of circuit discovery: When different input positions are relevant on different inputs, as in the Character Counting Task, one could either define a single circuit across inputs in which every input position is connected to a single node that performs a potentially complex computation, or define per-input circuits where the wiring is input-dependent; however, per-sample path patching is not very scalable, and resulting per-input circuits would require further interpretation to understand how they are formed across inputs.

## I.3 Additional Discussion about DLA

Direct Logit Attribution (DLA) extends the logit lens method to study individual model components. Specifically, it projects the output of a model component (thus an update on the residual stream) into vocabulary space, and interpret the component by inspecting the tokens it promotes or suppresses. This method has gained popularity in recent years [23, 54, 16, 19], especially in the research of interpreting factual recall in language models [24, 62, 11, 61]. However, in this section, we argue that DLA is only suitable for studying model components that *directly* affect the model's final output, and is not well-suited for components whose effect is mediated by other components. In the circuits found by path patching [54, 13], we can see many components that do not connect to the final output directly, which suggests a substantial part of their effect on the final predictions is mediated by further components. DLA shows their direct effect, which may even be non-causal "side effects". Intuitively speaking, some information is meant to be read by a downstream component, e.g., the S-Inhibition Heads' output in IOI circuit is meant to be read by the query matrix of Name Mover Heads, and such information may not necessarily be visible when projecting to the vocabulary space. Dao et al. [15] also point out such limitations of DLA.

In Table 4 we show the results of applying DLA to attention heads' output in IOI circuit. We can see that the expected information is not visible in most cases. The best result comes from the S-Inhibition head 8.10, with only 7.5% of cases where the expected name is in the top-30 tokens and there is no other name being suppressed more than it. The rare cases where the expected name is visible can also be explained by the small direct effect on the final output as depicted by Figure 3(b) in [54].

Therefore, researchers should be cautious when using DLA and should be aware of its limitations. A good usage example is the IOI circuit [54], where the authors first identify those attention heads directly affecting the final logits, and only apply DLA to them and design other experiments to interpret other components. Importantly, in the context of factual recall task, we find that the information given by InversionView about the upper layer attention heads is often visible via DLA, indicating these heads contribute to model's output directly. Thus, our results can serve as confirmation that prior results relying on DLA in this task are generally reliable.

## I.4 Methods Generating in Input Space

Feature visualization [43, 41] generates inputs maximally activating a certain neural network unit, and interprets an individual neural network unit (e.g., neurons) to understand its general role across inputs, while InversionView interprets specific values of inner representations, by finding inputs that result in the same vector. When the input changes, the value and thus the interpretation may change. Adversarial or counterfactual example generation methods [27, 59, 46, 45] generate input that is similar to the original input but results in different outcome. Some of them are also used to explain the model. While similar in input space, the adversarial/counterfactual input is likely to be quite different in internal representation space, leading to a different output. In contrast, we are interested in how different inputs in input space are represented very similarly in internal representation space.

| Attn Head, position | Info by InversionView (See Table 2) | Top 30 promoted rate | Top 30 promoted & 1st name rate | Top 30 suppressed rate | Top 30 suppressed & 1st name rate |
|---|---|---|---|---|---|
| 9.9, END | IO name | 99.5% | 98.8% | 0% | 0% |
| 7.9, END | S name; Position of ... | 0% | 0% | 3.2% | 3.2% |
| 8.10, END | S name; Position of ... (most of time) | 0% | 0% | 27.4% | 7.5% |
| 0.1, S2 | S name | 0% | 0% | 0% | 0% |
| 0.10, S2 | S name | 0% | 0% | 0% | 0% |
| 2.2, S1+1 | S name (most of time) | 0% | 0% | 0% | 0% |
| 4.11, S1+1 | S name; Position of ... | 0% | 0% | 0% | 0% |

Table 4: Applying DLA to the heads in IOI circuit. Except the first row, all heads do not directly connect to final output according to the IOI circuit, the results show DLA cannot decode their information. We do not include those heads in which only position information is encoded. "Top 30 promoted (suppressed) rate" means the fraction of input examples where the expected name (IO name for the first row, S name for other rows) is inside the top 30 tokens promoted (suppressed) by the head's output. "Top 30 promoted (suppressed) & 1st name rate" means the expected name is not only inside the top 30 promoted (suppressed) tokens, but also the most promoted (suppressed) name among a list of common and single-token names, so it does not count when another name is ranked higher. Note that a name can be associated with two tokens (with and without a space before it), when calculating the rate, either of them satisfying the condition will count. The rate is calculated over 1000 random IOI examples. As we can see, except for the first row, the expected name is not observable most of the time.

Similar to InversionView, GAN inversion methods [58] also study the mapping between input space and representation space, with a focus on interpreting the semantics of GAN's latent space and manipulating generation.

## J   Automated Interpretability

We further explore whether the process of obtaining the common information from a collection of inputs can be automated by LLMs. We use Claude 3[14] [3] In preliminary experiments we also try GPT-4 [1] but we find Claude 3 works better in our case. In the prompt given to Claude 3, we first describe the task it needs to perform, the terminology we are using (e.g., F1, F2, etc.), the rules (e.g., the pattern it finds should be applicable to each inputs in the $\epsilon$-preimage), input and output form, and the crucial steps it should follow. In addition, we also provide it with 3 demonstrating examples in conjunction with the correct answers. Each example corresponds to a specific activation site and token position (e.g., $a^{0,0}$, A1). In each example, there are 2-3 specific query inputs, each query input is accompanied with 20 (sometimes less) samples that are *inside* the $\epsilon$-preimage. Claude 3 needs to find the pattern for each query input, and summarize its findings across several query inputs, which is the information contained in general in that activation. In addition to the content described above (the common part shared between prompts), we give it a questioning example, which is the content we would like it to interpret. The questioning example shares the same form as the demonstrating example, except that it contains 5 query inputs and their corresponding samples. In addition, when two separate interpretations are needed based on different A1 value, we run the generation twice with examples of different A1 value, instead of giving the model a mixture of two cases and resorting to its own capacity.

We think the following findings from our experiments are worth mentioning: 1) It's hard for the model to align digits of the same place (e.g., comparing all F1 digits), because the samples are presented as a single flattened string instead of a 2-dimensional table. We find that explicitly adding the variable name can largely mitigate this problem, they may serve as certain kind of keys. For example, "7(F1)

[14]Version: claude-3-opus-20240229

1(F2) 1(F3) + 9(S1) 9(S2) 4(S3)". 2) The generated interpretation is sometimes not consistent. The model may generate different conclusion even with the same prompt, but this usually only happens to less important information. 3) The model does not strictly follow the rule, i.e., the common pattern should match *all* inputs, even though we state this repeatedly in different ways in the prompt. The model will say "always" even when there is a counterexample. We should keep 2) and 3) in mind when reading the results.

We run the generation for each entry in Table 3 once, using the samples generated from the corresponding activation. The results are shown in Table 5, accompanied with human interpretation for comparison. The interpretation given by Claude 3 reflects the main information in almost all cases. Even when the information becomes more complex in layer 1, the interpretation quality does not significantly decline. This implies that automated interpretation by LLM is promising. On the other hand, we can also see there are some problems: 1) Some of the model's claims are spurious, these claims are usually ranked low by the model, indicating they are not very obvious. 2) The model sometimes does not explain in a desirable manner. For example, for the entry "$x^{0,\mathrm{mid}}$, A3", the information includes A2 and whether A1=1, which means when A1=1, the sum of F1 and S1 is known. Thus Claude 3 concludes that (F1, S1) is approximately represented. Though this is somewhat true, it is not a fundamental piece of information, and there is a more concise summary. 3) The interpretation tends to be verbose, even though we ask the model to be succinct and provide it with some examples. Despite the problems, we think the automatically generated interpretation is by and large satisfactory and informative, and we think some of the problems can probably be solved or mitigated by engineering better prompts.

| Start of Table 5 | | |
|---|---|---|
| Entry in Table 3 | Interpretation from Table 3 | Interpretation by Claude 3 |
| $a^{0,0}$, A1 | 1-2 digits from F1 and S1 | *used as example in the prompt* |
| $a^{0,1}$, A1 | 1-2 digits from F1 and S1 | F1 and S1 (sometimes exactly and sometimes approximately. Their order can sometimes vary) |
| $a^{0,2}$, A1 | 1-2 digits from F1 and S1 | the digits in the hundreds place (F1 and S1): - They tend to be large single digits (7, 8, 9) that add up to 10 or more - Sometimes one or both are fixed to a certain single digit value - Sometimes their values can swap positions between F1 and S1 There are no strong consistent patterns observed for the digits in the tens or ones places (F2, S2, F3, S3). No other types of patterns (e.g. on A1 or carries) are consistently observed. |
| $a^{0,3}$, A1 | 1-2 digits from F2 and S2; C2 | F2 and S2 (fuzzy, interchangeable); F3 (sometimes fuzzy); S2 (sometimes fuzzy) |
| $x^{0,\mathrm{mid}}$, A1 | F1 and S1; C2 | 1) F1 and S1 are either the same digit, or a pair of digits that always appear together in the hundreds place (sometimes with order swapped). 2) Whether F2+S2 produces a carry (the value of C2) is often consistent in each example, either always 0 or always 1. 3) There is no consistent pattern found for the ones digits F3 and S3. |
| $x^{0,\mathrm{post}}$, A1 | F1 and S1; C2 | 1) F1 and S1, either exactly or an approximate version, with order sometimes varying 2) S2, usually 3) C2, occasionally F3, S3 and other parts of the addition rarely contain any mutual information between the query and generated strings. |
| $a^{1,0}$, A1 | Fuzzy info about F1, S1 and C2; Fuzzy info about F3 and S3 (sometimes) | 1) F1 and S1 digits (sometimes exact match, sometimes approximate match). 2) Whether the sum of F2 and S2 is greater than or equal to 10 (i.e. the value of C2). 3) F3 and/or S3 digits are often preserved from the query to the generated strings. |

| Entry in Table 3 | Interpretation from Table 3 | Interpretation by Claude 3 |
|---|---|---|
| $a^{1,1}$, A1 | **If** A1=1 (likely to be): 1-2 digit from F1 and S1 (sometimes their sum); C2; **Else**: Fuzzy info, including some info about F3 and S3 (sometimes) | **If** A1=1: 1. F1 and S1 digits from the query string appear in the generated strings, but their order may be swapped. 2. Occasionally some other information is shared, such as the carry digit C2, or a specific digit like F3, but this is not consistent across all examples. In summary, the key common information is the two digits that appear in the hundreds place of the query and generated strings, while their specific positions as F1 or S1 may vary. **Else**: The amount of information shared between the query and generated strings varies by example. In some cases, there are clear patterns like certain digits being consistently shared (e.g. S1, S3). Other digits may share approximate rather than exact values (e.g. F1). Some digits tend to vary more (e.g. F2, S2). Occasionally, the carry C2 is also shared. When there are too few generated strings, it is hard to conclusively identify patterns. More generated strings allow more confidence in the patterns found. |
| $a^{1,2}$, A1 | Fuzzy info about F2 and S2 (sometimes) | 1. F1 and S1 (sometimes fuzzy representation, their order can vary) 2. F2 and/or S2 (sometimes only one of them, sometimes partially match) 3. Occasionally F3 and/or S3 (partially match, their order can vary) |
| $a^{1,3}$, A1 | 1-2 digits from F1 and S1 (sometimes fuzzy) | 1) F1 and S1 values, either exactly or approximately. Their order can sometimes be swapped. 2) To a lesser extent, F2 and S2 values, either exactly or approximately. Patterns for F3/S3 are rarely seen. Other aspects like C2 are occasionally but not consistently shared. The level of correspondence varies a lot between examples. Examples with many missing values are hard to find patterns in. |
| $x^{1,\mathrm{mid}}$, A1 | F1 and S1; C2 | 1) The values of F1 and S1, either the exact values or their sum 2) The value or presence/absence of S2 3) The relationship between F2 and S2 (which is greater, or whether there is a carry) 4) Which digits are missing in the result (A1 or A2) |
| $x^{1,\mathrm{post}}$, A1 | F1 and S1; C2 | the F1 and S1 values from the query can be found in either order (F1, S1) or (S1, F1) in all generated strings for that query. In some cases, C2 (carry from tens place) is also consistent between query and generated strings, being either 0 or 1 in all strings for that query. F3 and S3 values do not seem to have any limitations or correspondence between query and generated strings. |
| $a^{0,0}$, A2 | whether A1=1 | - Sometimes A1 is consistent (always the same value) between query and generated strings - Sometimes the exact or approximate F3 and S3 values from the query appear together frequently in the generated strings - In some examples, there is no clear common information The values of (F1, S1) and (F2, S2) do not seem to be consistently shared between query and generated strings in these examples. |
| $a^{0,1}$, A2 | **If** A1=1: Almost no info **Else**: 1-2 digits from F2 and S2 | **If** A1=1: F2 (sometimes fuzzy); A1 **Else**: 1) The exact values or range of values for F2 and/or S2 2) The value of C2 (whether F2+S2≥10) 3) Occasionally F1/S1 or F3/S3, but not as common as F2/S2 4) A1 is usually not shared The most commonly shared information is the exact values or ranges for F2 and S2, as well as the resulting C2. F1/S1 and F3/S3 are less commonly shared. |
| $a^{0,2}$, A2 | **If** A=1: both F1 and S1 **Else**: 1-2 digits from F2 and S2 | **If** A=1: 1. F1 and S1 from the query string always appear in generated strings in the hundreds place, but their order may be swapped. 2. A1 is always the same in query and generated strings. **Else**: 1) F2 and S2 (sometimes S2 is fuzzy) 2) Number of digits in the final sum (related to whether A1 is shown in query) Less commonly, F1 and S1 are shared in a fuzzy way. |

| | | Continuation of Table 5 |
|---|---|---|
| Entry in Table 3 | Interpretation from Table 3 | Interpretation by Claude 3 |
| $a^{0,3}$, A2 | **If** A=1: 1-2 digits from F1 and S1 **Else**: 1-2 digits from F3 and S3; C3 | **If** A=1: 1) F1 and S1 are often the same pair of digits (sometimes with a bit of variation). The order of the two digits may vary. 2) A1 is always the same digit. 3) In some cases, the presence or absence of a carry from the tens place (C2) is consistent. The digits in the ones place (F3 and S3) are usually not shared between the query and generated strings. **Else**: 1) F3 and/or S3 often have specific shared values between the query and generated strings. The shared digit is usually in the range 2-9. 2) Occasionally other patterns exist in generated strings (e.g. ones place of answer) but are not truly shared with the query. 3) Besides the ones place operands, no other consistent patterns were found across F1/S1, F2/S2 or the answer digits. |
| $x^{0,\mathrm{mid}}$, A2 | A1 **If** A1=1: F1 and S1 **Else**: F2 and S2; C3 | **If** A1=1: 1) F1 and S1 have fixed values in each example, although the values differ between examples. Their order can vary. 2) A1 is always "1" in all examples. 3) In some examples, there is a consistent carry or lack of carry from the sum of F2 and S2 (C2). But this is not true for all examples. 4) There are no consistent patterns in the values of F2, S2, F3, S3 across all examples. **Else**: A1 is always exactly shared, the middle digits F2 and S2 are often approximately shared, and usually one of the first digits (F1/S1) and one of the last digits (F3/S3) from the query appears frequently in the generated strings. |
| $x^{0,\mathrm{post}}$, A2 | A1 **If** A1=1: F1 and S1 **Else**: F2 and S2; C3 | **If** A1=1: - F1 and S1 always match, although their order can be swapped. In some cases, the carry C2 (whether F2+S2 is $\geq$ 10) is also shared between the query and generated strings. **Else**: 1) The digits in the tens place (F2, S2), either the exact digits or their sum (carry C2). 2) The ones digit of the answer (A1). 3) Sometimes the digits in the ones place (F3, S3), either the exact digits or their sum (carry C3). |
| $a^{1,0}$, A2 | **If** A1=1: F1 and S1 (sometimes fuzzy); C2 (sometimes) **Else**: 1-2 digits from F2 and S2; | **If** A1=1: *used as example in the prompt* **Else**: - F2 and S2 often have fixed values that are shared between the query and generated strings - A1 and A2 (when present) are consistently single digits in both query and generated strings |
| $a^{1,1}$, A2 | **If** A1=1: Uncertain. F1 and S1 (sometimes); 1 digit from F3 and S3 (sometimes) **Else**: F2 and S2 | **If** A1=1: - Some or all digits from the hundreds place (F1 and S1), sometimes exactly, sometimes approximately - The carry to the hundreds place (C2), in some examples - The first digit of the answer (A1) The digits in the tens and ones place (F2, S2, F3, S3) usually vary and are not consistently shared. When no generated strings are provided, the common information is uncertain. **Else**: - F2 and S2 are often fixed to specific values in the query and generated strings of each example. - The sum of F2 and S2 determines the value of C2, which is consistent between the query and generated strings within each example. - A1 is usually a single digit but can sometimes expand to 2 digits if the sum reaches or exceeds 100 in the generated strings. The specific digits of A1 are not always shared. |
| $a^{1,2}$, A2 | 1-2 digits from F1 and S1 (sometimes fuzzy); 1-2 digits from F2 and S2 (sometimes fuzzy); | - F1 and S1 often approximately match between the query and generated strings, but their exact values and order can vary. - In some cases, F2 and S2 also approximately match, but less frequently and consistently than F1 and S1. - A1 always exactly matches between the query and generated strings. |
| $a^{1,3}$, A2 | **If** A1=1: F1 and S1 (sometimes fuzzy); C2 **Else**: F2 and S2 | **If** A1=1: 1) F1 and S1, their order can vary 2) Sometimes F2 and S2, either the exact values or approximate range of values 3) A1 is often the same F3 and S3 usually don't exhibit obvious patterns. The amount of shared information varies in different examples, with some having more (e.g. exact F2 and S2 values) while others have less. **Else**: 1) One or both digits in the tens place (F2 and/or S2) 2) One or more digits in the answer (A1 and/or A2) Less frequently, there are also patterns shared in the ones place (F3 and S3). The hundreds place (F1 and S1) least often contain common patterns. |

| | | Continuation of Table 5 |
|---|---|---|
| Entry in Table 3 | Interpretation from Table 3 | Interpretation by Claude 3 |
| $x^{1,\mathrm{mid}}$, A2 | A1 **If** A1=1: F1 and S1; C2 **Else**: F2 and S2; C3 | **If** A1=1: 1. F1 and S1 are always the same two digits but their order can vary. 2. The sum of F2 and S2 is always the same, implying the carry C2 is consistent. 3. A1 is always the same digit "1". There is no consistent pattern found for F3 and S3. **Else**: 1. F1 and S1: their specific values in query often show up in generated strings in an approximate way, and their order can sometimes vary. 2. F2 and S2: the specific values or a small range of values are often shared. 3. F3 and S3: a fuzzy version is often shared, meaning the specific values may vary but are close to those in query. 4. A1: often exactly the same between query and generated strings. |
| $x^{1,\mathrm{post}}$, A2 | **If** A1=1: F1 and S1; C2 **Else**: F2 and S2; C3 | **If** A1=1: - F1 and S1 have two possible values that are swapped between F1 and S1. In other words, (F1, S1) have two possible combinations that are the reverse of each other. - A1 is always the same in the query and all generated strings in each example. Sometimes the carry C2 (whether F2+S2≥10) is also shared between query and generated strings. **Else**: 1) Specific digits are often fixed or highly common in certain places (e.g. F2, S2) across query and equations. 2) The answer digit A1 is often the same or highly consistent across query and equations. 3) The carry C2 from tens place is sometimes consistent (0 or 1) across examples. 4) Other than above, the digits and relationships seem to vary substantially between examples. |
| $a^{0,0}$, A3 | whether A1=1 | 1. Digits in the answer part (A1, A2, etc.), either the exact digit or a range. This is observed in all 5 examples. 2. The range that F1 and S1 can take, and sometimes they can swap positions. This is observed in 2 out of 5 examples. Digits in F2, S2, F3, S3 do not show consistent patterns across examples. |
| $a^{0,1}$, A3 | 1-2 digits from F2 and S2 | - When the sum is a 2-digit number, A1 is sometimes shared between query and generated strings - When the sum is a 3-digit number, A2 is often (but not always) shared - If there is a shared A2, its value often matches the query - If both A1 and A2 are present, A1A2 is between 10-19 There are no consistent patterns in the operand digits F1/S1, F2/S2, F3/S3 or their sums. |
| $a^{0,2}$, A3 | 1-2 digits from F2 and S2 | 1) S2 or F2 sometimes takes on a consistent value (like 0, 9, 7) in the query and many generated strings. 2) The equations are often incomplete, with A1 and/or A2 missing in many generated strings. 3) When present, A2 sometimes has a consistent value (like 1, 2, 6) in the query and generated strings, but not always. 4) No consistent patterns were found for the (F1, S1) and (F3, S3) digit pairs. |
| $a^{0,3}$, A3 | 1 digit from F3 and S3 (2 when F3=S3); C3 | 1) Certain digits from F3 and/or S3, although the order of F3 and S3 may vary in generated strings compared to the query. 2) The existence of A1 and A2 in complete equations, but not always. Other patterns like digits from F1, S1, F2, S2 are rarely shared. |
| $x^{0,\mathrm{mid}}$, A3 | A2; F2 and S2; C3; whether A1=1 | 1. An exact or approximate representation of (F1, S1), sometimes with order varying 2. Specific values or range of values for F2 and S2, sometimes along with their sum (C2) 3. Sometimes specific values for F3 and/or S3 4. Often an exact or approximate representation of (A1, A2) The amount and type of information shared varies between examples. But in general, there are usually some constraints on the digits in each place value shared by the query and generated strings. |
| $x^{0,\mathrm{post}}$, A3 | A2; F2 and S2; C3; whether A1=1 | 1. F1 and S1: exact match or fuzzy match, order of F1 and S1 may vary 2. F2 and S2: exact match or fuzzy match, order of F2 and S2 may vary, sometimes have size relationship (like F2 ≤ S2) 3. F3: exact match or fuzzy match 4. S3: exact match or fuzzy match 5. A1: exact match 6. A2: exact match Not every example has all the above common information. |

| Continuation of Table 5 | | |
|---|---|---|
| Entry in Table 3 | Interpretation from Table 3 | Interpretation by Claude 3 |
| $a^{1,0}$, A3 | **If** A1=1: 1-2 digits from F2 and S2; C3 (sometimes); **Else**: 1-2 digits from F3 and S3 | **If** A1=1: 1. F2 and/or S2 (sometimes their roles as F2 or S2 can vary) 2. A pattern about A1 and A2: when A1 is a specific value (usually "1"), A2 is provided; otherwise A2 is not provided 3. Sometimes A1 4. Rarely A2 when A1 is a specific value **Else**: - Fuzzy/approximate matching of F3 and S3 values - Exact matching of F3, S2, S3 values in some cases - Partial matching of A2 (last digit) in some cases Overall, ones place digits (F3, S3) and occasionally other digits (S2, A2) tend to be shared in a fuzzy or exact manner. Hundreds place digits (F1, S1, A1) are rarely shared. |
| $a^{1,1}$, A3 | **If** A1=1: 1-2 digits from F2 and S2; C3 (sometimes); **Else**: 1-2 digits from F3 and S3 | **If** A1=1: F2 and S2 are always certain fixed digits in each example, shared by the query and all generated strings in that example. Sometimes the first answer digit A1 is also shared between query and some generated strings, but this is not fully consistent. The last answer digit shows no consistency between query and generated strings. **Else**: 1) S3 (sometimes F3) 2) Fuzzy representation of (A1, A2) in some cases The ones place digit (F3 or S3) tends to be exactly preserved, while the answer (A1, A2) is sometimes preserved approximately but not exactly. Other digits and the carry do not show consistent patterns. |
| $a^{1,2}$, A3 | **If** A1=1: 1-2 digits from F2 and S2; C3 (sometimes); 1-2 digits from F1 and S1 (fuzzy); **Else**: 1-2 digits from F3 and S3 | **If** A1=1: 1. F1 and S1 (order may vary, values may be fuzzy or exact match) 2. F2 (may be exact or fuzzy match) 3. S2 (may be exact or fuzzy match, may appear in F2 or S2) 4. A1 (always exact match) 5. A2 (may be exact match or not present) The amount of shared information varies across examples. F3, S3 are not shared. **Else**: Across the examples, the most common pattern is that either F3 or S3 has a fixed single digit value shared by the query and all generated strings in that example. The specific digit varies between examples but is consistent within each example. |
| $a^{1,3}$, A3 | **If** A1=1: 1-2 digits from F2 and S2; C3 (sometimes); **Else**: 1-2 digits from F3 and S3 | **If** A1=1: 1. (F1, S1), but usually in an approximate or fuzzy way. Their exact values and order may vary in generated strings. 2. (F2, S2), usually exactly the same as in query string. In a few cases, they are shared in an approximate way. 3. A1, sometimes shared between query and generated strings. 4. S3, occasionally shared if it is present. The amount and type of shared information seem to vary from example to example. Some only share (F2, S2), while others share (F1, S1) and A1 as well, although in an approximate manner. **Else**: - The last digit of F (F3) is often shared between query and generated strings - The last digit of the answer (A2 for 2-digit answer, A3 for 3-digit answer) is very frequently shared - Sometimes S3 or other answer digits are also shared, but less reliably In general, the ones place digits of the operands and answer tend to be shared between the query and generated strings in each example. |
| $x^{1,\mathrm{mid}}$, A3 | **If** A1=1: A2; F2 and S2; C3; whether A1=1 **Else**: A2; F2 and S2; F3 and S3; whether A1=1 | **If** A1=1: F1 and S1 (sometimes fuzzy values or ranges); F2 and S2 (sometimes unordered); F3 (rarely); A1; A2 **Else**: F3 and S3, sometimes only appearing in an approximate manner and sometimes their relative order can vary; A2, always precisely shared between the query string and generated strings. |
| $x^{1,\mathrm{post}}$, A3 | **If** A1=1: A2; F2 and S2; C3; whether A1=1 **Else**: A2; F2 and S2; F3 and S3; whether A1=1 | **If** A1=1: 1. Approximate or exact values of F1 and S1, their order can vary 2. Exact values or a couple possible values of F2 and S2, their order can vary 3. Exact or approximate values of A1 and A2 The information about F3 and S3 does not seem to be shared. **Else**: - Some or all digits in F2, F3, S2, S3 (F2 and S2, F3 and S3 can be swapped) - The last digit of the answer (A2) - Whether there is a carry from tens place to hundreds place (C2) |
| $a^{0,0}$, A4/E | whether A1=1 | 1) The value of A1 (in 4 out of 5 examples) 2) An approximate or exact match of (F1, S1) pair (in 2 out of 5 examples) 3) The sum of F3 and S3 (in 1 out of 5 examples) Other occasional commonalities include the value of A2 and whether A3 is complete. |

| Continuation of Table 5 | | |
|---|---|---|
| Entry in Table 3 | Interpretation from Table 3 | Interpretation by Claude 3 |
| $a^{0,1}$, A4/E | 1-2 digits from F3 and S3 | Across the examples, the most consistent commonality is in the ones place (F3 and S3). They often have the same or very similar values in the query and generated strings. Sometimes a digit in a certain place of the final answer is also shared between query and many/all generated strings, like the first digit (A1) or last digit (A3). But this is not always the case. The hundreds place (F1/S1) and tens place (F2/S2) digits usually vary without a clear mutual pattern between the query and generated strings. |
| $a^{0,2}$, A4/E | 1-2 digits from F3 and S3 | 1) F3 and S3 digits (sometimes approximately). Present in 4 out of 5 examples. 2) A1 and/or A2 digits (only in example 0). |
| $a^{0,3}$, A4/E | 1-2 digits from F3 and S3 | - Certain digit positions (often F3 or S3) having the same value in the query and all/most generated strings - When sums are shown, certain digits (often A1 or A3) being restricted to a small set of values or range The specific digit positions and values/ranges vary between examples, but the general pattern of certain positions being fixed or constrained is consistent. |
| $x^{0,mid}$, A4/E | A3; F3 and S3; whether A1=1 | - F1 and S1 sometimes appear in generated strings with their order swapped - F3 and S3 sometimes appear in generated strings with their order swapped - F3, S3 and A3 of the query string often appear unchanged in the same position in generated strings |
| $x^{0,post}$, A4/E | A3; F3 and S3; whether A1=1 | - F3 is often shared between the query string and generated strings - S3 is sometimes shared between the query string and generated strings - A3 is consistently the same value in the query string and all generated strings for each example There is no strong pattern for F1, S1, F2 or S2. |
| $a^{1,0}$, A4/E | 1-2 digits from F3 and S3; whether next token is E | 1. Specific digits or narrow ranges in F3 and S3 2. The value or a narrow range of A1 3. The ending digits or a narrow range of ending digits in A3 4. Sometimes the carry C1 or C2 The digits in F1, S1, F2, S2 and the full value of A2, A3 are usually not shared. |
| $a^{1,1}$, A4/E | 1-2 digits from F3 and S3; whether next token is E | 1) F3 (always the same or frequently appears) 2) S3 (always the same or frequently appears) 3) One or more answer digits, especially the last one A3 (always the same or frequently appears) |
| $a^{1,2}$, A4/E | 1-2 digits from F3 and S3 | The ones digit (F3) of the query string consistently shows up in the ones place (either F3 or S3) of the generated strings. Sometimes the other ones digit in generated strings has a specific value when query F3 is in a certain place. Occasionally, the carry (C3) from the ones place addition is also shared between the query and generated strings. |
| $a^{1,3}$, A4/E | 1-2 digits from F3 and S3 | only the ones digits (F3 and sometimes S3) are consistently shared, while the other parts of the addition problems vary between the query and generated strings in each example. |
| $x^{1,mid}$, A4/E | A3; F3 and S3; whether A1=1 | - F3 and S3 have some fixed values (varying by example) that always sum to the same total. The order of the two digits doesn't matter. - As a result, A3 is always a fixed value for each example. - There is sometimes a consistent carry over amount from the tens to hundreds place, resulting in a fixed A1 value. |
| $x^{1,post}$, A4/E | F3 and S3; whether A1=1 | - Specific digits in the answer (A3) - Specific digits in the operands (F3, S3) - Whether there is a carry in a certain place (C2, C3) The shared information varies across examples, but usually relates to the ones or tens place digits and carries. |

Table 5: Interpretation for 3 digit addition produced by Claude 3, compared with human interpretation from Table 3. In general, the automated information is very informative, and the human interpretations 3 is contained in almost all cases, though the output tends to be more verbose. The LLM outputs, with some human post-checking, can thus further speed up interpretation.

## K   Compute Resources

We ran all experiments on NVIDIA A100 cards. The decoder is trained for 4-6 hours on 2 GPUs for the first three case studies, without exhaustively tuning efficiency of the implementation, which we believe could further speed up training. Regarding the factual recall task, we train the decoder

for less than 1 day on 4 GPUs. Generation of $\epsilon$-preimage samples (including forward passes on the probed model to calculate distance metric) is fast for the first 3 tasks, and it takes around 9 hours on 4 GPUs for factual recall task (for 200 query inputs). Patching experiments run quickly, as they are done for small models.

