# OpenReview forum: "InversionView: A General-Purpose Method for Reading Information from Neural Activations"
_NeurIPS.cc/2024/Conference — NeurIPS 2024 poster_

### Official Review · Reviewer_fZCU · 2024-07-08

**Soundness:** 2
**Presentation:** 2
**Contribution:** 3
**Rating:** 4
**Confidence:** 3

**Summary:**

The paper introduces the method InversionView, which finds inputs that give rise to similar activations (the preimage). As the preimage grows exponentially with sequence length the authors train a conditional decoder to generate inputs from a target activation.
The authors show three use cases: a character counting task, Indirect Object Identification, and 3-digit addition.

**Strengths:**

The approach of generating text examples that lead to similar activation patterns is intuitive and could be useful for interpretability since results are very human readable.
The example use cases (especially the toy task of character couting) seem promising.

**Weaknesses:**

- the abstract could be a bit more specific
- in general, I think the reader is left wondering for a bit too long what the authors actually do and how the method works.
- line 28: "which in turn helps us put together the algorithm implemented by the model." promises a bit too much
- motivation and description of the method is too vague
- very few details on the method/decoder that generates the examples that are later inspected
- results of experiments are a bit hard to pass. It would be nice if the reader is guided through those a bit better at different levels of abstraction
- the approach (generating human interpretable examples that achieve a specific activation patters) seems closely related to feature visualization/ adversarial example and counterfactual example generation/ in general methods that generate in the input space, however those works are not discussed at all

**Questions:**

- a graphic showing how to get from source transformer activations to pre image samples would be helpful
- Figure 2 is hard to parse: I would suggest highlighting your conclusion/interpretation (aka one activation encodes target character but not count, one activation encodes count but not target character etc) in the figure.
- in general, for the experiments, I would suggest to clearly state the task and your findings before jumping into technical details
- 3.3 "We apply InversionView to the components of the circuit" seems like you apply inversionView to specific activations found by [39] but then you state "InversionView unveils the information contained in the activation sites, with
results agreeing with those of Wang et al. [39], while avoiding the need for tailored methods."
So it seems like you needed the "tailored methods" to know where to apply inversionView in the first place, or am I misunderstanding sth?

**Limitations:**

- I think when moving from toy tasks to SOTA models, inversionView might become very prone to human tendency for pattern matching
- in addition polysemanticity might be very relevant for InversionView results but is not discussed at all
- related work is not discussed in sufficient detail

---

> ### Author Rebuttal · Authors · 2024-08-07
>
> Dear reviewer fZCU,
>
> Thanks for your feedback on our paper!
>
> ## Reply Regarding Questions
>
> ### Questions 1-3: writing and demonstrating suggestions
>
> Thanks for your suggestions. Most importantly, we have created two figures describing the decoder and the workflow, included in the global rebuttal PDF. We will also address your other two suggestions.
>
> ### It seems like you needed the "tailored methods" to know where to apply inversionView in the first place, or am I misunderstanding sth?
>
> Yes, there is a misunderstanding. There are two key parts in Wang et al. [39]: They (1) use path patching to identify important components, and then (2) study the function of these components using tailored methods. By "avoiding the need for tailored methods", we refer to (2). In order to know where to apply InversionView, we only need (1), i.e., a general-purpose method for  finding circuits.
>
> For example, [39] use path patching to identify certain heads ("S-Inhibition heads") affecting Name Mover head queries, and then use tailored patching experiments to show that they output both token and position signals. These experiments  were designed specifically to disentangle these two effects, ablating or inverting token or position information. On the other hand, InversionView directly reads out these two kinds of information (Figure 17 shows an example for an S-Inhibition head containing position information), obviating the need for guessing possible information and tailoring patching experiments.
>
> We will add this specific example to the paper.
>
> ## Reply Regarding Weaknesses
>
> ### Weaknesses 1-6: Writing and Presentation
>
> Thanks for your suggestions, which we appreciate and which we will implement.
>
> Key points:
>
> - We will provide more specific information about our studies in the abstract.
>
> - The two new figures described above will make it easy for the reader to quickly grasp how the method works.
>
> - We will change the sentence in line 28 to "which in turn helps us identify how the information flows through the model. This is crucial to obtain the algorithm implemented by the model. "
>
> - We respectfully submit that we do have formally described the method/decoder in Appendix C (decoder architecture), D.1, E.1, F.1 (decoder training details for individual tasks), and A.2 (sampling details). We will expand these sections with more discussion.
>
> - We will revise the description of results to provide more high-level intuition. We would be grateful for any more detailed guidance.
>
> ### The approach seems related to feature visualization/ adversarial example and counterfactual example generation
>
> Thanks for pointing out other methods that also generate in input space. We explain the difference between InversionView and these methods below. We will add this comparison to the paper.
>
> Feature visualization generates inputs that *maximally activate* a certain neural network unit, while InversionView finds inputs that result in the *same* vector. Whereas feature visualization interprets an individual neural network unit (e.g., neurons) to understand its general role across inputs, InversionView  interprets specific values of inner representations of neural networks. When the input changes, the value and thus the interpretation may change.
>
> Adversarial example or counterfactual example generation methods generate input that is similar to the original input but results in different outcome. While similar in input space, the adversarial/counterfactual input is likely to be quite different in internal representation space, leading to a different output. In contrast, we are interested in how different inputs in input space are represented very similarly in internal representation space.
>
> ## Reply Regarding Limitations
>
> ### On SOTA models, inversionView might become prone to human tendency for pattern matching
>
> We would like to argue that pattern matching is fundamental for interpretation. Pattern matching is essential when interpreting neurons, studying the function of an attention head, or, as done in our paper, inspecting preimages. Neural activations may contain arbitrarily complex information, hard to capture by a fixed set of templates. Therefore, analysis by an intelligent entity (artificial or biological), capable of identifying novel patterns, is necessary. As we discuss in our paper, LLMs are useful but not good enough to replace humans at this task of pattern matching yet.
>
> Moreover, InversionView is a method for generating hypotheses, not final interpretations. Generated hypotheses must be verified with intervention experiments, as we do in the paper. An incorrect hypothesis due to an error in pattern matching will be identified at this stage, ruling out dependence on subjectivity in pattern matching.
>
> ### Polysemanticity might be very relevant for InversionView results but is not discussed
>
> Thanks for raising this point. Polysemanticity is widely observed when one wants to find a unified interpretation across all inputs for a model component such as a neuron -- in contrast, InversionView interprets a specific activation (vector) on a specific input. Nonetheless, concepts similar to Polysemanticity may be defined in our setting. Specifically, we observe cases where the same (activation site, position) pair encodes different kinds of information on different inputs -- a certain kind of "polysemanticity". For example, in the factual recall task (see global rebuttal), some heads' outputs encode subject information on some inputs, and relation information on other inputs. InversionView is very helpful for studying this kind of "polysemanticity", because it decodes per-example information instead of average information. We will discuss in the paper.
>
> ### Related work is not discussed in sufficient detail
>
> We will add work related to feature visualization / adversarial example. We request the reviewer to suggest any other related work they find missing from the paper. We would be happy to include it too.

---

> > ### Comment · Reviewer_fZCU · 2024-08-10
> >
> > Thank you for addressing my concerns. After reading your rebuttal and the other reviews I have updated my score to a weak reject since it is still hard for me to assess if the limitations regarding clarity of presentation will be sufficiently addressed in the final version. However, other reviewers seemed to quite like the paper, so I will not stand in the way of accepting the paper for publication.
> > Regarding related work: I would appreciate discussing not only differences, but also similarities to related work (for example GAN inversion techniques). Many ideas there seem quite related even when the final goal is not always interpretability. This can help readers from other domains to more quickly understand your work.

---

> > > ### Author Response · Authors · 2024-08-10
> > >
> > > Thank you for raising your score, and thanks for your suggestion regarding related work. We agree with your suggestion, many readers from other fields may wonder how InversionView compares to these works, both in terms of similarities and differences. We will be happy to discuss both differences and similarities to related work.

---

### Official Review · Reviewer_P3CM · 2024-07-12

**Soundness:** 3
**Presentation:** 3
**Contribution:** 3
**Rating:** 6
**Confidence:** 3

**Summary:**

The method proposed in the paper seeks to decipher the information encoded within neural network activations. The core idea is to examine subsets of inputs that produce similar activations, utilizing a trained decoder model conditioned on these activations. The authors perform their analysis on three different tasks: character counting, indirect object identification, and 3-digit addition.

**Strengths:**

* The proposed method helps us find which parts of a network handle specific tasks. For instance, in a counting task, we can see which activations detect the target character and which ones do the counting. This is interesting because it helps us understand how neural networks work inside.

* Showing the underlying algorithm the model uses to solve tasks, like adding 3-digit numbers, gives useful insights.

**Weaknesses:**

* Applying the method to extremely large models and more complex tasks is challenging.

* Selecting an appropriate distance metric and epsilon is challenging and requires tuning, making it difficult to adapt the method to different tasks. This also highlights the authors' impressive effort in making the method work.

**Questions:**

* Do you think it is possible to use other distance metrics besides L2-based metrics? Additionally, why do you believe the L2 distance is effective in this context? Is there something unique about the geometric space of these tasks that makes L2 distance particularly suitable? What types of tasks, in general, can be effectively solved using L2-based metrics?

**Limitations:**

Yes

---

> ### Author Rebuttal · Authors · 2024-08-06
>
> Dear reviewer P3CM,
>
> Thanks for your feedback on our paper!
>
>
> ## Reply Regarding Questions
>
>
> ### It is possible to use other distance metrics besides L2-based metrics? Why do you believe the L2 distance is effective in this context? Is there something unique about the geometric space?
>
> Yes, we think it's possible to use other distance metrics. Since our work explores a novel direction, we would like to start with simple and common metrics, in order to show that the idea works without complicated metrics.
>
> Therefore, we use L2 distance because of its simplicity and straightforward intuition, so that readers can easily grasp the idea and realize how much we can learn from the geometric space, instead of regarding L2 distance as the optimal choice.
>
> In addition, in the paper we show L2-based metrics are effective empirically, as the information obtained using the simple L2-based metrics is well supported by causal and quantitative experiments.
>
> ## Reply Regarding Weaknesses
>
> ### Applying the method to extremely large models and more complex tasks is challenging
>
> We agree that interpretation of extremely large models tends to be more challenging in general. On large models and complex tasks, a potential challenge is the complexity of the information itself, but this can in principle be overcome by scaling the decoder so it can learn more complex inverse mappings, and basing the decoder on pretrained language models can help. We show the feasibility of this approach in the factual recall task on GPT2-XL (see global rebuttal), where we read information from a model 10x larger than in the IOI case study.
>
> ### Selecting an appropriate distance metric and epsilon is challenging and requires tuning
>
>
> As we mentioned above, we show that the method works with the most common and simple distance metric. There was in fact no need at all for effortful tuning of these aspects -- though using more sophisticated metrics could be a topic for follow-up work. Regarding the threshold $\epsilon$, we simply choose a small threshold for which the preimage shows meaningful patterns. Note that, unlike neural network training, where people train models repeatedly to tune hyperparameters, in the preimage returned by InversionView, one can immediately see possible interpretations resulting from different thresholds. We have more detailed discussion regarding the threshold, and our method's robustness to it, in Appendix A.4.
>
> In summary, the fact that simple distance metric and threshold choice work highlights the robustness of our method.

---

> > ### Comment · Reviewer_P3CM · 2024-08-11
> >
> > Thank you for your response. I still believe this work takes a novel approach, so I will maintain my score.

---

> > > ### Author Response · Authors · 2024-08-11
> > >
> > > Thank you. We are grateful for your support

---

### Official Review · Reviewer_vTBG · 2024-07-13

**Soundness:** 4
**Presentation:** 3
**Contribution:** 4
**Rating:** 7
**Confidence:** 4

**Summary:**

This work is mainly based on the representational geometry of activations in the activation space and chooses samples whose distances are within a defined $\epsilon$-preimage distance. A single two-layer transformer decoder model is trained using activations of each layer from the investigated model. After finishing decoder training, the Euclidean distance is used as a metric to select generated samples based on each activation presentation of each layer, and compare those samples with the query input sentence. This approach is evaluated and analysed under three tasks, i.e., character counting with a 2-layer and 1-head transformer, IOI with GPT2-small and 3-digit addition with 2-layer, 4-attention head transformer. The comprehensive experiment results and analysis confirm the geometry hypothesis, and provide lots of insights to future work.

**Strengths:**

- Activation representation of each layer from small neural networks to GPT2-small can be visualized and explained based on the geometry hypothesis with a trained decoder using those activations as input
- This hypothesis is evaluated using different tasks, i.e., character counting, IOI and 3-digit addition, and lots of meaningful analysis and insights are discussed
- Experiments are solid and the appendix includes many details about each case study.

**Weaknesses:**

- I suggest authors use another figure to demonstrate the whole training and evaluation pipeline, i.e., what input is used for training probed model, what query input is for decoder training and what input and output are from the trained decoder, which will help to understand the whole framework much easier for readers.
- This method is mainly based on manual investigation to select and analyse generated samples, which might need large labour resources to extend to large-scale LLMs.

**Questions:**

- In line 115, what does $\mathbf{z}$ represent here? and what is the difference between $\mathbf{z}$ and $\mathbf{z}^q$?
- In line 108, Is this input the same as the query input in figure 1? Another figure introducing the whole pipeline might help a lot to understand.
- Does this neighbour of geometry hypothesis always hold across all LLM activations?

**Limitations:**

I am looking forward to how to automate this approach and apply it to the larger LLM interpretability analysis.

---

> ### Author Rebuttal · Authors · 2024-08-07
>
> Dear reviewer vTBG,
>
> Thanks for your feedback on our paper!
>
> ## Reply Regarding Questions ##
>
> ### In line 115，what does $\mathbf{z}$ represent? ###
>
> Sorry for not writing it clearly. The $\mathbf{z}$ is the same as $f(\mathbf{x})$ in the previous part of the paper. So it simply represents an arbitrary activation that is compared with the query activation $\mathbf{z}^q$. We will make it clear in the next version.
>
> ### In line 108, is this input the same as the query input in figure 1? ###
>
> Yes, "input" here refers to the same as the query input in Figure 1. It's a sequence of tokens. Note that we use the modifier "query" in the context of comparing it to its neighbourhood during interpration. Thus, in line 108, we do not use "query" because the input is not mentioned in the context of a neighbourhood.
>
>
> ### Does this neighbour of geometry hypothesis always hold across all LLM activations? ###
>
>
>
> Because transformers define continuous functions, the hypothesis should hold in principle. Of course, that doesn't guarantee it will always hold in a practically meaningful sense. As experimentally demonstrated in our paper, the hypothesis holds for the conducted case studies.
>
>
> ## Reply Regarding Weaknesses ##
>
>
> ### Another figure to demonstrate the whole training and evaluation pipeline ###
>
> Thanks for your suggestion. We have made a new figure, which we have included in the global rebuttal PDF, and which we will add it to the paper in the next version.
>
> ### Need large labour resources to extend to large-scale LLMs ###
>
>
>
> 1. Neural activations can contain any kinds of information, so that any kind of template-based interpretation is likely not expressive enough. InversionView allows decoding various kinds of information. We believe manual or LLM-based interpretation is likely necessary given the variability of the information that can be encoded.
> 2. InversionView is naturally suitable for using LLMs to automate interpretation, because it produces samples in input space that LLM can easily read, rather than abstract data structures.
> 3. As our experiment (*Appendix J Automated Interpretability*) shows, LLMs can detect the main information in the preimage, despite occasional hallucination. We use manual investigation throughout in order to ensure correctness, since this is a scientific paper. We firmly believe that automated interpretation can be further improved in the future with better prompt engineering and better LLMs.
> 3. Note that, as InversionView is a method for reading information from neural activations, it facilitates reverse-engineering (which by nature requires a lot of work) but it does not require reverse-engineering to work. Our paper aims to show that InversionView is one of the tools in the ecosystem of interpretability research, rather than an all-inclusive solution. In the paper, we reverse-engineer models and provide causal verification in order to show the correctness of the information given by InversionView. The number of samples one needs to inspect to interpret a single activation vector does not necessarily increase with the model size, as we show in the case of factual recall task. For larger models, one may use it to study a specific part of the model.
> 4. Exhaustive interpretation or reverse-engineering requires a lot of work -- this is a feature of mechanistic interpretability research in general. InversionView is a faster way to generate accurate hypotheses. When combined with existing methods, InversionView makes the overall workflow more efficient, thereby decreasing guesswork and promoting faster iteration cycles in mechanistic interpretability research.

---

> > ### Comment · Reviewer_vTBG · 2024-08-12
> > **Reply by Reviewer vTBG**
> >
> > Thanks for those helpful responses. I think this work is solid and I maintain my score.

---

> > > ### Author Response · Authors · 2024-08-12
> > >
> > > Thank you. We are grateful for your support.

---

### Official Review · Reviewer_Mv2t · 2024-07-14

**Soundness:** 3
**Presentation:** 3
**Contribution:** 3
**Rating:** 6
**Confidence:** 3

**Summary:**

In this paper, the authors propose InversionView, a method to inspect the information encoded in neural activations. The proposed method is based on checking the activations difference given different inputs. The authors showcase the effectiveness of this tool on mainly three tasks: character counting, indirect object identification, and 3-digit addition.

Strengths

1. The topic of model interpretation is important.
2. The empirical results confirm the effectiveness of the proposed method.
3. The paper is well written and easy to follow. In addition, the authors provide abundant experiments to showcase the effectiveness.

Weaknesses

1. Although being described through text, the precise algorithm for the proposed method is unclear to me.
2. It is unknown how the proposed method scales with larger models. The largest model size in the paper is limited to GPT-2, which is not considered large nowadays.

Overall, I think the paper is interesting and recommend a weak acceptance. I will consider raising my ratings if the above weaknesses are addressed.

**Strengths:**

1. The topic of model interpretation is important.
2. The empirical results confirm the effectiveness of the proposed method.
3. The paper is well written and easy to follow. In addition, the authors provide abundant experiments to showcase the effectiveness.

**Weaknesses:**

1. Although being described through text, the precise algorithm for the proposed method is unclear to me.
2. It is unknown how the proposed method scales with larger models. The largest model size in the paper is limited to GPT-2, which is not considered large nowadays.

**Questions:**

See the weaknesses.

**Limitations:**

The authors have adequately addressed the limitations.

---

> ### Author Rebuttal · Authors · 2024-08-06
>
> Dear reviewer Mv2t,
>
> Thanks for your feedback on our paper!
>
> ## Reply Regarding Weaknesses
>
> ### The precise algorithm for the proposed method is unclear
>
> In the global rebuttal, we provide a new figure showing the training and sampling pipelines. In its caption, we also provide detailed explanation.
>
> If you are unclear about the the decoder. We also provide a new figure showing decoder architecture in global rebuttal. The decoder model is basically a decoder-only transformer combined with some additional MLP layers. In order to condition the decoder on the query activation, the query activation is first passed through a stack of MLP layers to decode information depending on the activation sites and then made available to each attention layer of the transformer part of the decoder. There are also details about it in Appendix C in the paper (we should have linked section 2.2 to that).
>
> We will add the new figures to the next version of the paper.
> Taken together, these changes will make the precise algorithm much more accessible to the reader.
>
>
>
> ### How the proposed method scales with larger models
>
> Thank you for raising this point. Please refer to the global rebuttal, where we describe our new experiment on a larger model, with 10x more parameters than in the IOI case study.  We find that InversionView continues to produces interpretable results, and allows us to read out interesting information content.

---

> > ### Comment · Reviewer_Mv2t · 2024-08-11
> >
> > Thank you for your response. I remain my stance to accept this paper.

---

> ### Author Response · Authors · 2024-08-11
>
> Thank you. We are grateful for your support.

---

### Author Rebuttal · Authors · 2024-08-07

We thank all reviewers for their reviews. We are encouraged that they found our method to be effective (Reviewer Mv2t) and to provide useful and intuitive insights (Reviewer vTBG, P3CM, fZCU), our experiments solid (Reviewer vTBG), and the paper well-written and easy to follow (Reviewer Mv2t).

We have addressed the weaknesses and specific questions from each reviewer in the individual responses. Below, we address two important points that are relevant to multiple reviews.

## Regarding writing and demonstration suggestions ##
We thank the reviewers for pointing out various potential improvements in writing and presentation.
We apologize for compressed writing; as we have a lot of content but limited space, we had moved many details to appendix. We will revise text for better readability, and add figures to make presentation more straightforward. If the paper is accepted, with an additional page, we believe we can further improve the paper in this respect.

Most importantly, besides improving the writing, we provide a figure for the overall pipeline, and a figure for the decoder architecture; both are shown in the attached PDF (Figures 1 and 2). By illustrating the InversionView workflow, this concretely addresses key concerns of Reviewers Mv2t, vTBG, fZCU.

## A new case study on a larger model ##

Several reviewers (Mv2t, vTBG, P3CM) asked about applicability to larger models. Since submission, we have done experiments on a larger model, GPT-2 XL, which has 1.5B parameters --- a 10-fold increase over the maximum model size in the submission.  InversionView continues to produce interpretable results, and allows us to read out high-level information content. We've also updated the web application, so you can check results via the link provided in the paper (https://inversion-view.streamlit.app). We will incorporate the content about this experiment in the paper.

Below, we provide more details on the factual recall task, decoder training and sampling. We will include these in the final version of the paper. We observed many interesting examples. Due to space limitations, we put only one figure for it in the rebuttal PDF, but we strongly recommend playing around with our webAPP.


### Factual Recall: Background  ###

The factual recall task is defined as predicting the attribute given a prompt containing the subject and relation.
The model is given a prompt such as *"LGA 775 is created by"*,  containing the subject *"LGA 775"* and the relation *"is created by"*. The model predicts the next token *"Intel"*,  an attribute of the subject. This task requires retrieving relevant knowledge, and we may expect  that neural activations contain high-level concepts.

Previous work [1] suggests that, in attention layers of the upper part of the model, attributes of the subject are moved to the residual stream of the last token.

### Factual Recall: Implementation Details  ###

In this case study, our intention is not to provide a full interpretation of the computations performed to solve this task, which we deem out of scope for this paper. Rather, we focus on a relatively small set of important attention heads in upper layers, and check if InversionView produces interpretable results. We select the 25 most important attention heads in the upper part of the model, as prior work found that attribute retrieval tends to happen there.[1] We estimate the importance of attention heads by the attribution method from Ferrando et al.[2]





The decoder model is fine-tuned from GPT-2 Medium (the components for processing query activation are randomly initialized), because we expect a more complex inverse mapping from activation to inputs to be learned. Concretely, to interpret activations encoding a certain attribute, the decoder may need to memorize knowledge about different subjects sharing the same attribute.

To train the decoder model, we collect text from 3 datasets: factual statements from COUNTERFACT[3] and BEAR[4], and general text from MiniPile[5].

We trained the decoder on outputs of the selected attention heads. We filtered out query activations resulting from heavily attending to BOS (weight > 0.6), which occurs in many attention heads in higher layers but likely results in attention outputs with little information.


We use the same distance metric as in the IOI study, and set  $\epsilon=0.25$. As mentioned in the paper, a larger threshold produces more coarse-grained information; we found this to provide more stable results on this large model, potentially reflecting its higher-dimensional geometry.

### Factual Recall: Observation  ###

We provide a sample figure in the attached PDF.
We observed many interesting examples, and strongly recommend playing around with our webAPP. When doing so, you may be interested in different kinds of information. When not attending exclusively to BOS, head $h^{29,9}$, $h^{30.8}$, $h^{32,12}$, $h^{33,0}$, $h^{37.7}$ almost always move information about the subject, and $h^{24,24}$, $h^{25,7}$, $h^{27,16}$, $h^{28,21}$, $h^{29,20}$, $h^{33,9}$ almost always move information about the relation, while other heads usually exhibit a mix of behaviors (see the complete list of heads in our webAPP).

[1]: Geva, Mor, et al. "Dissecting recall of factual associations in auto-regressive language models." arXiv preprint arXiv:2304.14767 (2023).

[2]: Ferrando, Javier, and Elena Voita. "Information flow routes: Automatically interpreting language models at scale." arXiv preprint arXiv:2403.00824 (2024).

[3]: Meng, Kevin, et al. "Locating and editing factual associations in GPT." Advances in Neural Information Processing Systems 35 (2022): 17359-17372.

[4]: Wiland, Jacek, Max Ploner, and Alan Akbik. "BEAR: A Unified Framework for Evaluating Relational Knowledge in Causal and Masked Language Models." arXiv preprint arXiv:2404.04113 (2024).

[5]: Kaddour, Jean. "The minipile challenge for data-efficient language models." arXiv preprint arXiv:2304.08442 (2023).

---

### Decision · Program_Chairs · 2024-09-25

**Decision:**

Accept (poster)

**Comment:**

After carefully reading the reviews and the rebuttal, I feel the authors manage to answer most if not all of the concerns raised by the reviewers. And as such, I think the quality of the manuscript increases sufficiently for it to be considered for acceptance.
In particular, reviewer fZCU, with the lowest score, most concerns seem to be around presentation and clarity. The rebuttal provides answers to the punctual questions raised by fZCU in detail and explain how the draft will be improved. I believe these changes will answer the concerns raised. But I strongly urge the authors to incorporate all the changes they suggested in the rebuttal and additional experiments, as well as improving the clarity of the manuscript in the camera ready version of the paper. Improving these points will be important for the paper to have the impact it deserves once published.